# The helicase domain of human Dicer prevents RNAi-independent activation of antiviral and inflammatory pathways

Morgane Baldaccini ⓘD, Léa Gaucherand ⓘD, Béatrice Chane-Woon-Ming, Mélanie Messmer ⓘD, Floriane Gucciardi & Sébastien Pfeffer ⓘD ✉

## Abstract

In mammalian somatic cells, the relative contribution of RNAi and the type I interferon response during viral infection is unclear. The apparent inefficiency of antiviral RNAi might be due to self-limiting properties and mitigating co-factors of the key enzyme Dicer. In particular, the helicase domain of human Dicer appears to be an important restriction factor of its activity. Here, we study the involvement of several helicase-truncated mutants of human Dicer in the antiviral response. All deletion mutants display a PKR-dependent antiviral phenotype against certain viruses, and one of them, Dicer N1, acts in a completely RNAi-independent manner. Transcriptomic analyses show that many genes from the interferon and inflammatory response pathways are upregulated in Dicer N1 expressing cells. We show that some of these genes are controlled by NF-kB and that blocking this pathway abrogates the antiviral phenotype of Dicer N1. Our findings highlight the crosstalk between Dicer, PKR, and the NF-kB pathway, and suggest that human Dicer may have repurposed its helicase domain to prevent basal activation of antiviral and inflammatory pathways.

**Keywords** Dicer; Innate Immunity; NF-kB; RNAi; Virus
**Subject Categories** Immunology; Microbiology, Virology & Host Pathogen Interaction; RNA Biology

## Introduction

RNAi is an evolutionary conserved cellular defense mechanism against invading nucleic acids. It is based on the detection and cleavage of a trigger double-stranded (ds) RNA by Dicer, a type III ribonuclease. The resulting small interfering (si) RNAs serve as guides for effector Argonaute proteins that will act on target RNAs in a sequence specific manner (Meister and Tuschl, 2004). By its nature, RNAi has been shown to be a prominent defense mechanism against viruses in plant and invertebrate organisms

(Guo et al, 2019; tenOever, 2016), but its exact contribution to the innate antiviral response in mammals remains to be firmly established. Indeed, several studies failed to identify virus-derived siRNAs in infected mammalian somatic cells (Girardi et al, 2013; Backes et al, 2014; Parameswaran et al, 2010). It was also reported that molecular mechanisms, such as poly-ADP-ribosylation of AGO2, could prevent the activation of RNAi upon viral infection (Seo et al, 2013) or that when viral siRNAs did accumulate, they were inefficiently loaded into Argonaute proteins (Tsai et al, 2018). However, antiviral RNAi does appear to be active in specific conditions, in particular in pluripotent stem cells (Maillard et al, 2013). It also seems to be incompatible with other innate immune defense pathways that rely on the type I interferon response (IFN-I). As a result, certain studies have shown that inactivating the IFN-I response allows RNAi to take over. The pattern recognition receptors that are inactivated in these studies are members of the RIG-I like receptor (RLR) family and they also recognize dsRNA (Ahmad and Hur, 2015; Pichlmair and Reis e Sousa, 2007). However, as opposed to Dicer, they do not act by cleaving dsRNA but rather by inducing a phosphorylation cascade that will ultimately result in the production of the autocrine- and paracrine-acting interferon α and ß cytokines followed by the transcriptional activation of hundreds of interferon stimulated genes (ISGs) (Ivashkiv and Donlin, 2014). Thus, in mammalian somatic cells deficient for MAVS or IFNAR, the production of long-dsRNA-derived siRNAs can be detected and are shown to be dependent on Dicer activity (Maillard et al, 2016). Besides, the RLR LGP2 has been shown to directly interact with Dicer, blocking siRNA production and along with TRBP preventing the correct miRNA maturation (Takahashi et al, 2018b, 2018a; van der Veen et al, 2018). Finally, a recent study showed that the IFN-induced adenosine deaminase ADAR1 could compete with RNAi to act on dsRNA accumulating during a viral infection (Uhl et al, 2023). Thus, in mammalian cells, RNAi seems to be functionally incompatible with other innate immune mechanisms, which are suggested to tone down Dicer involvement in the antiviral response.

One interesting observation is that the amino-terminal helicase domain is quite conserved between RLRs and Dicer, which might indicate a functional replacement during evolution (Baldaccini and

Université de Strasbourg, Architecture et Réactivité de l'ARN, Institut de Biologie Moléculaire et Cellulaire du CNRS, 67000 Strasbourg, France.
✉E-mail: s.pfeffer@ibmc-cnrs.unistra.fr

Pfeffer, 2021). In mammalian Dicer, the helicase domain exerts some molecular constraints that limit Dicer's cleavage processivity on long dsRNA molecules (Ma et al, 2008). Accordingly, a synthetic Dicer lacking the first two domains of the helicase, Hel1 and Hel2i, and named Dicer N1 displays better cleavage properties of an artificial dsRNA than the full-length version of Dicer (Kennedy et al, 2015). More recently, a naturally occurring isoform of Dicer, lacking the Hel2i domain, has been identified in human stem cells and shown to possess some antiviral properties in an RNAi-dependent manner (Poirier et al, 2021). Similarly, a Hel1-truncated version of Dicer, coined Dicer$^O$, is specifically expressed in mouse oocytes and is better adapted to cleave long dsRNA molecules than pre-miRNAs (Flemr et al, 2013). Uncovering the structure of Dicer$^O$ isoform revealed that the N-terminal part of the helicase domain plays an important role for the correct positioning of the pre-miRNA and that removing it potentially increases the affinity for longer dsRNA structures (Zapletal et al, 2022). On top of having a molecular self-limiting effect, this helicase domain also allows Dicer to interact with proteins that regulate its activity. We have recently shown that during Sindbis virus (SINV) infection the helicase domain of human Dicer specifically interacts with proteins that are involved in the IFN-I response such as PACT, the RNA helicase DHX9, ADAR1 and the dsRNA-activated protein kinase PKR (Montavon et al, 2021). In addition, we reported in the same study that the expression of the helicase-truncated Dicer N1 confers a strong antiviral activity against SINV, and that this phenotype is dependent on the presence of PKR. Similarly, another study showed a genetic link between Dicer and PKR in mouse embryonic stem cells and proposed that Dicer can hinder PKR activity in a non-canonical manner (Gurung et al, 2021). Dicer thus seems to have an additional antiviral activity linked to PKR that is limited by its helicase domain.

In this study, we sought to uncover the mechanism underlying Dicer N1 antiviral activity. By using HEK293T NoDice cells (Bogerd et al, 2014) stably expressing either Dicer WT or N1, we showed that Dicer N1 cells are more resistant than Dicer WT cells to infection with alphaviruses SINV and Semliki forest virus (SFV), and human enterovirus 71 (EV71). However, this antiviral effect was virus-dependent, as Dicer N1 had no impact on vesicular stomatitis virus (VSV) infection, and we even observed a proviral effect of Dicer N1 on SARS-CoV-2 infection. Interestingly, we found that the antiviral effect of Dicer N1 is RNAi-independent, as cells expressing a catalytically inactive Dicer N1 were equally protected against SINV infection. We also tested other helicase-truncated mutants of Dicer and observed that individual deletion of each subdomain (Hel1, Hel2i, or Hel2) also conferred an antiviral property to Dicer. However, as opposed to Dicer N1, this phenotype appeared to be partially due to RNAi. Nevertheless, all of the helicase-deletion mutants also required the presence of PKR to maintain their antiviral effect. We thus focused our investigations on the implication of PKR in the phenotype of Dicer N1 and showed that it is required for its antiviral activity independently from its kinase function. Transcriptomic analysis of mock- or SINV-infected cells uncovered that Dicer N1 expressing cells have a higher basal expression of a large subset of genes, including a number that are involved in the antiviral response. We further showed that those genes are under the control of transcription factors such as STAT1, STAT2, and NF-kB/p65. Finally, we

confirmed the importance of the latter pathway in the phenotype of Dicer N1 cells by alleviating the antiviral effect in cells treated with a chemical inhibitor of NF-kB. Based on these results, we propose that one reason for human Dicer to have maintained a self-limiting helicase domain is to prevent basal activation of antiviral and inflammatory pathways.

# Results

## Effect of Dicer N1 expression on SINV viral cycle

We previously showed that a partial helicase-deletion mutant of human Dicer expressed in HEK293T NoDice cells presented an antiviral activity against SINV (Montavon et al, 2021). We used the two monoclonal cell lines NoDice FHA-Dicer WT #4 and NoDice FHA:Dicer N1 #6 (afterward respectively referred to as Dicer WT and Dicer N1 cells) to better characterize the impact of Dicer WT or Dicer N1 expression on SINV viral life cycle. We used a non-modified SINV strain and two different GFP-expressing strains that either express the fluorescent protein from a duplication of the subgenomic promoter (SINV-GFP) or from a fusion with the capsid protein (SINV-2A-GFP) (Thomas et al, 2003). The latter expressed GFP at higher levels than the former (Appendix Fig. S1A). We infected Dicer WT and Dicer N1 cells with increasing MOIs of all three viruses and measured the accumulation of infectious particles by plaque assay 24 h post-infection (hpi) (Fig. 1A). For all viruses, we observed a significant antiviral effect at every MOI in cells expressing Dicer N1 compared to cells expressing Dicer WT. The effect was more pronounced in cells infected with the SINV-2A-GFP, which seems to be attenuated at lower MOIs compared to both other viral strains (Fig. 1A). We also monitored viral protein accumulation in cells infected for 24 h at an MOI of 0.02. In Dicer N1 cells, both the SINV Capsid and GFP proteins accumulated to lower levels, or even below detection limits for SINV-2A-GFP, compared to Dicer WT cells (Fig. 1B). We also monitored dsRNA accumulation during infection as a proxy for viral replication. We used the dsRNA-specific J2 antibody (Richardson et al, 2010) to perform immunostaining of cells infected for 24 h at an MOI of 0.02 with either of the three SINV strains. Compared to Dicer WT cells, Dicer N1 cells showed a decreased dsRNA accumulation for all three viruses, with a stronger effect for the SINV-2A-GFP virus (Fig. 1C). In order to verify that the drop in viral accumulation in Dicer N1 cells was not simply due to a delay in replication, we also measured viral titers by plaque assay in cells infected for 48 h with SINV-GFP at two different MOIs (0.01 and 0.001) and for 72 h at an MOI of 0.001. In all cases, we could still observe a significant decrease in viral titers in Dicer N1 cells compared to Dicer WT cells (Appendix Fig. S1B). Finally, to further confirm the impact of Dicer N1 expression on viral replication, we measured the accumulation of genomic RNA by RT-qPCR in cells infected with SINV-GFP at an MOI of 0.02 for 24 h. In agreement with the previous results, Dicer N1 cells showed a very strong decrease of SINV genomic RNA accumulation compared to Dicer WT cells (Fig. 1D). In parallel, we also performed semi-quantitative strand-specific RT-PCR on the same samples and observed that the antigenomic viral RNA accumulated to lower levels in Dicer N1 cells than in Dicer WT cells (Appendix Fig. S1C). Collectively, these data indicate that the antiviral activity of Dicer N1 leads to a defect in SINV viral replication.

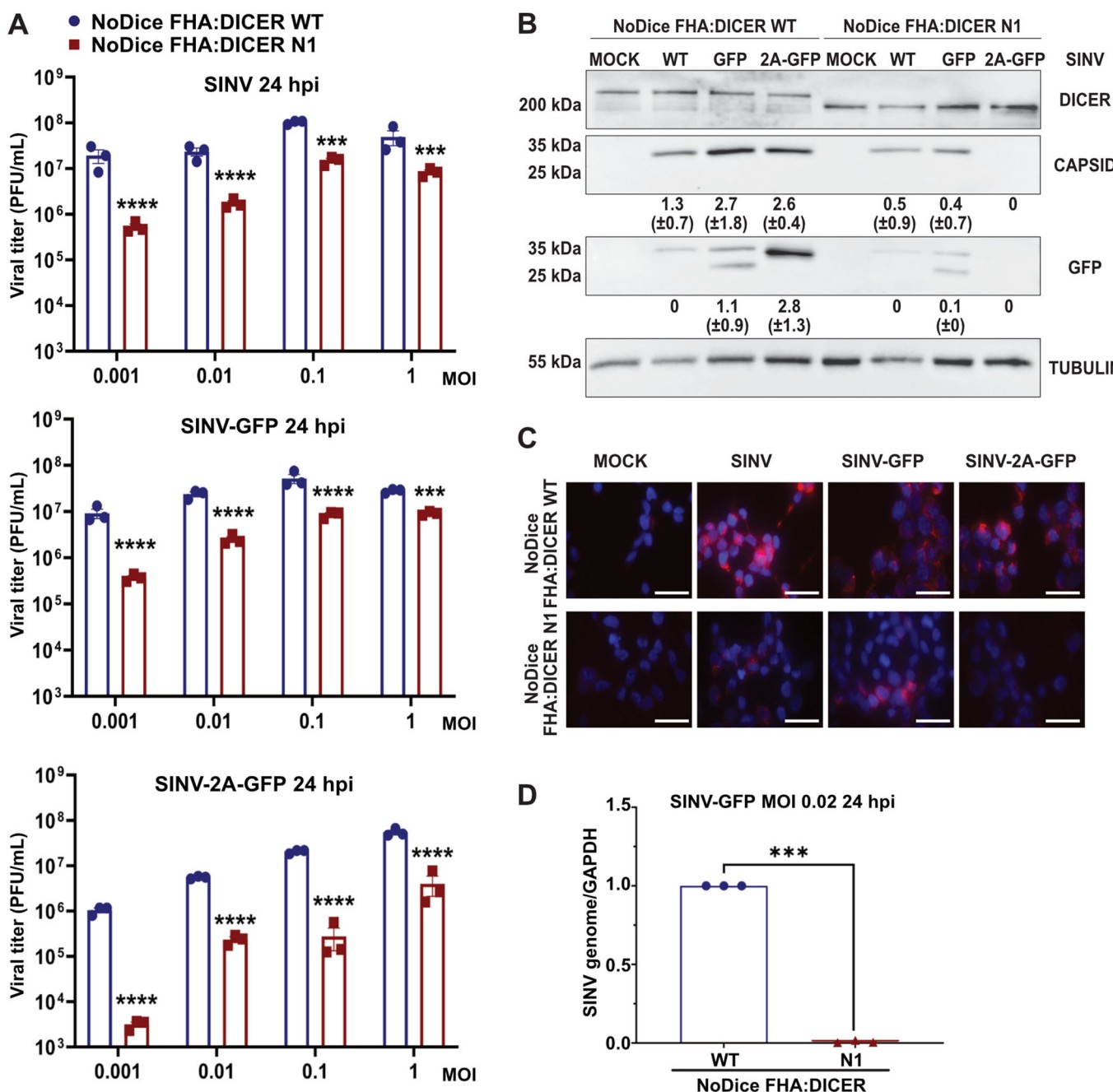

**Figure 1. Dicer N1 decreases SINV viral accumulation in human cells.**

(A) Mean ($+/-$ SEM) of SINV, SINV-GFP, and SINV-2A-GFP viral titers in NoDice FHA:DICER WT and N1 cells infected at an MOI ranging from 0.001 to 1 for 24 h ($n = 3$ biological replicates) from plaque assay quantification. Ordinary two-way ANOVA test with Sidak's correction. \*\*\*$p < 0.001$; \*\*\*\*$p < 0.0001$. (B) Western blot analysis of DICER, CAPSID, and GFP expression in SINV, SINV-GFP, and SINV-2A-GFP infected NoDice FHA:DICER WT and N1 cells at an MOI of 0.02 for 24 h. Alpha-Tubulin was used as loading control. Band intensity was quantified and normalized to Tubulin for three independent biological replicates. Mean values ($+/-$ SD) are indicated under the corresponding lane. (C) Immunofluorescence analysis on NoDice FHA:DICER WT and N1 cells in mock, SINV, SINV-GFP and SINV-2A-GFP infected cells at an MOI of 0.02 for 24 h. J2 antibody (red) was used to detect dsRNA upon infection. DAPI was used to stain the nuclei (blue). Scale bar $= 100$ µm, $n = 3$ biological replicates. (D) RT-qPCR on SINV genome in NoDice FHA:DICER WT and N1 cells infected with SINV-GFP at an MOI of 0.02 for 24 h. Mean ($+/-$ SEM); $n = 3$ biological replicates. Unpaired t-test. \*\*\*$p < 0.001$. Source data are available online for this figure.

## Dicer N1 is antiviral against other positive-strand RNA viruses

To see whether the antiviral activity of Dicer N1 could be generalized to other viruses, we infected Dicer WT and Dicer N1 cells with viruses from different families. We first tested another *Togaviridae* family member, the Semliki forest virus (SFV), and

observed a lower viral titer in the supernatants of Dicer N1 cells than in Dicer WT cells infected for 24 h at an MOI of $1.10^{-4}$ (Fig. 2A, top panel). In addition, dsRNA accumulation, as assessed by J2 immunostaining, was barely detectable in Dicer N1 cells, whereas it was very high in Dicer WT cells (Fig. 2A, bottom panel). We then infected Dicer WT and N1 cells at an MOI of 0.1 with a (+) RNA virus from the *Picornaviridae* family, human EV71, and

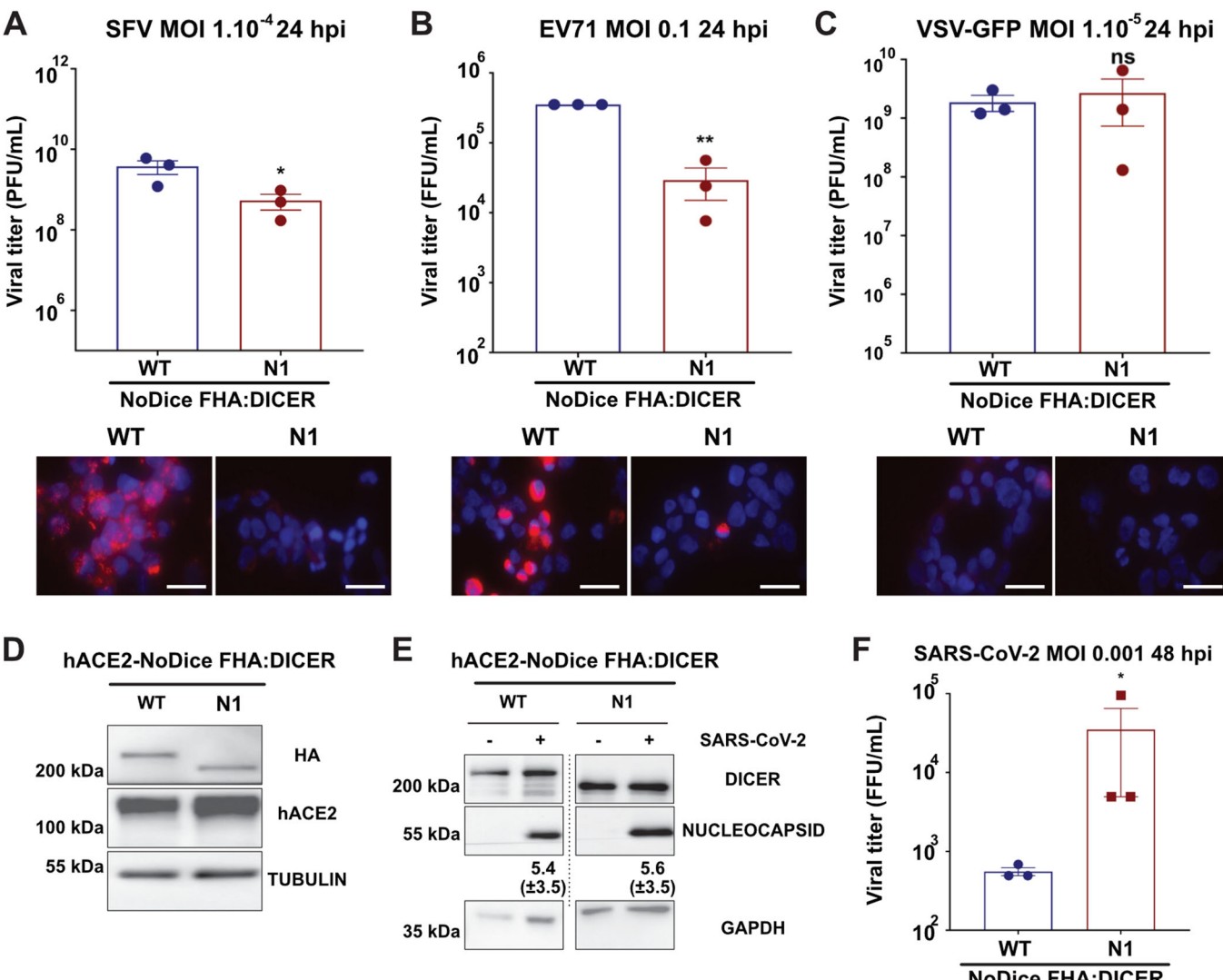

**Figure 2.  Dicer N1 is antiviral against alphaviruses and an enterovirus but not against a rhabdovirus and a coronavirus.**

(A) (Top) Mean ($+/-$ SEM) of SFV viral titers in NoDice FHA:DICER WT and N1 cells infected at an MOI of $1.10^{-4}$ for 24 h ($n = 3$ biological replicates) from plaque assay quantification. Unpaired t-test with Welch's correction. $*p < 0.05$. (Bottom) Immunofluorescence analysis on NoDice FHA:DICER WT and N1 cells in SFV infected cells. J2 antibody (red) was used to detect dsRNA upon infection. DAPI was used to stain the nuclei (blue). Scale bar = 100 μm, $n = 3$ biological replicates. (B) (Top) Mean ($+/-$ SEM) of EV71 viral titers in NoDice FHA:DICER WT and N1 infected at an MOI of 0.1 for 24 h ($n = 3$ biological replicates) from TCID50 quantification. Unpaired t-test. $**p < 0.01$. (Bottom) Immunofluorescence analysis on NoDice FHA:DICER WT and N1 cells in EV71 infected cells. J2 antibody (red) was used to detect dsRNA upon infection. DAPI was used to stain the nuclei (blue). Scale bar = 100 μm, $n = 3$ biological replicates. (C) (Top) Mean ($+/-$ SEM) of VSV-GFP viral titers in NoDice FHA:DICER WT and N1 infected at an MOI of $1.10^{-5}$ for 24 h ($n = 3$ biological replicates) from plaque assay quantification. Unpaired t-test. ns: non-significant. (Bottom) Immunofluorescence analysis on NoDice FHA:DICER WT and N1 cells in VSV-GFP infected cells. J2 antibody (red) was used to detect dsRNA upon infection. DAPI was used to stain the nuclei (blue). Scale bar = 100 μm, $n = 3$ biological replicates. (D) Western blot analysis of human ACE2 and HA (for DICER) expression in NoDice FHA:DICER WT and N1 cells ($n = 3$ biological replicates). Alpha-Tubulin was used as loading control. (E) Western blot analysis of DICER and NUCLEOCAPSID expression in SARS-CoV-2 infected NoDice FHA:DICER WT and N1 cells at an MOI of 0.001 for 48 h. GAPDH was used as loading control. Band intensity was quantified and normalized to GAPDH for three independent biological replicates, then represented as mean ($+/-$ SD) under the corresponding lane. (F) Mean ($+/-$ SEM) of SARS-CoV-2 viral titers in NoDice FHA:DICER WT and N1 infected at an MOI of 0.001 for 48 h ($n = 3$ biological replicates) from TCID50 quantification. Unpaired t-test. $*p < 0.05$. Source data are available online for this figure.

measured viral titer in the supernatant by TCID50 at 24 hpi. We observed that the EV71 titer was again lower in Dicer N1 compared to WT cells (Fig. 2B, top panel). J2 immunostaining also confirmed a lower accumulation of dsRNA in Dicer N1 cells compared to WT cells (Fig. 2B, bottom panel). We also tested the effect of Dicer N1 on a single-stranded negative-sense RNA virus from the *Rhabdoviridae* family, i.e. vesicular stomatitis virus (VSV). We infected Dicer WT and N1 cells at an MOI of $1.10^{-5}$ for 24 h with an engineered VSV expressing the GFP protein and titrated the virus in the supernatant by plaque assay. We did not detect any significant difference in viral titers between Dicer WT and N1 cells (Fig. 2C, top panel). As previously reported (Weber et al, 2006), we were not able to detect dsRNA accumulation by J2 immunostaining, in either Dicer WT or N1 cells infected with VSV-GFP. Since dsRNA is a canonical Dicer substrate upon infection, this result indicates that the absence of dsRNA during VSV infection might prevent the activation and antiviral activity of Dicer N1.

Finally, we infected the two cell lines with the single-stranded positive-sense RNA virus Severe Acute Respiratory Syndrome Coronavirus 2 (SARS-CoV-2). To be able to use this virus in the HEK293T cells we work with, we first transduced them with a lentiviral vector expressing human ACE2 (hACE2), which is an essential receptor for SARS-CoV-2 infection (Hoffmann et al, 2020). We verified hACE2 expression by western blot with a specific antibody (Fig. 2D). We infected both hACE2-expressing Dicer WT and Dicer N1 cells with SARS-CoV-2 at an MOI of $1.10^{-3}$ for 48 h, and measured viral nucleocapsid expression by western blot analysis. Nucleocapsid expression could be detected in both Dicer WT and N1 cells and seemed to be slightly higher in the latter (Fig. 2E). Then, we measured viral titer in the supernatant by TCID50 at 48 hpi. We observed that the SARS-CoV-2 titer was higher in Dicer N1 compared to WT cells (Fig. 2F). We also performed J2 immunostaining on SARS-CoV-2 infected cells, and saw that dsRNA was detected equally well in both Dicer WT and N1 cells (Appendix Fig. S2A). Finally, we also measured viral RNA accumulation by RT-qPCR, and saw that both ORF1a and Spike RNAs accumulated to higher levels in Dicer N1 than in Dicer WT cells, although the increase was not significant for Spike RNA (Appendix Fig. S2B).

Altogether, these last results suggest that Dicer N1 expression can have an antiviral effect for two different (+) RNA viruses (SFV and EV71), and a proviral effect for a third one (SARS-CoV-2). This antiviral property does not seem to be active in the case of a (-) RNA virus, VSV, which might be related in this case to the difference in dsRNA accessibility.

## RNA interference is not involved in Dicer N1 phenotype

We showed that Dicer N1 expression prevents the accumulation of viral genomic and antigenomic RNAs during SINV infection and of dsRNA during SINV, SFV, and EV71 infections. Since it was previously shown that Dicer N1 was more active than Dicer WT to cleave an artificial dsRNA into siRNAs (Kennedy et al, 2015), we first hypothesized that the ribonuclease activity of Dicer N1 was responsible for its antiviral activity. To test this hypothesis, we used an affinity-based purification approach (Hauptmann et al, 2015) to isolate Ago-associated small RNAs from Dicer WT or N1 cells infected with SINV-GFP at an MOI of 0.02 for 24 h and deep-sequenced them. We first mapped the reads to the human and

SINV-GFP genomes and observed that the vast majority of small RNAs had a cellular origin (Fig. EV1A). Only 0.2 to 1.2% of sequences could be mapped to the viral genome. Interestingly, the percentage of viral reads was four-to-five-fold lower in Dicer N1 cells than in WT cells. We further analyzed the viral reads and determined their size distribution. Small RNAs mapped to both viral genomic and antigenomic RNAs, with a bias in favor of the former, and showed a peak at 22 nt, consistent with a Dicer-mediated cleavage (Fig. 3A, Appendix Fig. S3A). However, there was no real difference in size distribution or strand origin between Dicer WT and N1 cells. Small RNAs mapped to the viral genome showed a strong enrichment at the 5' extremity of the genomic RNA in both Dicer WT and N1 cells (Fig. 3B, Appendix Fig. S3B), in agreement with previous reports (Kong et al, 2023; Zhang et al, 2021). The relative number of reads that mapped to both strands of the viral RNA was similar in Dicer WT and N1 cells. It therefore appears that both Dicer WT and Dicer N1 are competent for the generation of what looks like viral siRNAs that are loaded in Argonaute proteins, but we did not see an increase in their number in Dicer N1 expressing cells. At this stage, we therefore cannot say that the observed antiviral phenotype of Dicer N1 is linked to a stronger RNAi activity. We also checked whether the expression of Dicer N1 had an impact on the miRNA profile. The analysis of the 50 most abundant miRNAs retrieved in the sequencing data did not reveal striking differences between Dicer WT and N1 cells (Fig. EV1B). Similarly, the global analysis of all miRNAs identified in the libraries showed that some lowly abundant miRNAs were differentially expressed, but when looking at miRNAs expressed at a level sufficient for exerting a physiologically relevant effect the observed differences were minimal (Fig. EV1C).

To further confirm that the Dicer N1 phenotype was independent of its role in RNAi, we generated a catalytic-deficient version of Dicer N1 by introducing mutations in both of its RNAse III domains (Fig. 3C). As previously, we used the HEK293T NoDice 2.20 cell line (Bogerd et al, 2014) that we transduced with lentiviral vectors expressing a Flag-HA-tagged version of Dicer N1 with mutations in the catalytic domain (N1-CM). We then selected a clone, NoDice FHA:Dicer N1-CM #2.17 (afterward called Dicer N1-CM), that expressed Dicer at levels similar to the Dicer WT and Dicer N1 cell lines previously generated (Fig. 3D). The levels of PKR and TRBP were mostly similar to the ones in Dicer N1 cells. By blotting AGO2 in these cell lines, we observed that it was absent in the Dicer N1-CM cells, which is expected for a miRNA-free Argonaute protein and is consistent with previous observations (Gibbings et al, 2012). We confirmed the defect in miRNA processing in the Dicer N1-CM cells by measuring miR-16 accumulation by northern blot analysis. As expected, we saw no mature miR-16 form and the accumulation of its precursor in Dicer N1-CM cells (Fig. 3E). We then infected Dicer WT, N1, and N1-CM cells with SINV-GFP at an MOI of 0.02 for 24 h and measured the infection rate by western blot analysis and plaque assay. The level of the capsid protein was strongly reduced in both Dicer N1 and Dicer N1-CM cells compared to Dicer WT cells (Fig. 3F). To confirm these observations, we titrated the infectious viral particles in the supernatant and observed a strong and significant decrease of viral titers in both Dicer N1 and Dicer N1-CM compared to Dicer WT cells (Fig. 3G). To rule out any clone-dependent effect, we performed the same experiment in polyclonal cells expressing Dicer WT, N1, or N1-CM. As for the

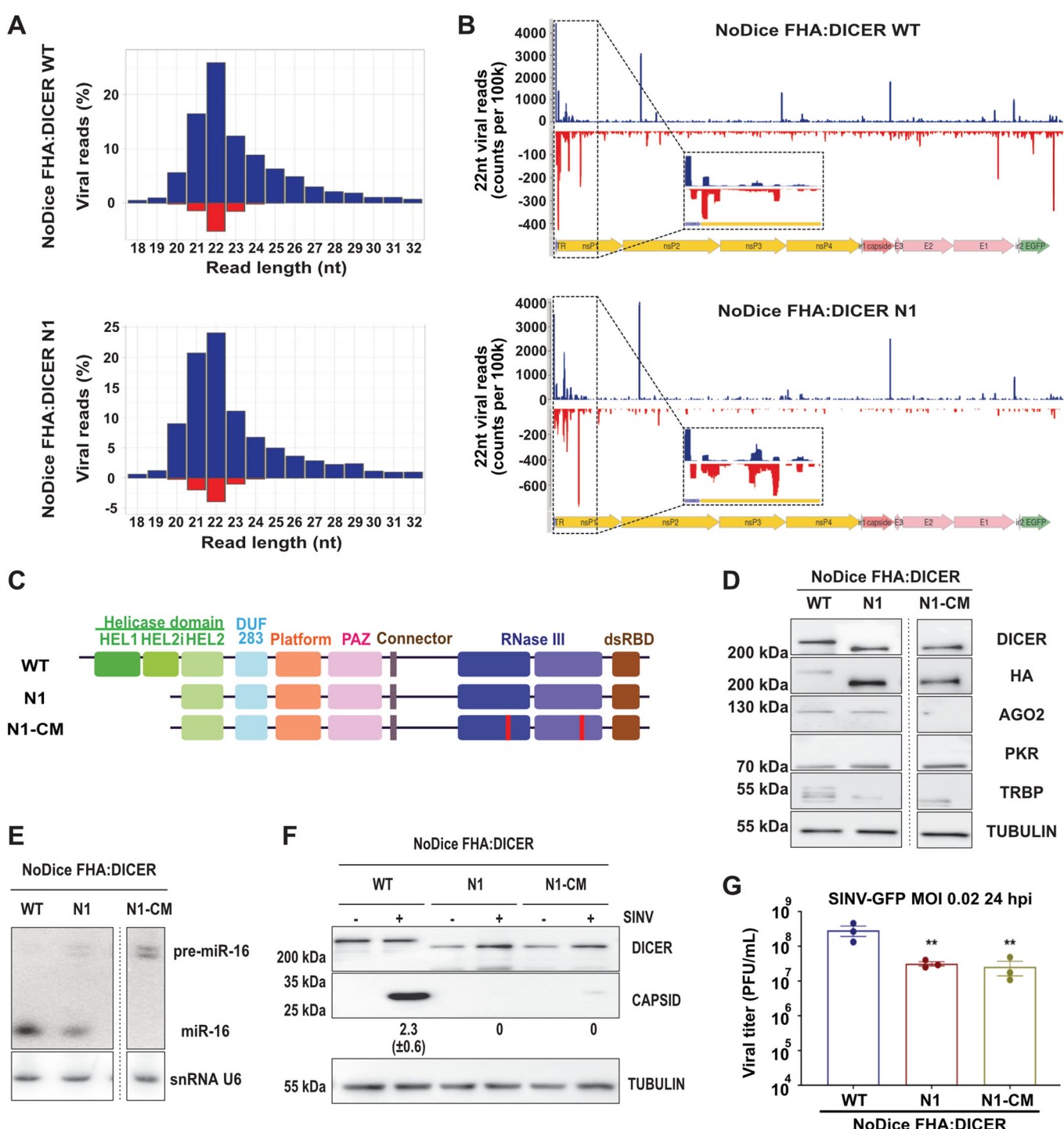

monoclonal cell lines, Dicer N1 and N1-CM cells displayed a drop in Capsid protein accumulation compared to Dicer WT cells (Fig. EV1D). Accordingly, the viral titer was strongly reduced in Dicer N1 and N1-CM cells compared to Dicer WT cells (Fig. EV1E). We showed previously that Dicer N1 could no longer interact with the Dicer co-factors TRBP and PACT and with the kinase PKR (Montavon et al, 2021). Thus, we verified by co-immunoprecipitation the interaction of Dicer N1-CM with these

proteins. The results confirmed that Dicer N1-CM behaved similarly to Dicer N1 regarding the loss of these interactions (Fig. EV1F).

Taken together, these results clearly show that the antiviral phenotype of Dicer N1 does not depend on its catalytic activity, and therefore on RNAi, which is in agreement with our previous observations that the antiviral effect of Dicer N1 did not depend on AGO2 (Montavon et al, 2021).

**Figure 3.   Dicer N1 catalytic activity is not involved in its antiviral phenotype.**

(A) Representative histograms of the distribution of viral reads (in percent) per RNA length upon AGO proteins immunoprecipitation and small RNA sequencing in NoDice FHA:DICER WT (top) and N1 (bottom) cells infected with SINV-GFP at an MOI of 0.02 for 24 h. Data from Replicate 1. Blue: positive strand; red: negative strand. (B) Representative graphs of the mapping of 22-nt-reads on SINV-GFP genome in NoDice FHA:DICER WT (top) and N1 (bottom) cells. On the bottom: magnification of the mapping for the first 1000 nt. Data from Replicate 1. Blue: positive strand; red: negative strand. (C) Schematic representation of two Dicer mutants, N1 and N1-CM (catalytic mutant). In N1-CM, the two catalytic mutations are highlighted in red. (D) Western blot analysis of DICER, HA, AGO2, PKR, and TRBP expression in the monoclonal cell lines NoDice FHA:DICER WT, N1, and N1-CM ($n = 3$ biological replicates). Alpha-Tubulin was used as a loading control. (E) Northern blot analysis of miR-16 expression in the same samples as in (D) ($n = 3$ biological replicates). Expression of snRNA U6 was used as a loading control. (F) Western blot analysis of DICER and CAPSID expression in SINV-GFP infected cells at an MOI of 0.02 for 24 h. Alpha-Tubulin was used as loading control. Band intensity was quantified and normalized to Tubulin for three independent biological replicates, then represented as mean ($+/-$ SD) under the corresponding lane. (G) Mean ($+/-$ SEM) of SINV-GFP viral titers in the same samples as in (D), infected at an MOI of 0.02 for 24 h ($n = 3$ biological replicates) from plaque assay quantification. Ordinary one-way ANOVA test with Dunnett's correction. **$p < 0.01$. Source data are available online for this figure.

## Impact of Dicer helicase subdomain deletions on viral infection

The region deleted in Dicer N1 contains both the Hel1 and Hel2i domains of the helicase, and was designed based on the presence of an alternative initiation codon at this position (Kennedy et al, 2015). Other deletion mutants have been reported in the literature, such as the rodent-specific Dicer[O], which is naturally expressed in oocytes (Flemr et al, 2013). Similar to Dicer N1, this truncated Dicer was shown to be more potent at generating siRNAs, and was further functionally and structurally studied under the name Dicer ΔHel1 (Zapletal et al, 2022). Finally, another Dicer isoform has been recently reported in human embryonic stem cells and was coined antiviral Dicer (AviD) for its RNAi-related antiviral property against different viruses (Poirier et al, 2021). This isoform lacks the Hel2i connector domain. To test the behavior of all these deletion mutants during SINV infection, we generated lentiviral constructs expressing a Flag-HA tagged version of Dicer ΔHel1, ΔHel2i and a version lacking the third helicase domain ΔHel2 (Fig. 4A). We used these lentiviral constructs, as well as catalytically inactive versions of them called CM, to generate monoclonal NoDice cell lines. We first verified the correct expression of the Dicer isoforms and checked their impact on AGO2, TRBP, and PKR expression (Fig. EV2A). We also checked that the catalytically inactive versions of the constructs were indeed impaired in miRNA biogenesis by measuring miR-16 expression by northern blot analysis (Fig. EV2B). We then infected cells expressing Dicer WT, ΔHel1, ΔHel2i, ΔHel2, or their catalytically inactive versions (CM) with SINV-GFP at an MOI of 0.02 for 24 h and monitored Capsid protein accumulation by western blot analysis. The levels of Capsid were lower in all cell lines expressing helicase-truncated versions of Dicer compared to Dicer WT cells, but seemed to go up when the RNAse III activity was mutated, although slightly less so in the case of ΔHel2 (Fig. 4B). To really assess the impact of the expression of these mutants on viral production, we titrated by plaque assay the supernatants of the previous experiment, using Dicer N1 cells as control. We observed that expression of all helicase deletion mutants, Dicer ΔHel1, ΔHel2i, and ΔHel2, resulted in a significant drop in viral titer, although slightly less than in Dicer N1 expressing cells, and that this effect was lost in cells expressing the CM versions of the mutants (Fig. 4C). Altogether, these results suggest that deleting any helicase subdomain confers an antiviral activity to human Dicer.

We previously showed that Dicer N1 antiviral phenotype was dependent on the expression of PKR (Montavon et al, 2021). We therefore tested the impact of the expression of the various helicase

deletion mutants on viral infection in NoDice cells compared to NoDiceΔPKR cells. We transduced the aforementioned cell lines with Dicer WT, N1, ΔHel1, ΔHel2i, and ΔHel2 to generate polyclonal cell lines. We again made sure that these cell lines expressed the different Dicer constructs as well as similar levels of interacting proteins such as AGO2 and TRBP (Fig. EV2C), and that they were also competent for miRNA production (Fig. EV2D). We then infected all those cell lines with SINV-GFP at an MOI of 0.02 for 24 h and measured the expression of the Capsid protein by western blot analysis. We could observe a lower accumulation in NoDice cells transduced with Dicer ΔHel1, ΔHel2, and ΔHel2i compared to cells transduced with Dicer WT (Fig. 4D), although the effect was less evident for cells expressing Dicer ΔHel2. On the contrary, in NoDiceΔPKR cells transduced with Dicer ΔHel1, ΔHel2, and ΔHel2i, we did not observe such a strong decrease in Capsid protein levels. We then determined the viral titers produced upon SINV-GFP infection of NoDice and NoDiceΔPKR cells expressing Dicer WT, N1, ΔHel1, ΔHel2i, or ΔHel2. While we confirmed the antiviral effect of the helicase deletion mutants compared to the WT Dicer in NoDice cells, we saw that this effect was completely abrogated in NoDiceΔPKR cells (Fig. 4E). We also performed an infection kinetic using SINV-GFP at an MOI of 2 in NoDice and NoDiceΔPKR cells expressing Dicer WT or the different helicase deletion mutants and monitored GFP fluorescence every hour for 24 h. The results, shown in Fig. 4F, confirmed that the percentage of GFP-positive cells increased slower in NoDice cells expressing the different helicase mutants than in the ones expressing Dicer WT, with a stronger effect in the presence of Dicer N1 or Dicer ΔHel2i. In NoDiceΔPKR cells, the difference in GFP-positive cells accumulation over time was less marked between conditions and the differences were not significant (Fig. 4F). We confirmed these observations by performing a plaque assay in the same conditions at 12 and 16 hpi. The viral titers were significantly lower at both time points in NoDice cells expressing Dicer helicase mutants than in the ones expressing Dicer WT, while this was not the case in NoDiceΔPKR cells, except for a modest effect in NoDiceΔPKR expressing Dicer ΔHel2i at 16 hpi (Fig. EV2E). We can therefore conclude that the antiviral phenotype of Dicer N1 can also be observed with smaller deletion mutants lacking individual sub-domains of the helicase. However, as opposed to Dicer N1, it seems that in the case of Dicer ΔHel1, ΔHel2i, or ΔHel2, this phenotype is dependent on the catalytic activity of Dicer, and thus is likely RNAi-dependent. Surprisingly, the antiviral phenotype of Dicer ΔHel1, ΔHel2i, or ΔHel2 is also lost in cells that do not express PKR, indicating that for these mutants as well, RNAi is not the sole explanation for their behavior. We thus decided to look more into the role of PKR.

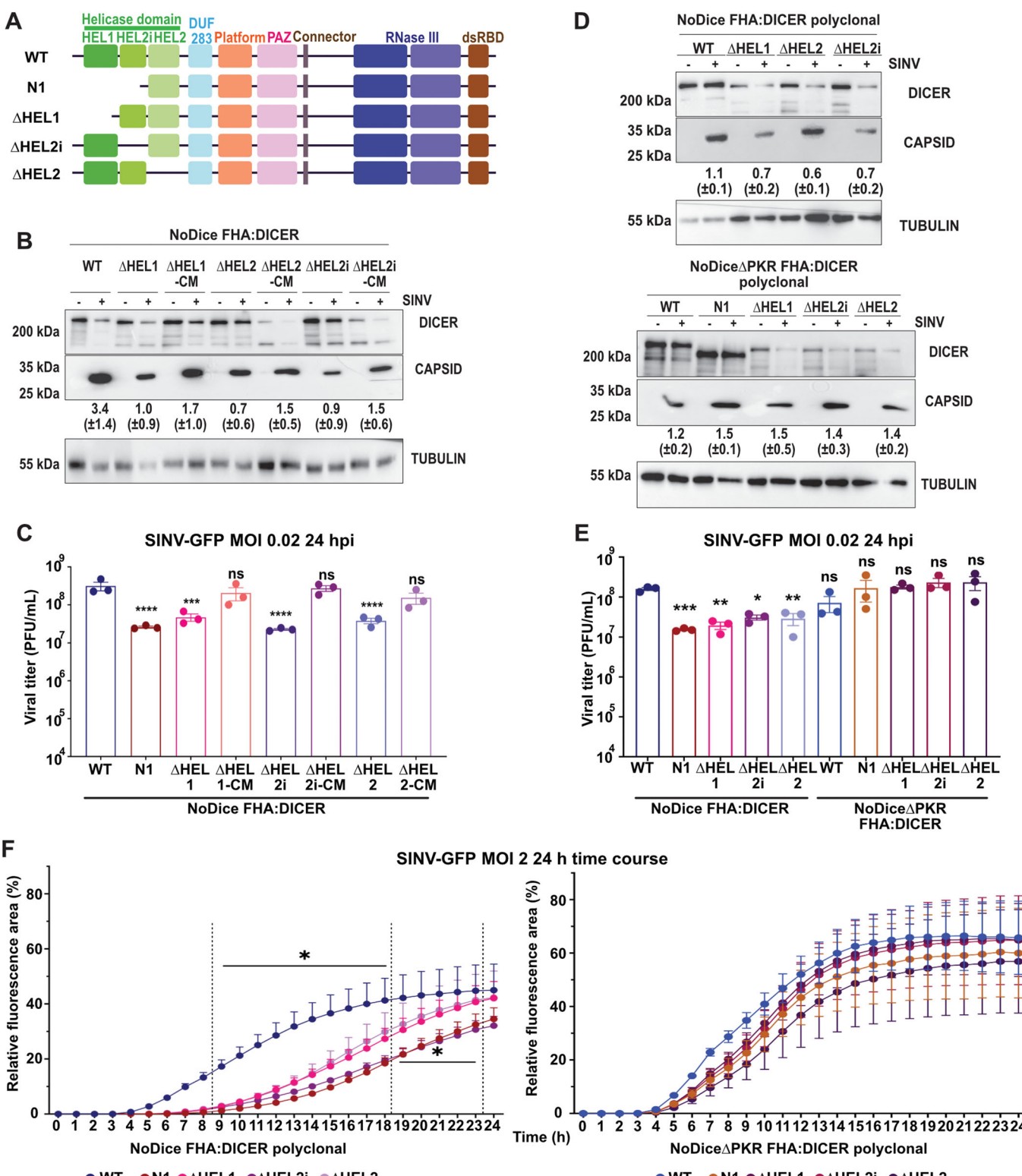

## PKR, but not its catalytic activity, is directly involved in the Dicer N1 phenotype

As shown above, the antiviral phenotype of Dicer helicase mutants seems to be strictly dependent on the expression of PKR. To

validate the implication of PKR, we decided to re-express it in NoDiceΔPKR cells to see whether this would complement the antiviral phenotype of Dicer N1. We generated constructs expressing the WT (Lemaire et al, 2008) or mutant versions of PKR as indicated in Fig. 5A. These mutants are: K296R, which is

**Figure 4.  Dicer helicase sub-domains as well as PKR are important for Dicer antiviral activity.**

(A) Schematic representation of the different Dicer helicase subdomain mutants. (B) Western blot analysis of DICER and CAPSID expression in SINV-GFP infected cells at an MOI of 0.02 for 24 h. Monoclonal NoDice FHA:DICER helicase mutant and their respective catalytic mutant cells are represented: WT, ΔHEL1, ΔHEL1-CM, ΔHEL2, ΔHEL2-CM, ΔHEL2i, and ΔHEL2i-CM. Alpha-Tubulin was used as loading control. Band intensity was quantified and normalized to Tubulin for three independent biological replicates, then represented as mean (+/− SD) under the corresponding lane. (C) Mean (+/− SEM) of SINV-GFP viral titers in the same samples as in (B), infected at an MOI of 0.02 for 24 h ($n = 3$ biological replicates) from plaque assay quantification. Ordinary one-way ANOVA test with Dunnett's correction. ***$p < 0.001$; ****$p < 0.0001$; ns: non-significant. (D) Western blot analysis of DICER and CAPSID expression in SINV-GFP infected cells at an MOI of 0.02 for 24 h. Polyclonal NoDice FHA:DICER (top) and NoDiceΔPKR FHA:DICER (bottom) helicase mutant cells are represented. Alpha-Tubulin was used as loading control. Band intensity was quantified and normalized to Tubulin for three independent biological replicates, then represented as mean (+/− SD) under the corresponding lane. (E) Mean (+/− SEM) of SINV-GFP viral titers in the same samples as in (D), infected at an MOI of 0.02 for 24 h ($n = 3$ biological replicates) from plaque assay quantification. Ordinary one-way ANOVA test with Dunnett's correction. *$p < 0.05$; **$p < 0.01$; ***$p < 0.001$; ns: non-significant. (F) Mean (+/− SEM) of SINV-GFP fluorescence detection per cell area in the same samples as in (D) and (E), infected at an MOI of 2 between 0 and 24 h ($n = 3$ biological replicates). In NoDice cells, all differences were significant (*$p < 0.05$) between 9 and 18 hpi, but were only significant for Dicer N1 and ΔHEL2i-expressing cells between 19 and 23 hpi. At earlier time points, significance was not reached. In NoDiceΔPKR cells, significance was never reached. Ordinary two-way ANOVA test with Dunnett's correction. Source data are available online for this figure.

not able to bind ATP, homodimerize, autophosphorylate, or phosphorylate its targets (Dey et al, 2005), and T451A, which is still able to form homodimers but has lost its ability to autophosphorylate the T451 residue and to phosphorylate its targets (Taylor et al, 2001). We first transduced constructs expressing MYC:CTRL, MYC:PKR WT, MYC:PKR K296R, or MYC:PKR T451A in NoDiceΔPKR FHA:Dicer WT cells and infected them with SINV-GFP at an MOI of 0.02 for 24 h. We analyzed the expression of PKR and its phosphorylated form by western blot analysis, which confirmed that all constructs were expressed at high levels, but only WT PKR could be phosphorylated upon SINV infection (Fig. 5B). Interestingly, there was some basal level of phosphorylated PKR in the mock-infected cells expressing WT PKR, which could be due to a higher expression of PKR in transduced cells compared to WT cells. In terms of viral capsid accumulation, no real difference could be detected between the different conditions (Fig. 5B). We measured viral particle production by plaque assay in SINV-GFP infected cells, and could not detect any difference between samples (Fig. 5C). This result indicates that in the presence of Dicer WT, expression of a WT or inactive mutant PKR has no significant impact on SINV infection. We then transduced the same MYC:CTRL or PKR constructs as above in NoDiceΔPKR FHA:Dicer N1 cells and performed the same analysis after infection with SINV-GFP. The western blot analysis was consistent with the previous one and confirmed the correct expression of the PKR constructs as well as the activation of only the WT PKR version (Fig. 5D). However, we noted a small reduction of Capsid protein accumulation in cells expressing MYC:PKR WT, K296R, or T451A compared to the MYC:CTRL condition (Fig. 5D). This was confirmed by the plaque assay analysis that revealed a significant drop in viral titer in NoDiceΔPKR Dicer N1 cells expressing WT or inactive PKR compared to cells expressing the negative control (Fig. 5E). We could thus restore the antiviral phenotype of Dicer N1 by re-expressing PKR in cells where its expression was ablated. Interestingly, this did not require a catalytically active or homodimerization competent PKR since the same complementation could be observed with two different mutants that were both unable to phosphorylate their substrates. We also tested whether the expression of WT or inactive PKR had an impact on SINV infection in the absence of Dicer. We therefore transduced NoDiceΔPKR FHA:CTRL cells with constructs expressing MYC:CTRL, MYC:PKR WT, K296R, or T451A and infected them for 24 h with SINV-GFP at an MOI of 0.02. Although we could

measure a small effect on Capsid protein expression (Fig. EV3A), there was no significant difference in viral titers (Fig. EV3B) between conditions, which indicates that the presence of Dicer N1 is required to observe a PKR-dependent antiviral effect. Finally, because we observed that Dicer N1 expressing cells appeared to be more sensitive to SARS-CoV-2 infection, we also tested the effect of removing PKR in this setting. We therefore transduced hACE2 in NoDiceΔPKR cells and then transduced them with FHA:Dicer WT or FHA:Dicer N1 constructs before infecting them with SARS-CoV-2 for 48 h at an MOI of 0.001. We verified that these cells expressed ACE2 and the WT or N1 isoforms of Dicer, and monitored viral Nucleocapsid accumulation by western blot analysis (Fig. 5F) and viral titers by TCID50 (Fig. 5G). While NoDice cells expressing Dicer N1 showed the previously observed increase in viral infection levels compared to Dicer WT cells, we could not observe the same difference in NoDiceΔPKR cells. The infection levels were globally higher in these cells compared to NoDice cells, but we did not observe a further increase in Dicer N1 vs. Dicer WT cells. Overall, these results suggest that the antiviral activity towards SINV or proviral activity towards SARS-CoV-2 of Dicer N1 is linked to PKR, and in the case of the antiviral effect, does not require the canonical activation of PKR through homodimerization and phosphorylation.

## Expression of Dicer N1 induces major changes in the transcriptome

To decipher the mechanism behind Dicer N1 antiviral property, we performed total RNAseq analysis on both Dicer WT and Dicer N1 cells either mock-infected, infected with SINV-GFP at an MOI of 2 for 12 h or infected with SINV-GFP at an MOI of 0.02 for 24 h. We first verified the percentage of viral reads in the libraries and observed that it was higher (about 8-fold for the genomic RNA) in Dicer WT than in Dicer N1 cells (Fig. EV4A). This confirmed the antiviral effect of Dicer N1 at the level of viral replication. Next, we compared the overall profiles of gene expression in the different conditions (Fig. 6A). The analysis identified 5 groups of genes according to their expression patterns. The first group was composed of genes overexpressed in N1 cells compared to WT cells. The second one was composed of genes specifically downregulated in WT cells upon infection, whereas the third one represented genes downregulated upon infection in both cell lines. Finally, the fourth and fifth groups comprised genes that were upregulated in both WT and N1 cells or in WT cells only, respectively. The first two groups were of interest as several genes were

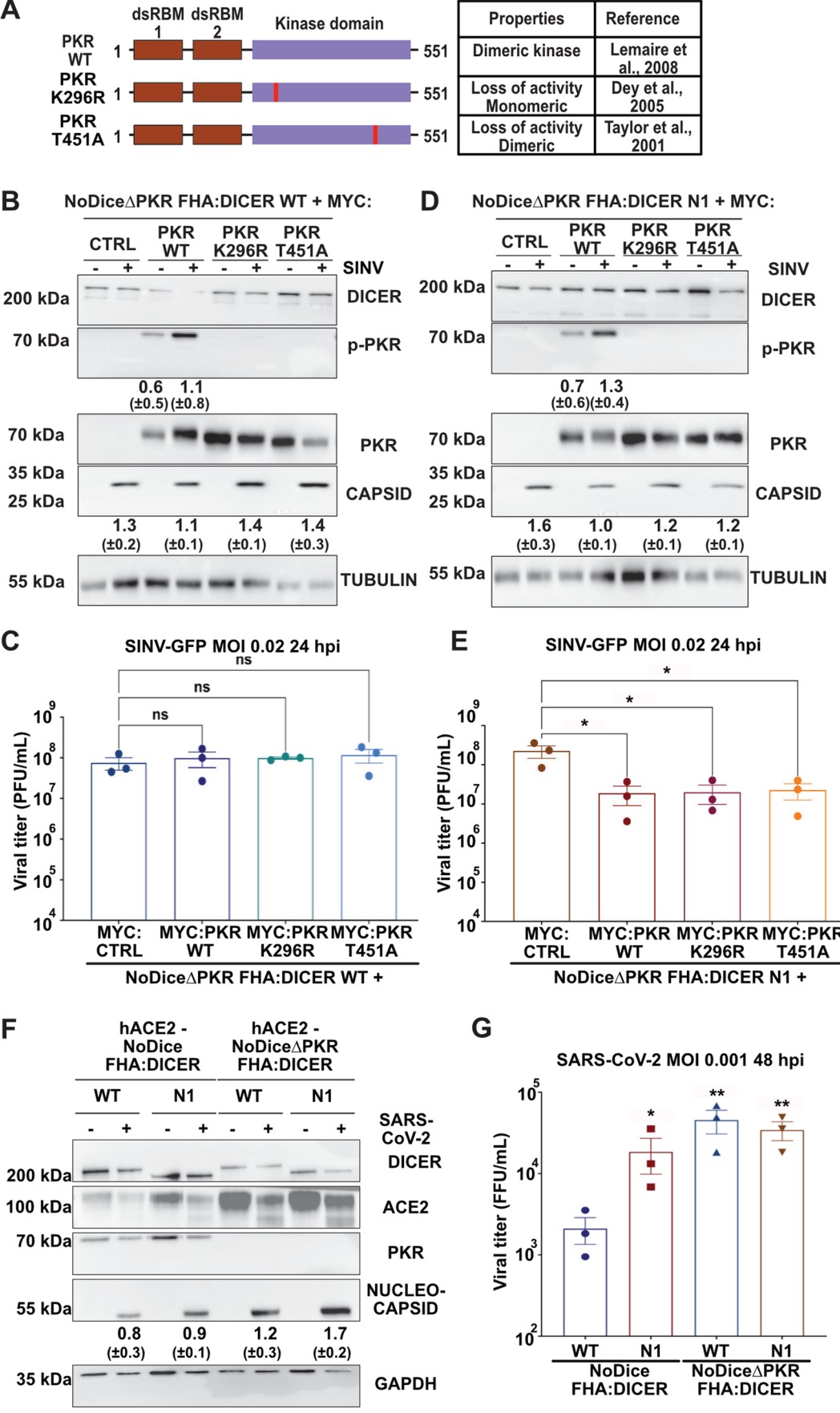

**Figure 5. PKR requirements for Dicer N1 anti- or pro-viral activity.**

(A) Schematic representation of PKR and its two point mutants: K296R and T451A. (B) Western blot analysis of DICER, p-PKR, PKR, and CAPSID expression in SINV-GFP infected NoDiceΔPKR FHA:DICER WT cells expressing MYC:EMPTY CTRL vector, MYC:PKR, MYC K296R or MYC:T451A at an MOI of 0.02 for 24 h. Alpha-Tubulin was used as loading control. Band intensity was quantified and normalized to Tubulin for three independent biological replicates, then represented as mean (+/− SD) under the corresponding lane; p-PKR = p-PKR/PKR quantification. (C) Mean (+/− SEM) of SINV-GFP viral titers in the same samples as in (B), infected at an MOI of 0.02 for 24 h ($n = 3$ biological replicates) from plaque assay quantification. Ordinary one-way ANOVA test with Dunnett's correction. ns: non-significant. (D) Western blot analysis of DICER, p-PKR, PKR and CAPSID expression in SINV-GFP infected NoDiceΔPKR FHA:DICER N1 cells expressing MYC:EMPTY CTRL vector, MYC:PKR, MYC K296R or MYC:T451A at an MOI of 0.02 for 24 h. Alpha-Tubulin was used as loading control. Band intensity was quantified and normalized to Tubulin for three independent biological replicates, then represented as mean (+/− SD) under the corresponding lane; p-PKR = p-PKR/PKR quantification. (E) Mean (+/− SEM) of SINV-GFP viral titers in the same samples as in (D), infected at an MOI of 0.02 for 24 h ($n = 3$ biological replicates) from plaque assay quantification. Ordinary one-way ANOVA test with Dunnett's correction. *$p < 0.05$. (F) Western blot analysis of DICER, human ACE2, PKR and NUCLEOCAPSID expression in SARS-CoV-2 infected NoDice FHA:DICER WT or N1 and NoDiceΔPKR FHA:DICER WT or N1 cells at an MOI of 0.001 for 48 h. GAPDH was used as loading control. Band intensity was quantified and normalized to GAPDH for three independent biological replicates, then represented as mean (+/− SD) under the corresponding lane. (G) Mean (+/− SEM) of SARS-CoV-2 viral titers in the same samples as in (F), infected at an MOI of 0.001 for 48 h ($n = 3$ biological replicates) from TCID50 quantification. Unpaired t-test with Welch's correction. *$p < 0.05$; **$p < 0.01$. Source data are available online for this figure.

upregulated in Dicer N1 cells no matter whether they were infected or not. We then performed a differential gene expression analysis of mRNAs using DESeq2 (Love et al, 2014) to determine the impact of the infection on the cell transcriptome. The results confirmed that in Dicer WT cells, infection by SINV resulted in major changes in the expression levels of a large number of mRNAs, which were visible both at 12 and 24 hpi (Figs. 6B and EV4B). However, in Dicer N1 cells, these changes were substantially more limited at both time points of infection (Figs. 6C and EV4B), consistent with the fact that the infection was attenuated in these cells and therefore did not result in a strong cellular response. Strikingly, when we compared the transcriptomes of mock-infected Dicer WT and Dicer N1 cells, we found that there was a large number of upregulated mRNAs, as well as a smaller number of downregulated mRNAs (Fig. 6D). It seems therefore that the expression of Dicer N1 results in changes at the transcriptional level, so we set out to examine whether any of these perturbations could explain the phenotype we observed in these cells. To determine whether specific clusters of genes were specifically deregulated between Dicer WT and N1 cells, we looked at gene set enrichment analysis (GSEA) against the human Hallmark gene sets, which represent specific well-defined biological states or processes with coherent expression (Figs. 6E and EV4C, D). In Dicer N1 cells, gene sets linked to interferon alpha, interferon gamma, inflammatory and NF-kB responses were significantly enriched compared to Dicer WT cells (Fig. 6E). When looking at the individual enrichment plots for hallmarks such as IFN alpha, gamma or inflammatory response and TNF alpha signaling via NF-kB, we clearly noticed a strong enrichment score in Dicer N1 vs. Dicer WT mock cells (Fig. EV4C). These results uncovered a basal innate immunity signature linked to a subset of genes in Dicer N1 cells. In contrast, gene sets belonging to the same hallmarks (IFN alpha, IFN gamma, inflammatory response, and TNF alpha signaling via NF-kB) were depleted in Dicer WT infected vs. mock cells (Fig. EV4D).

Altogether, these results highlight transcriptomic differences in Dicer N1 compared to WT cells in mock and SINV-infected cells. Globally, immune-related processes seem to be enriched in Dicer N1 cells compared to WT cells, and this, even in the steady state.

## Dicer N1 expression increases the activation of immune-related transcription factors

Given the seemingly upregulation of innate immune genes in the Dicer N1 cells, we looked for transcription factors that are known to play important roles in innate immunity and analyzed the expression levels of their target genes in the different samples. We retrieved lists of the known targets of NF-kB/p65, STAT1, STAT2, IRF2, and IRF3 and extracted from these lists genes that were differentially expressed either during infection in Dicer WT or Dicer N1 cells, or between mock-infected Dicer WT and Dicer N1 cells. We then computed the cumulative frequencies of deregulated transcripts by comparing (i) Dicer WT SINV-GFP 24 hpi vs. Dicer WT mock, (ii) Dicer N1 SINV-GFP 24 hpi vs. Dicer N1 mock and (iii) Dicer N1 mock vs. Dicer WT mock. This analysis was done for NF-kB/p65 (Fig. 7A) and STAT2 (Fig. 7B) targets, which were the ones with the greater changes, but also for IRF2 (Fig. EV5A), IRF3 (Fig. EV5B), and STAT1 (Fig. EV5C) targets. The results indicate first that there are more NF-kB target genes that are regulated upon SINV infection in Dicer WT compared to Dicer N1 cells (see the black vs. red curves in Fig. 7A). This was less evident for STAT2 targets, probably due to the lower number of genes involved (Fig. 7B). Surprisingly, there were significantly more upregulated NF-kB and STAT2 targets when comparing Dicer N1 mock to Dicer WT mock than when comparing Dicer WT SINV-infected to Dicer WT mock conditions (see the blue vs. black curves in Fig. 7A,B). A global representation in the form of heatmaps for the same sets of genes that are known targets of NF-kB/p65 or STAT2 also allowed to visualize the stronger changes induced by the infection in Dicer WT cells compared to Dicer N1 cells, and that a large number of these targets were more expressed in Dicer N1 mock than in Dicer WT mock condition (Fig. 7C,D). We also represented DESeq2 results obtained for NF-kB/p65 and STAT2 targets in Dicer WT infected vs mock and Dicer N1 mock vs Dicer WT mock comparisons on volcano plots (Fig. EV5D,E). Some of the most upregulated NF-kB targets are also listed in Fig. EV5F. These plots indicate that there are specific NF-kB and STAT2 target genes upregulated in Dicer N1 mock compared to Dicer WT cells that are not differentially expressed in Dicer WT SINV-infected compared to mock cells.

We validated the increased expression of some selected target genes by RT-qPCR and could confirm that PTGS2, APOBEC3B, MX1, OAS3, and IFIT3 mRNAs were indeed significantly upregulated in mock-infected Dicer N1 compared to Dicer WT cells (Fig. 7E). Conversely, very mild or no expression increase and sometimes a decrease in expression was detected for the selected target genes in Dicer N1 infected compared to mock cells, consistent with the RNAseq analysis (Fig. EV5G). Besides, the

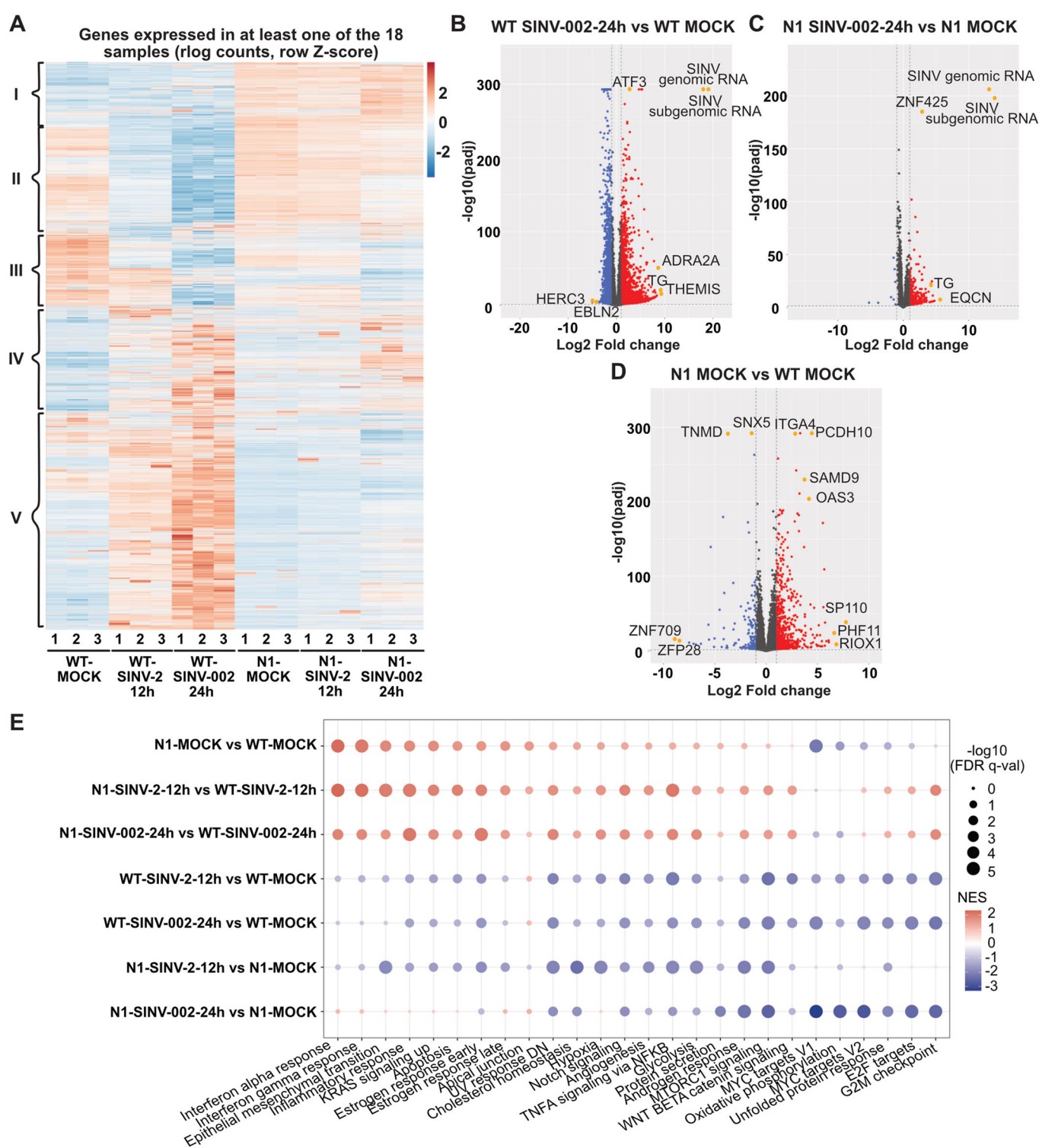

decrease in mRNA levels observed in WT cells upon infection was expected as SINV blocks RNA Pol II transcription via its nsP2 protein (Akhrymuk et al, 2018). We also verified whether the observed upregulation of NF-kB targets was accompanied by an increase in basal levels of IFN-I levels in Dicer N1 cells. We measured expression of *IFNß* mRNA by RT-qPCR in Dicer WT and Dicer N1 cells, but although there was a tendency for an

increased expression in Dicer N1 cells, it was not significant (Fig. 7F). This was not too surprising given the fact that HEK293T cells are known for expressing low levels of IFNß even when stimulated (Ferreira et al, 2020).

Overall, the transcriptome changes induced by Dicer N1 expression, which we validated for selected transcripts by RT-qPCR, allow us to postulate that these cells are in a pre-activated

**Figure 6.  Dicer N1 and WT cells exhibit transcriptomic differences linked to an immune-related response.**

(A) Z-score hierarchical clustering heatmap of genes identified by RNA sequencing analysis as differentially expressed between WT vs. N1 cells, either mock or SINV-infected cells (MOI 2 12 hpi or MOI 0.02 24 hpi). Gene clusters are delimited with black brackets and numbered from I to V. (B–D) Volcano plots for differentially expressed genes (DEGs) between SINV-infected (MOI of 0.02 for 24 h) and mock NoDice FHA:DICER WT cells (B); SINV-infected and mock NoDice FHA:DICER N1 cells (C); mock NoDice FHA:DICER WT and mock NoDice FHA:DICER N1 cells (D). Each gene is marked as a dot (red: upregulated ≥ 2-fold, blue: downregulated ≤ 2-fold, gray: unchanged). The horizontal line denotes an adjusted $p$-value of 0.05 (Wald test, DESeq2 package) and the vertical ones the Log2 fold change cut-off (−1 and 1) ($n = 3$ biological replicates). (E) Hallmark gene set enrichment analysis. Only results of 26 out of the 50 human Hallmark gene sets are depicted here. Colors indicate NES (normalized enrichment score) that are either positive (red) or negative (blue). Dot size corresponds to −log10(FDR q-val) of enrichment. Source data are available online for this figure.

state that might explain their increased resistance to SINV infection. This state appears to be more a pre-inflammatory state rather than an overall pre-activation of IFN-I response. We therefore decided to follow up on the implication of the NF-kB pathway.

## The NF-kB pathway is implicated in the antiviral phenotype of Dicer N1

We showed above that the presence of PKR but not its canonical kinase activity was needed for the Dicer N1 antiviral phenotype. It has been reported that PKR can be involved in the activation of the immune response in a non-canonical manner. In particular, PKR has been described as an inducer of NF-kB/p65 transcriptional activity independently of its kinase function (Bonnet et al, 2000). Since we also found that targets of NF-kB are induced specifically in Dicer N1 cells, we investigated whether this pathway was involved in the antiviral phenotype of these cells. NF-kB/p65 is known to be phosphorylated once IkBα is released from the complex, which results in NF-kB/p65 activation upon translocation into the nucleus (Christian et al, 2016; Kanarek and Ben-Neriah, 2012). We looked at the expression and phosphorylation levels of NF-kB/p65 in Dicer WT and N1 cells during a time course of SINV-GFP infection at an MOI of 2 from 3 to 24 hpi (Fig. 8A). In Dicer WT cells, we observed an increase in p65 phosphorylation at 6 hpi, which corresponds to the end of the first replication cycle. However, at 12 hpi the phosphorylation decreased and became undetectable at 24 hpi. In Dicer N1 cells, the level of phosphorylated p65 was already quite elevated in the mock-infected condition and remained high until 12 hpi where it slightly decreased until 24 hpi (Fig. 8A). At the same time, we noticed a higher level of NF-kB/p65 protein at both 12 and 24 hpi. To further link the phosphorylation of p65 to a deregulation of PKR in Dicer N1 cells, we infected NoDiceΔPKR cells expressing FHA:Dicer N1 and transduced with either a MYC:CTRL or a MYC:PKR WT construct and measured the levels of p65, its phosphorylated form and capsid protein by western blot analysis. We confirmed the effect of the lack of PKR on SINV capsid accumulation and observed an increase in phospho-p65 in both mock- and SINV-infected cells only when PKR expression was restored (Fig. 8B). The increase in NF-kB activity in Dicer N1 cells expressing PKR could also be validated by measuring expression of its target PTGS2 by RT-qPCR (Fig. 8C).

Given the higher activation levels of p65 in Dicer N1 cells, we then decided to block the NF-kB pathway to test its role during SINV infection. To do so, we used the chemical compound BAY 11-7082, which inhibits IkBα phosphorylation by the IKK kinases and thus prevents NF-kB/p65 activation (Pierce et al, 1997). We treated Dicer N1 cells with two different concentrations of BAY 11-

7082 during 1 h. Then, we infected cells with SINV-GFP at an MOI of 0.02 for 24 h and evaluated the impact of the drug treatment on the infection efficiency. As shown in Fig. 8D, we observed a dose-dependent decrease in the phosphorylation of p65 in mock-infected cells treated with BAY 11-7082 compound compared to cells treated with DMSO only. Conversely, the level of IkBα was increased in cells treated with the inhibitor compared to the control. These results indicate that the treatment was effective. We further validated that the inhibitor worked as expected by analyzing the mRNA levels of the previously validated NF-kB target PTGS2 by RT-qPCR. In Dicer N1 cells treated with BAY 11-7082, the level of this mRNA was reduced in a dose-dependent manner compared to the control condition (Fig. 8E). The effect of the NF-kB inhibitor on viral infection could be visualized first by the increased accumulation of the capsid protein in SINV-GFP infected cells treated with 5 μM of BAY 11-7082 compared to DMSO-treated cells (Fig. 8D). Finally, we measured the effect of NF-kB inhibition on viral particle production, and observed a significant increase in viral titers in Dicer N1 cells treated with 5 μM of BAY 11-7082 compared to the negative control (Fig. 8F). The BAY 11-7082 treatment had no effect on SINV-GFP infection in Dicer WT cells, as assessed by western blot analysis of the capsid protein and by plaque assay (Appendix Fig. S4A,B). Overall, these results indicate that by blocking the NF-kB/p65 pathway with BAY 11-7082, the antiviral activity of Dicer N1 can be reverted, suggesting that it is partially mediated through the NF-kB/p65 pathway.

## Discussion

In this study, we examined the role played by Dicer during viral infection in human cells. We specifically focused on the importance of its helicase domain, as it is known to play important roles in modulating its activity. Indeed, some of the deletion mutants that we used in our analysis, such as Dicer N1, Dicer ΔHel1 or Dicer ΔHel2i have been reported to display increased RNAi activity directed against either an artificial dsRNA substrate or specific viruses (Kennedy et al, 2015; Poirier et al, 2021; Zapletal et al, 2022). Our results confirmed that helicase-truncated Dicer proteins displayed an antiviral phenotype against some, but not all, viruses. While we observed that the catalytic activity of Dicer was involved for some deletion mutants, we also showed that the presence of the PKR protein was an essential feature for the antiviral phenotype of all deletion mutants tested. In particular, the Dicer N1 variant, which was shown to be capable of RNAi activity in cells that did not express PKR (Kennedy et al, 2015) was antiviral in our hands only in the presence of PKR, and remained antiviral when rendered catalytically inactive. A link between Dicer and PKR was also

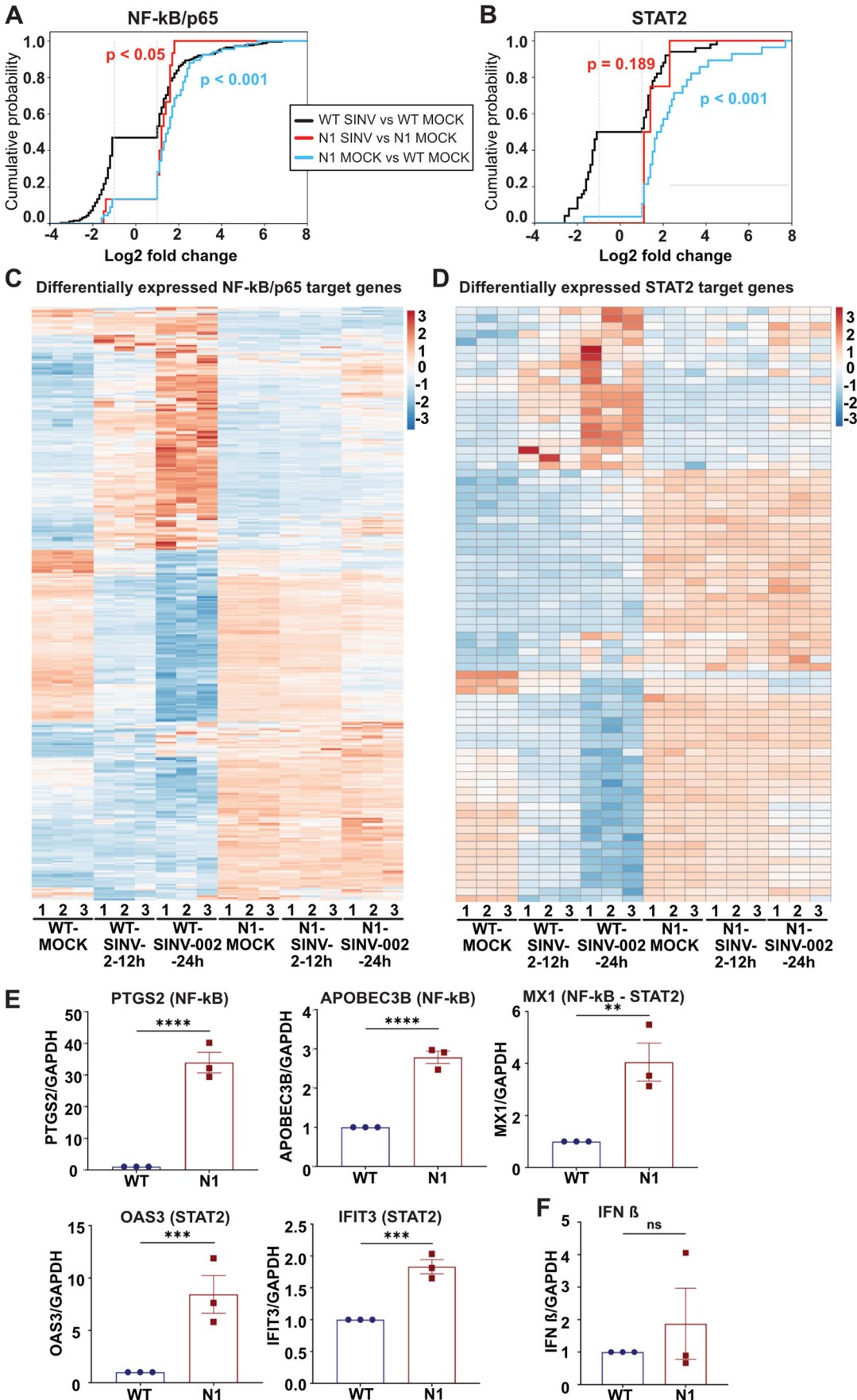

**Figure 7. NF-kB/p65 and STAT2 transcription factors are involved in Dicer N1 cells transcriptomic changes.**

(A,B) Histograms representing the cumulative probability of differentially expressed genes controlled by the transcription factors NF-kB/p65 (A) or STAT2 (B), plotted according to their Log2 fold change. The vertical lines stand for the Log2 fold change cut-offs (−1 and 1). The two-sample Kolmogorov–Smirnov test was used to assess whether each distribution was statistically different from the distribution of NoDice FHA:DICER WT cells infected with SINV vs. mock. *p*-values are indicated on each histogram. Black: WT SINV-002-24h vs WT MOCK; red: N1 SINV-002-24h vs N1 MOCK; blue: N1 MOCK vs WT MOCK. (C,D) Z-score hierarchical clustering heatmap of genes identified by RNA sequencing analysis as differentially expressed between WT vs. N1 cells, either mock or SINV-infected cells (MOI 2 12 hpi or MOI 0.02 24 hpi). Each heatmap represents differentially expressed genes controlled by one transcription factor: NF-kB/p65 (C) or STAT2 (D). (E) RT-qPCR on upregulated genes in the N1 MOCK vs WT MOCK condition and controlled by either NF-kB/p65 or STAT2. Mean (+/− SEM); *n* = 3 biological replicates. Unpaired t-test. **$p < 0.01$; ***$p < 0.001$; ****$p < 0.0001$. (F) RT-qPCR on IFN ß gene in WT and N1 cells in mock condition. Mean (+/− SEM); *n* = 3 biological replicates. Unpaired t-test. ns: non-significant. Source data are available online for this figure.

proposed in mouse embryonic stem cells where Dicer appears to prevent the PKR-induced IFN-I response (Gurung et al, 2021). However, no mechanism was proposed, and the authors hypothesized that it might be linked to miRNA production. We do not believe that miRNAs are involved in the phenotype of Dicer N1 since there were no dramatic changes in miRNA profiles induced by its expression. Our data seem to indicate that the PKR-mediated antiviral effect observed in cells expressing helicase-truncated Dicer proteins is dominant over the RNAi-mediated effect. One would expect that RNAi would be more potent when PKR is not expressed since it should compete with Dicer for dsRNA binding. The fact that it is not the case during the antiviral response raises interesting perspectives. Surprisingly, we also observed that expression of Dicer N1 had a proviral effect in the case of SARS-CoV-2, and that this phenotype was also dependent on PKR expression. This observation definitely rules out the involvement of RNAi since the virus levels are increased in this case.

PKR is a key effector protein involved in pathways such as apoptosis, autophagy, cell cycle control, and immunity (Williams, 1999). Besides its well-known role in translation control via phosphorylation of the eIF2α factor (Donnelly et al, 2013), it also plays key roles in modulating IFN-I signaling pathways and can negatively regulate STAT1 and STAT3 transcriptional activity in a kinase-dependent manner (Raven et al, 2006). It has also been involved in the NF-kB pathway, but in this case it can do so both in a catalytic-dependent (Gil et al, 2001) and independent way, via the IKK kinase (Bonnet et al, 2000; Ishii et al, 2001). Here, we showed that Dicer N1 expressing cells had a different transcriptome from Dicer WT expressing cells and showed a significant enrichment of gene sets related to IFN alpha/gamma and inflammatory/NF-kB responses. Specifically, we could identify among the differentially expressed genes a substantial fraction of targets of key immune transcription factors such as STAT2 and NF-kB. We retrieved and validated the differential expression levels for known antiviral effectors such as PTGS2, APOBEC3B, or OAS3 (Jiang et al, 2008; Lehman et al, 2022; Manjunath et al, 2023; Ryman et al, 2002). This observation is reminiscent of other studies that showed that Dicer could be involved in non-RNAi-related signaling pathways. For instance, *Drosophila* Dicer-2 can induce the expression of a small antiviral peptide, Vago, in a tissue- and virus-specific manner (Deddouche et al, 2008). More recently, plant DCL2 was involved in auto-immunity activation upon *DCL4* loss and in the activation of defense gene expression such as nucleotide-binding domain/leucine-rich repeat immune receptors (Nielsen et al, 2023). By measuring expression levels of *IFNß* mRNA in Dicer N1 expressing cells, we could hypothesize that most of the observed effect results from a deregulation of the NF-kB pathway without a requirement

for cytokine production. We confirmed this hypothesis by treating Dicer N1 expressing cells with an NF-kB inhibitor, which resulted in the abrogation of their antiviral phenotype. We can therefore assume that the transcriptional deregulation of key genes involved in the innate immune response and under the control of NF-kB was responsible for the observed effect on virus infection. Furthermore, the fact that Dicer N1 cells presented increased levels of SARS-CoV-2 infection is also in favor of an involvement of NF-kB signaling. Indeed, SARS-CoV-2 is known to benefit from the upregulation of the inflammatory response, mainly through the NF-kB pathway, which is essential for the replication and propagation of the virus (Nilsson-Payant et al, 2021). A very recent study confirmed the proviral role of the NF-kB pathway in SARS-CoV-2 infected lung cells (Bhargava et al, 2023).

Our study reinforces the non-canonical roles of Dicer in immune signaling pathways and its interplay with other antiviral responses. Dicer helicase domain sequesters PKR away from its conventional signaling function. We showed that Dicer N1 antiviral activity only depends on the presence of PKR but not on its dimerization or catalytic activity. Even though we cannot at this stage formally conclude on the mechanism at play, we can speculate that PKR co-factors TRBP and/or PACT might be involved. Indeed, PKR can be activated by PACT upon stress (Farabaugh et al, 2020; Ito et al, 1999). Conversely, TRBP can inhibit PKR directly or sequester PACT away from PKR (Chukwurah and Patel, 2018; Daher et al, 2009). Since both proteins also interact with Dicer via its helicase domain and thus cannot interact with it anymore in the Dicer N1 cells, they are expected to preferentially interact with other partners such as PKR in the Dicer N1 background and they could limit or change its canonical function. This could in theory explain the activation of the NF-kB pathway in a PKR-dependent manner in the presence of a helicase-truncated Dicer. Interestingly, it was previously shown that the NF-kB pathway could promote SINV infection, but only in mature non-dividing neurons (Yeh et al, 2019). In the proliferating cell line that we used, we rather showed that the basal activation level of NF-kB in Dicer N1 cells is antiviral. In agreement with this observation, inhibiting NF-kB in Dicer WT cells has no effect on SINV, which is similar to what has been reported in the Yeh et al study. Our results therefore indicate that in actively dividing cells, the NF-kB status dictates the fate of SINV infection, and this can be in part controlled by the Dicer isoform expressed. Of course, it remains to be seen whether our observations, which were done in one specific cell line, can be generalized to other cellular systems. In particular, this point is crucial for cells that naturally express isoforms of helicase-truncated Dicer, i.e. oocytes and stem cells: it would be important to check whether signaling pathways are deregulated and whether

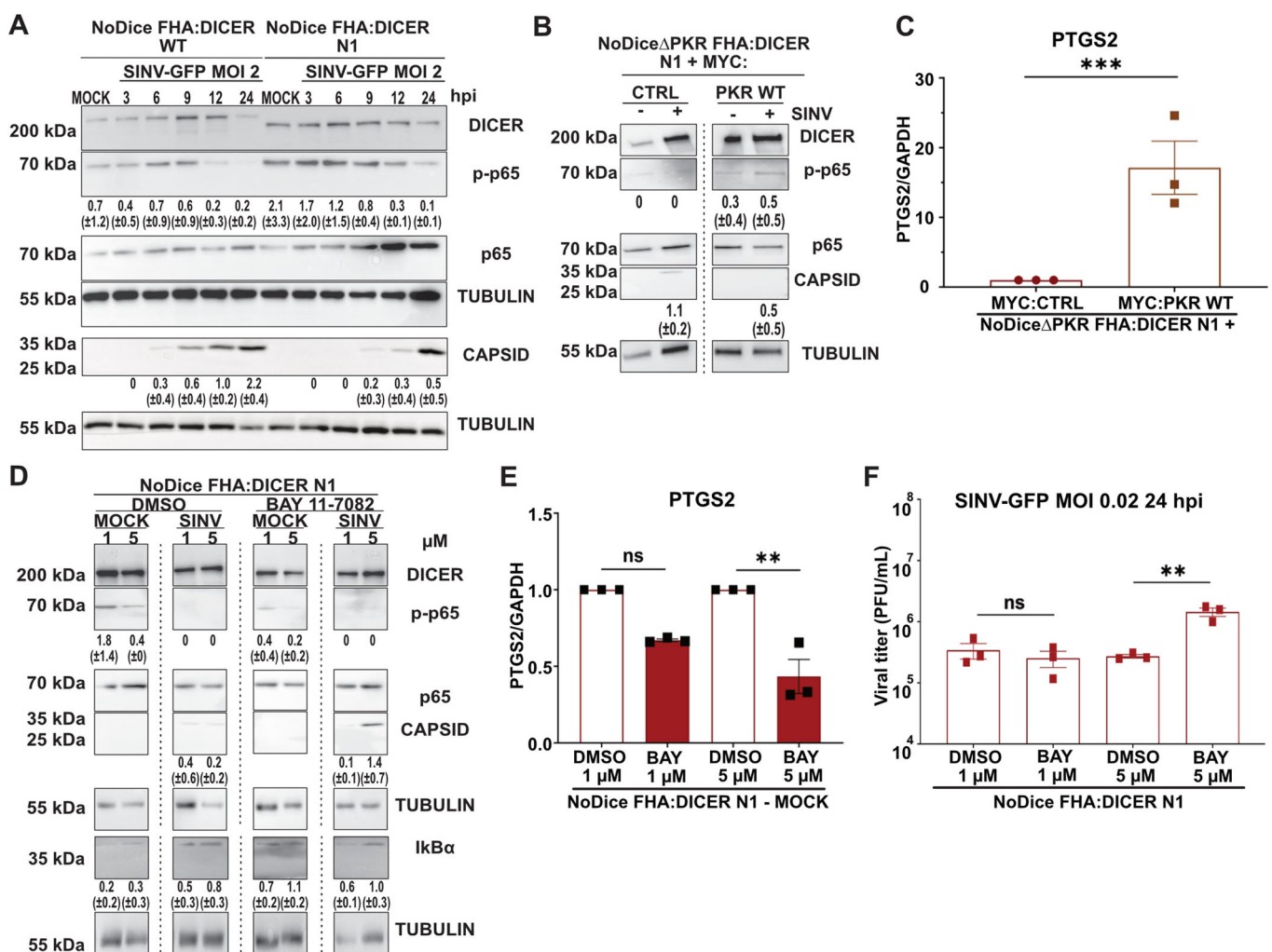

**Figure 8. NF-kB/p65 is activated by PKR in Dicer N1 cells and is involved in their antiviral phenotype.**

(A) Western blot analysis of DICER, p-p65, p65, and CAPSID expression in SINV-GFP infected NoDice FHA:DICER WT and N1 cells at an MOI of 0.02 between 3 and 24 h. Alpha-Tubulin was used as loading control. Band intensity was quantified and normalized to Tubulin for three independent biological replicates, then represented as mean (+/− SD) under the corresponding lane; p-p65 = p-p65/p65 quantification. (B) Western blot analysis of DICER, p-p65, p65, and CAPSID expression in SINV-GFP infected NoDiceΔPKR FHA:DICER N1 cells expressing MYC:EMPTY CTRL vector or MYC:PKR at an MOI of 0.02 for 24 h. Alpha-Tubulin was used as loading control. Band intensity was quantified and normalized to Tubulin for three independent biological replicates, then represented as mean (+/− SD) under the corresponding lane; p-p65 = p-p65/p65 quantification. (C) RT-qPCR on PTGS2 (NF-kB/p65 target) in the same samples as in (B), in mock condition only. Mean (+/− SEM), n = 3 biological replicates. Unpaired t-test. ***p < 0.001. (D) Western blot analysis of DICER, p-p65, p65, CAPSID, and IkBα expression in SINV-GFP infected NoDice FHA:DICER N1 cells at an MOI of 0.02 for 24 h. Before infection, cells were treated with the NF-kB/p65 inhibitor, BAY 11-7082, or the vehicle (DMSO) at the indicated concentrations for 1 h. Alpha-Tubulin was used as loading control. Band intensity was quantified and normalized to Tubulin for three independent biological replicates, then represented as mean (+/− SD) under the corresponding lane; p-p65 = p-p65/p65 quantification. (E) RT-qPCR on PTGS2 (NF-kB/p65 target) in NoDice FHA:DICER N1 mock cells treated with BAY 11-7082 or the vehicle (DMSO) at the indicated concentrations for 1 h. Mean (+/− SEM), n = 3 biological replicates. Ordinary one-way ANOVA with Sidak's correction. **p < 0.01; ns: non-significant. (F) Mean (+/− SEM) of SINV-GFP viral titers in the same samples as in (D), infected at an MOI of 0.02 for 24 h (n = 3 biological replicates) from plaque assay quantification. Ordinary one-way ANOVA test with Sidak's correction. **p < 0.01; ns: non-significant. Source data are available online for this figure.

this contributes to the antiviral effect that has been reported for the AviD isoform (Poirier et al, 2021).

Overall, our study reinforces the idea of a crosstalk between RNAi and other innate immune pathways, in this case NF-kB, and highlights a non-canonical function of human Dicer upon infection (Fig. 9). We showed that Dicer helicase domain is linked to the regulation of immune signaling pathways, and can therefore potentially prevent unwanted activation of these pathways. This new function depends on a non-catalytic action of PKR, which can activate the NF-kB pathway, when no longer interacting with Dicer.

In turn, depending on the virus, this activity results either in an antiviral or a proviral effect.

## Methods

### Plasmids, cloning and mutagenesis

N1 DICER, N1-CM DICER, ΔHEL1 DICER, and ΔHEL1-CM DICER were generated by PCR mutagenesis from pDONR-DICER

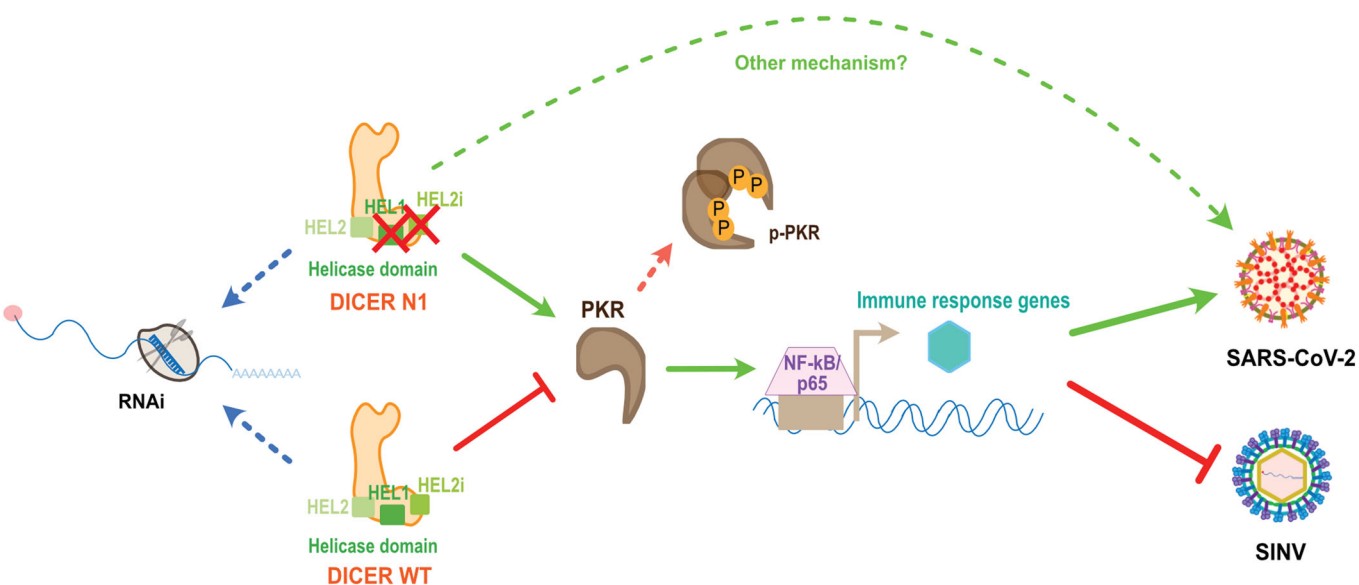

**Figure 9.   The Dicer N1 antiviral property is primed by PKR activation of NF-kB/p65 in mock cells.**

With the first two parts of its helicase domain deleted, Dicer N1 is antiviral against SINV but proviral for SARS-CoV-2. This phenotype does not depend on RNAi, but instead relies on the presence of PKR in N1 cells. Nonetheless, PKR catalytic and/or dimerization activities are not required in that case. PKR is needed to activate a basal transcriptional response in Dicer N1 cells relying on immune-related transcription factors such as NF-kB/p65. NF-kB/p65 is directly involved in Dicer N1 antiviral phenotype thanks to its priming in Dicer N1 mock cells. However, we cannot exclude the existence of another mechanism that may favor SARS-CoV-2 infection.

described in (Girardi et al, 2015). ΔHEL2i and ΔHEL2 DICER and their catalytic mutants (CM) were generated by InFusion mutagenesis (Takara Bio) on pDONR-DICER and pDONR-DICER catalytic mutant vectors. pLenti Flag-HA-V5 vector was modified from pLenti6-V5 gateway vector (Thermo Fisher scientific V49610) by Gibson cloning. WT, N1, N1-CM, ΔHEL1, ΔHEL2i, ΔHEL2, ΔHEL1-CM, ΔHEL2i-CM and ΔHEL2-CM DICER from pDONR plasmids were cloned in pLenti flag-HA-V5 by Gateway recombination.

pLenti MYC:PKR and pLenti human ACE2 vectors were purchased from VectorBuilder. pLenti MYC:CTRL vector was modified from pLenti MYC:PKR by InFusion cloning (Takara Bio). PKR K296R and PKR T451A were obtained by InFusion mutagenesis (Takara Bio) on pLenti MYC:PKR vector.

All primers used are listed in Appendix Table S1.

## Cell lines

HEK293T, HEK293T/NoDice (2.20), HEK293T/NoDiceΔPKR cell lines were a gift from Pr. Bryan Cullen (Duke University, Durham NC, USA) and described in (Bogerd et al, 2014; Kennedy et al, 2015). Vero E6 cells were bought at ATCC (CRL-1586). Cell lines were tested for Mycoplasma contamination.

## Cell culture and transfection

Cells were maintained in Dulbecco's modified Eagle medium (DMEM, Gibco, Life Technologies) supplemented with 10% fetal bovine serum (FBS, Clontech) in a humidified atmosphere with 5% $CO_2$ at 37 °C. Transfections were performed using Lipofectamine 2000 (Invitrogen, Fisher Scientific) according to the manufacturer's instructions.

## BAY 11-7082 treatment

NoDice FHA:DICER WT and N1 cells were treated with BAY 11-7082 (Merck) by replacing the culture medium with a medium containing the indicated BAY 11-7082 concentrations at 1 μM or 5 μM or the corresponding volume of DMSO only in the control conditions. Treatment was maintained for 1 h and media was changed with the infection media with SINV-GFP at an MOI of 0.02 for 24 h. Proteins, RNA and supernatants were then collected and analyzed.

## Lentivirus production and generation of stable cell lines

The lentiviral supernatant from single transfer vector was produced by transfecting HEK293T cells with 1.7 μg of the transfer vector (either pLenti6 FHA-V5 or pLenti MYC), 0.33 μg of the pVSV envelope plasmid (Addgene #8454) and 1.33 μg of pSPAX2 packaging plasmid (Addgene #12260) using Lipofectamine 2000 reagent (Invitrogen, Fisher Scientific) according to the manufacturer's protocol. HEK293T cells were maintained in DMEM (Gibco, Life Technologies) supplemented with 10% Fetal bovine serum (FBS, Clontech). One well from a 6-well plate at 70% confluency was used for the transfection. The medium was replaced 6 h post-transfection. After 48 h, the medium containing viral particles was collected and filtered through a 0.45 μm PES filter. Different cell lines were transduced: NoDice (for N1, N1-CM, ΔHEL1, ΔHEL1-CM, ΔHEL2i, ΔHEL2i-CM, ΔHEL2, and ΔHEL2-CM), NoDiceΔPKR (for WT, NA, ΔHEL1, ΔHEL1-CM, ΔHEL2i, ΔHEL2i-CM, ΔHEL2, and ΔHEL2-CM), NoDiceΔPKR FHA:CTRL or FHA:DICER WT or FHA:DICER N1 (for CTRL, PKR, K296R, T451A), NoDice or NoDiceΔPKR FHA DICER WT or N1 (for ACE2). One well of a 6-well-plate was transduced using 500 μL of filtered

lentiviral supernatant either expressing DICER or PKR or ACE2 constructs, 500 μL of DMEM supplemented with 10% FBS and 4 μg/mL polybrene (Merck, Sigma-Aldrich) for 6 h. Then, the medium was changed with DMEM supplemented with 10% FBS for 24 h and the resistant cell clones were selected with the right antibiotic and subsequently maintained under the selection. For NoDice, the selection lasted 2 weeks with blasticidin at 15 μg/mL (Invivogen). For NoDiceΔPKR, the selection lasted 2 weeks with blasticidin at 10 μg/mL (Invivogen). For NoDiceΔPKR FHA:CTRL or DICER WT, the selection lasted 11 days with hygromycin at 100 μg/mL and for NoDiceΔPKR FHA:DICER N1 it was the same but with hygromycin at 150 μg/mL (Invivogen). Lastly, the ACE2-transduced cells were selected for 10 days with zeocin at 15 μg/mL (Invivogen).

## Viral stocks, virus infection

Viral stocks of SINV WT, SINV-2A-GFP, and SINV-GFP were produced as described in (Girardi et al, 2015). SINV-2A-GFP was described in (Thomas et al, 2003). Cells were infected with SINV (strain AR339), SINV-2A-GFP, and SINV-GFP at an MOI of 0.02 and for SINV-GFP at another MOI of 2 and samples were collected at the different indicated time points in a BSL2 facility.

Viral stocks of SFV were propagated in Vero E6 cells from the initial stock (strain UVE/SFV/UNK/XX1745; EVAg 001V-02468). Cells were infected at an MOI of $1.10^{-4}$ for 24 h in a BSL2 facility.

Viral stocks of EV71 were produced by one passage in BHK21 cells from the initial stock (ATCC VR-1432). Cells were infected at an MOI of 0.1 for 24 h in a BSL2 facility.

Viral stocks of VSV-GFP were propagated in BHK21 cells from the initial stock (strain Indiana isolate PI10; described in (Mueller et al, 2010)). Cells were infected at an MOI of $1.10^{-5}$ for 24 h in a BSL2 facility.

Viral stocks of SARS-CoV-2 were produced by two passages in Vero E6 cells from the initial stock (strain human/DEU/HH-1/2020; EVAg 002V-03991). Cells were infected at an MOI of 0.001 for 48 h in a BSL3 facility.

## Analysis of viral titer by plaque assay

For SINV, SINV-2A-GFP, and SINV-GFP, Vero E6 cells were seeded in 96-well plates and infected with 10-fold serial dilutions of infection supernatants for 1 h. The inoculum was removed and cells were covered with 2.5% carboxymethyl cellulose and cultured for 72 h at 37 °C in a humidified atmosphere of 5% CO₂. Plaques were counted manually under the microscope and viral titer was calculated according to the formula: *PFU/mL = #plaques/(Dilution\*Volume of inoculum)*.

For SFV and VSV-GFP, Vero E6 cells were seeded in 24-well plates format and were infected with 10-fold serial dilutions of infection supernatants for 1 h. The inoculum was removed and cells were covered with 2.5% carboxymethyl cellulose and cultured for 48 h at 37 °C in a humidified atmosphere with 5% CO₂. For plaque visualization, the carboxymethyl cellulose was removed. Cells were fixed with 4% formaldehyde (Merck, Sigma-Aldrich) in PBS (phosphate buffered saline, Gibco) for 20 min at room temperature (RT). Cells were then stained with a 1X crystal violet solution for 20 min at RT (2% crystal violet, Sigma; 20% ethanol; 4% formaldehyde, Merck). Plates were washed with clear water and

plaques were counted manually. The viral titer was calculated according to the formula: *PFU/mL = #plaques/(Dilution\*Volume of inoculum)*.

## Analysis of viral titer by TCID50

For EV71 and SARS-CoV-2, Vero E6 cells were seeded in 96-well plates already containing 10-fold serial dilutions of infection supernatants. Cells were maintained 3 to 4 days at 37 °C in a humidified atmosphere with 5% CO₂. Wells containing dead cells were counted manually under the microscope and viral titer was calculated according to the Spearman-Kaerber 50% lethal dose formula described in (Wilham et al, 2010; Wulff et al, 2012).

## Live-cell imaging

One-hundred thousand NoDice and NoDiceΔPKR polyclonal FHA:DICER WT, N1, ΔHEL1, ΔHEL2i, and ΔHEL2 cells were seeded in a 24-well plate and infected with SINV-GFP at an MOI of 2. Uninfected cells were used as control. GFP fluorescence and phase contrast were observed using a CellcyteX live-cell imaging system (Discover Echo). Six images per well (10X objective) were acquired every 6 h for 24 h and were analyzed with the Cellcyte Studio software to determine cell confluency and GFP relative intensity.

## Western blot analysis

Proteins were extracted and homogenized in the appropriate volume of ice-cold lysis buffer (50 mM Tris-HCl pH 7.5, 5 mM EDTA, 150 mM NaCl, 0.05% SDS, 1% Triton X-100 and protease inhibitor cocktail (complete Mini, Merck)). Proteins were quantified using the Bradford method (Bio-Rad) and 30 μg of total protein extract were loaded on 10% acrylamide-bis-acrylamide gels or 4–20% Mini-PROTEAN TGX Precast Gels (Bio-Rad). Proteins were separated by migration at 135 V in 1X Tris-Glycine-SDS buffer (Euromedex). Proteins were electro-transferred on a nitrocellulose membrane in 1X Tris-Glycine buffer supplemented with 20% ethanol. Equal loading was verified by Ponceau S staining (Merck). Membranes were blocked for 1 h at RT under stirring in 5% milk (Carl Roth) diluted in PBS-Tween (PBS-T) 0.2%. Membranes were probed with the following antibodies overnight at 4 °C under stirring: anti-hDICER (1:1000, A301-937A, Euromedex, Bethyl), anti-PKR (1:1000, ab32506 Abcam), anti-PKR (1:1000, #12297 Cell Signaling), anti-p-PKR (1:1000, ab 81303 Abcam), anti-PACT (1:500, ab75749 Abcam), anti-TARBP2 (1:500, sc-514124, Cliniscience, Santa Cruz), anti-HA-HRP (1:10,000, 12013819001 Merck, Sigma-Aldrich), anti-c-Myc (1:1000, ab32072 Abcam), anti-p-eIF2α (1:1000, #9721 Cell Signaling), anti-α-Tubulin-HRP (1:10,000, 1E4C11 Fisher Scientific), anti-SINV capsid (1:5000, kind gift from Dr. Diane Griffin, Johns Hopkins University School of Medicine, Baltimore, MD), anti-GFP (1:1000, 11814460001 Merck, Sigma-Aldrich), anti-GAPDH-HRP (1:10,000, G9545 Merck, Sigma-Aldrich), anti-SARS-CoV-2 Nucleocapsid (1:1000, ab273167 Abcam), anti-AGO2 (1:250, kind gift from Pr. Gunter Meister, University of Regensburg), anti-ACE2 (1:1000, AF933 Bio-Techne), anti-NF-kB/p65 (1:1000, #8242 Cell Signaling), anti-p-NF-kB/p65 (1:1000, #3033 Cell Signaling) and anti-IkBα (1:1000, #4812 Cell Signaling). The detection was

performed using a specific secondary antibody coupled to the horseradish peroxidase (HRP): anti-mouse-HRP (1:4000, A4416 Merck, Sigma-Aldrich), anti-rabbit-HRP (1:10,000, #31460 Fisher Scientific), anti-rat-HRP (1:10,000, #31470 Fisher Scientific) and anti-goat-HRP (1:10,000, #A15999 Invitrogen). Detection was done using Chemiluminescent Substrate or SuperSignal West Femto maximum sensitivity substrate (Pierce, Fisher Scientific) and visualized with a Fusion FX imaging system (Vilber).

Panels IkBa (Appendix Fig. S4A), DICER (Fig. 4B), and IkBa (Fig. 8D) were contrasted and adjusted uniformly using an image processing software. All the bands were quantified using the FiJi software (Schindelin et al, 2012) and normalized to the housekeeping gene.

### RNA extraction

Total RNA was extracted using TRI Reagent solution (Invitrogen, Fisher Scientific) according to the manufacturer's instructions.

### Northern blot analysis

5 micrograms of total RNA were loaded on a 17.5% acrylamide-urea 4 M gel and resolved in 1X Tris-Borate-EDTA buffer. Small RNAs were electro-transferred onto a nylon Hybond-NX membrane (GE Healthcare) in 0.5X Tris-Borate-EDTA buffer. RNAs were chemically cross-linked to the membrane for 90 min at 65 °C using 1-ethyl-3-[3-dimethylaminopropyl]carbodiimide hydrochloride (EDC) (Merck, Sigma-Aldrich). The membrane was pre-hybridized for 30 min in Perfect Hyb plus (Merck, Sigma-Aldrich) at 50 °C in rotation. Oligodeoxyribonucleotide probes (see Appendix Table S1) were labeled at the 5'-end with 25 μCi of [γ-32P] dATP using T4 polynucleotide kinase (Fischer Scientific). The unbound [γ-32P]dATP was removed with MicroSpin G-25 column (GE Healthcare) and the probe was incubated overnight at 50 °C with the membrane in rotation. The membrane was washed twice with 5X SSC (Saline-Sodium citrate buffer, Euromedex), 0.1% SDS for 15 min at 50 °C and once with 1X SSC (Euromedex), 0.1% SDS for 5 min at 50 °C. The membrane was exposed on a phosphor-imaging plate in a cassette and the signal was recorded using a Typhoon FLA-7000 laser scanner (GE Healthcare).

### RT-qPCR analysis

DNaseI treatment was performed on 1 μg of extracted RNAs (Invitrogen, Fisher Scientific). DNase-treated RNAs were then retro-transcribed using a random nonameric primer with the SuperScript IV (SSIV) reverse transcriptase according to the manufacturer's instructions (Invitrogen, Fisher Scientific). Real-time quantitative PCR was performed on 1/10 dilution of cDNA using SYBR green PCR master mix (Fisher Scientific) and the primers listed in Appendix Table S1.

### Strand-specific semi-quantitative RT-PCR on SINV antigenome

Negative-strand-specific reverse transcription was performed using a primer specific to the 5' region of the SINV plus-strand genome (nucleotides 1 to 42—Appendix Table S1). 100 ng of RNA, 1 μL of 2 μM specific primer, 1 μL of 10 mM dNTPs and water up to 13 μL

final volume were mixed and incubated at 65 °C for 5 min. 4 μL of 5X SSIV Buffer (Invitrogen, Fisher Scientific), 1 μL of 0.1 M DTT (dithiothreitol), 1 μL of RNase inhibitor (Ribolock, Fisher Scientific) and 1 μL of SSIV reverse transcriptase (Invitrogen) were added and the mix was incubated at 55 °C for 10 min then at 80 °C for 10 min. 1/20 of the cDNA was amplified by PCR using the GoTaq DNA polymerase mix (Promega) with specific antigenome primers (Appendix Table S1). PCR products were loaded on a 1% agarose gel and gel pictures were taken with a Fusion FX imaging system (Vilber).

### Immunoprecipitation

Immunoprecipitations were done on tagged proteins. Cells were harvested, infected at the corresponding MOI and time and washed once with ice-cold PBS (Gibco, Life Technologies) and resuspended in 600 μL of ice-cold lysis buffer (50 mM Tris-HCl pH 7.5, 140 mM NaCl, 1.5 mM MgCl$_2$, 0.1% NP-40 and protease inhibitor cocktail (complete Mini, Merck, Sigma-Aldrich). Cells were lysed for 10 min incubation on ice and lysates were cleared with a 15 min centrifugation at $12,000 \times g$ and 4 °C. 25 μL of the lysates were kept as protein INPUT. Then, samples were divided in two and 40 μL of magnetic beads coated with monoclonal anti-HA or anti-MYC antibodies were added (MACS purification system, Miltenyi Biotech). The samples were incubated for 1 h at 4 °C under rotation. Samples were sequentially loaded onto μColumns (MACS purification system, Miltenyi Biotech). The columns were washed 4 times with 200 μL of lysis buffer. The elution was done with 95 °C-pre-heated 2X Laemmli buffer (20% glycerol, 4% SDS, 125 mM Tris-HCl pH 6.8, 10% (v/v) 2-β-mercaptoethanol, 0.004% Bromophenol Blue). Proteins were analyzed by western blot.

### Immunostaining

Cells were plated on Millicell EZ 8-well slide (Merck Millipore) and infected with the different viruses at the corresponding MOI and infection time. Then, cells were fixed for 10 min at RT with 4% formaldehyde (Merck, Sigma-Aldrich) diluted in PBS 1X (Gibco). Cells were incubated with a blocking solution (5% normal goat serum, 0.1% Triton X-100, PBS 1X) for 1 h at RT. Primary J2 antibody diluted in blocking solution was added at 1:1000 dilution for 3 h at RT. Cells were washed three times with PBS 1X-0.1% Triton X-100 (Phosphate buffered saline-Triton, PBS-T) and incubated 1 h at RT in the dark with a secondary antibody solution containing goat anti-mouse Alexa 594 (A11032, Invitrogen, Fisher Scientific) fluorescent-coupled antibody diluted to 1:1000 in PBS-T solution. After three washes with PBS-T, cell nuclei were stained with DAPI diluted to 1:5000 in PBS 1X for 2 min at RT in the dark (Life Technologies, Fischer Scientific). Slides were finally mounted on coverslips and the Fluoromount-G mounting media (Southern Biotech). Images were acquired using an epifluorescence BX51 (Olympus) microscope with a mercury source illuminator (U-RFL-T, Olympus) and with ×40 immersion oil objective.

### Small RNA cloning and sequencing

NoDice FHA:DICER WT or N1 cells were plated in P150mm Petri dishes at 25,000,000 cells per dish and infected the following day with SINV-GFP at an MOI of 0.02 for 24 h. Cells were collected in

1.3 mL of ice-cold lysis buffer (50 mM Tris-HCl pH 7.5, 150 mM NaCl, 5 mM EDTA, 1% NP-40, 0.5 mM DTT, 0.1 U/mL of RNase inhibitor and protease inhibitor cocktail (complete Mini, Merck, Sigma-Aldrich)). Cells were lysed by putting the extract on ice for 15 min. First, human AGO proteins were immunoprecipitated from whole extract to enrich for loaded small RNAs. AGO-IP was done following the protocol described in (Hauptmann et al, 2015). Briefly, 50 µL of magnetic Dynabeads coupled to G protein (Invitrogen, Fisher Scientific) were coupled to anti-Flag M2 antibody (F1804, Merck, Sigma-Aldrich) overnight at 4 °C under rotation. The day after, Flag-TNRC6B WT or mutant (with Tryptophans mutated to Alanines; called TNRC6B Ala) peptides were incubated with the coupled beads for 3 h at 4 °C under rotation. Meanwhile, lysates were cleared with a centrifugation step at 10,000 × g for 10 min at 4 °C and 100 µL were kept as INPUT (20 µL for proteins and 80 µL for RNAs). Then, lysates were divided in two and put on the beads coupled to the peptides and incubated at 4 °C for 3 h under rotation. After 4 washing steps with 300 µL of ice-cold washing buffer, beads were split: 280 µL were eluted with 1 mL of TRI Reagent solution (Invitrogen, Fisher Scientific) and RNAs were isolated according to the manufacturer's instructions; 20 µL were kept for protein analysis of IP efficiency by western blot and 20 µL of 95 °C-pre-heated 2X Laemmli buffer were added (20% glycerol, 4% SDS, 125 mM Tris-HCl pH 6.8, 10% (v/v) 2-β-mercaptoethanol, 0.004% Bromophenol Blue).

Small RNA libraries were prepared using the Illumina TruSeq small RNA library preparation kit according to the manufacturer's protocol (RS-200-0012, Illumina).

Samples were analyzed with a Bioanalyzer device, and only AGO-IPs with TNRC6B wild-type peptide in infected conditions were analyzed in triplicates for both cell lines, while only one AGO-IP TNRC6B Ala control was analyzed for each cell line.

Libraries were sequenced on an Illumina HiSeq 4000 sequencer as single-end 50 base reads at the GenomEast platform at the Institute of Genetics and Molecular and Cellular Biology (Illkirch, France). Image analysis and base calling were performed using RTA version 2.7.7 and bcl2fastq version 2.20.0.422.

## mRNA sequencing

NoDice FHA:DICER WT or N1 cells were plated in P100mm Petri dishes at 5,000,000 cells per dish and infected the following day with SINV-GFP at an MOI of 2 for 12 h or 0.02 for 24 h. Cells were collected in Trizol reagent (Invitrogen, Thermo Fisher Scientific) and RNAs were isolated according to the manufacturer's instructions.

Library preparation and sequencing were performed at the GenomEast platform at the Institute of Genetics and Molecular and Cellular Biology (Illkirch, France). RNA-Seq libraries were generated according to the manufacturer's instructions from 250 ng of total RNA using the Illumina Stranded mRNA Prep, Ligation kit (Reference Guide - PN 1000000124518) and IDT for Illumina RNA UD Indexes Ligation (Illumina, San Diego, USA). Briefly, Oligo(dT) magnetic beads were used to purify and capture the mRNA molecules containing polyA tails. The purified mRNAs were then fragmented at 94 °C for 2 min and copied into first-strand complementary DNA (cDNA) using reverse transcriptase and random primers. Second-strand cDNA synthesis further generated blunt-ended double-stranded cDNA and incorporated

dUTP in place of dTTP to achieve strand specificity by quenching the second strand during amplification. Following A-tailing of DNA fragments and ligation of pre-index anchors, PCR amplification was used to add indexes and primer sequences and to enrich DNA libraries (30 s at 98 °C; [10 s at 98 °C, 30 s at 60 °C, 30 sec at 72 °C] × 12 cycles; 5 min at 72 °C). Surplus PCR primers were further removed by purification using SPRIselect beads (Beckman-Coulter, Villepinte, France) and the final libraries were checked for quality and quantified using capillary electrophoresis.

Libraries were sequenced on an Illumina HiSeq 4000 sequencer as paired-end 100 base reads. Image analysis and base calling were performed using RTA version 2.7.7 and bcl2fastq version 2.20.0.422.

## Bioinformatics analysis of small RNA sequencing data

Sequencing reads of sRNA-seq libraries were processed and analyzed with the following workflow. Cutadapt v1.16 (Martin, 2011) was first run to trim the 3' adapter (command: cutadapt -a TGGAATTCTCGGGTGCCAAGG -e 0.1 --no-indels -O 6 -m 18 --discard-untrimmed -o <sample>-cutadapt.fastq <sample>.fastq) and an additional filter was applied to only keep 18- to 32-nt long trimmed reads for further analyses. Sample quality checks were performed before and after these preprocessing steps using FastQC v0.11.8 (https://www.bioinformatics.babraham.ac.uk/projects/fastqc/). Preprocessed reads were then mapped simultaneously to the human (GENCODE Human (GRCh38.p13) release 41) and SINV-GFP (private sequence derived from NC_001547.1 – RefSeq database) genomes, using Bowtie v1.3.1 (Langmead et al, 2009) (command: bowtie -q --phred33-quals -v 2 -y -a --best --strata -m 30 -x hg38coreGencodeSINVGFP <sample>-preprocessed.fastq <sample > -hg38coreGencodeSINVGFP.bwtmap). Only alignments from the lowest mismatch stratum with at most 2 mismatches were reported for each read, provided that their number was not exceeding 30.

For each library, small RNA reads deriving solely from SINV-GFP were computationally extracted and further characterized. Representations of their length distribution and localization along the viral genome were both made in R using a custom Shiny application based on the ggplot2 (Wickham, 2009) and Bioconductor Gviz (Hahne and Ivanek, 2016) packages respectively.

Furthermore, expressed human miRNAs (miRBase v22.1 (Kozomara and Griffiths-Jones, 2011), among which known mirtrons (Wen et al, 2015), were also identified and quantified in each library using BEDTools v2.30.0 (Quinlan and Hall, 2010) by comparing their genomic coordinates to those of the original aligned reads and by counting only reads showing at least 80% overlap with the miRNA sequence (command: bedtools intersect -a <sample>-hg38coreGencodeSINVGFP.bwtmap.bed -b hsa-matmir.bed -f 0.80 -wa -wb -s). Multiple mapped reads were weighted by the number of mapping sites in other miRNAs, and the final counts obtained for each miRNA were rounded down before further analysis. Differential miRNA expression analysis between the two AGO-IP conditions was then performed with the DESeq2 package (Love et al, 2014) by using an adjusted *p*-value < 0.05 and an absolute log2 fold change >1 as thresholds to define statistical significance. The heatmap of the 50 most abundant miRs expressed in the AGO-IP samples (regularized-logarithm (rlog) transformed counts) and the MA plot of the DESeq2 results were made in R with

the Heatmaply (Galili et al, 2018) and ggplot2 (Wickham, 2009) packages respectively, inside the same custom Shiny application as previously mentioned.

## Bioinformatics analysis of mRNA sequencing data

RNA-seq data were processed and analyzed using the following workflow. The first 5'-end base of each read, a T-overhang added during the library preparation with the Illumina Stranded mRNA protocol, was first trimmed, before using Skewer v0.2.2 (Jiang et al, 2014) in paired-end mode for average quality read filtering and adapter trimming (command: skewer -Q 25 -x CTGTCTCTTATACACATCT -y CTGTCTCTTATA-CACATCT -l 31 -m pe -t 2 -o <sample> --quiet <sample>_R1.fq <sample>_R2.fq). Sample quality checks were performed before and after these preprocessing steps using FastQC v0.11.8 (https://www.bioinformatics.babraham.ac.uk/projects/fastqc/). Then, human full-length protein-coding transcripts (GENCODE Human (GRCh38.p13) release 41) and SINV-GFP both genomic and subgenomic transcripts (private sequence derived from NC_001547.1 – RefSeq database) were quantified using Salmon v1.10.0 (Patro et al, 2017) in mapping-based mode with the selective alignment algorithm and a decoy-aware transcriptome (command: salmon quant -p 6 -i index/gencode41.hg38.-sinv.decoys_index --libType A --seqBias --gcBias --numBootstraps 30 -1 data/<sample>_preprocessed_R1.fq.gz -2 data/<sample>_preprocessed_R2.fq.gz -o salmon-quants/<sample>). Transcript-level abundance and count estimates thus obtained were next imported into R and further analyzed with the tximport (Soneson et al, 2015) and DESeq2 (Love et al, 2014) packages. Briefly, original transcript-level counts were summarized to gene-level estimated counts and an offset was produced from transcript-level abundance estimates to correct for changes to the average transcript length across samples. Differential expression analyses between tested conditions were then conducted at the gene-level, and statistical significance was defined with an adjusted p-value < 0.05 and an absolute log2 fold change >1 thresholds. Finally, gene set enrichment analysis (GSEA) studies were performed using the GSEA_4.3.2 Mac application, as made available by the Broad Institute and the University of California, San Diego (https://www.gsea-msigdb.org/gsea/index.jsp) (Mootha et al, 2003; Subramanian et al, 2005). These analyses were conducted on the DESeq2 normalized counts of each comparison dataset against the Hallmark Gene Sets (v2023.1) provided by the Molecular Signatures Database (MSigDB) (Liberzon et al, 2015), and by using the Human_Ensembl_Gene_ID_MSigDB.v2023.1.Hs.chip annotation file and all other parameters by default, except the permutation type which was set to gene_set.

Heatmaps and volcano plots of RNA-seq differential expression data were generated in R using respectively the pheatmap package and a custom Shiny application based on the ggplot2 (Wickham, 2009) and ggrepel (https://github.com/slowkow/ggrepel) packages, while the global gene set enrichment dot plot was adapted from VisualizeRNA-seq (https://github.com/GryderArt/VisualizeRNAseq).

## Analysis of transcription factor enrichment

Lists of NF-kB/p65, STAT1, STAT2, IRF2, and IRF3 regulated genes were downloaded from Harmonizome (Rouillard et al, 2016). Genes from these lists that were differentially expressed between the indicated RNAseq experimental conditions were selected, and their log2 fold change was plotted as cumulative distribution histograms. The two-sample Kolmogorov–Smirnov test was used to assess whether each distribution was statistically different from the distribution of WT DICER cells infected with SINV vs. mock infected. p-values are indicated on each histogram.

## Statistical analysis

All the plaque assay statistical analyses were done with PRISM 10 (GraphPad) using one-way or two-way ANOVA with Sidak's or Dunnett's corrections with a p-value threshold at p < 0.05 on the logarithmic values.

All the qPCR statistical analyses were done with PRISM 10 (GraphPad) using t-test with Welch's correction with a p-value threshold at p < 0.05 on the log2 values or unpaired t-test on the log2 values.

Detailed statistical analysis for each figure can be found in the source data file "Statistics_details.xlxs".

# Data availability

The datasets produced in this study have been deposited in NCBI's Gene Expression Omnibus (Edgar et al, 2002) and are accessible through GEO Series accession number GSE241798.

# Peer review information

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

## Acknowledgements

We would like to thank members of the laboratory for fruitful discussions, as well as Pr. Petr Svoboda (Institute of Molecular Genetics of the Czech Academy of Sciences, Prague, Czech Republic) for sharing unpublished data and exchanging ideas, and the SCIGNE (Scientific Cloud Infrastructure in Grand Est) platform at the Hubert Curien Pluridisciplinary Institute (IPHC, Strasbourg, France) as well as its computing team for providing the technical support, computing and storage facilities needed to analyze part of the sequencing data. This work of the Interdisciplinary Thematic Institute IMCbio+, as part of the ITI 2021-2028 program of the University of Strasbourg, CNRS and Inserm, was

supported by IdEx Unistra (ANR-10-IDEX-0002), by SFRI-STRAT'US project (ANR-20-SFRI-0012), EUR IMCBio (IMCBio ANR-17-EURE-0023) and Equipex Insectarium ANR-11-EQPX-0022 under the framework of the French Investments for the Future Program. It also benefited from an IdEx University of Strasbourg 2022 Attractivity grant. Sequencing was performed by the GenomEast platform, a member of the 'France Génomique' consortium (ANR-10-INBS-0009). MB was funded by a doctoral fellowship from the Ministère de l'enseignement supérieur, de la recherche et de l'innovation and from the Fondation pour la Recherche Médicale (grant number FDT202204014935). LG was funded by a postdoctoral fellowship from the Fondation pour la Recherche Médicale (funding number SPF202209015746). SP also received funding from the European Research Council (ERC-CoG-647455 RegulRNA).

## Author contributions

**Morgane Baldaccini**: Conceptualization; Formal analysis; Validation; Investigation; Methodology; Writing—original draft; Writing—review and editing. **Léa Gaucherand**: Formal analysis; Validation; Investigation; Methodology; Writing—review and editing. **Béatrice Chane-Woon-Ming**: Data curation; Software; Formal analysis; Visualization; Writing—review and editing. **Mélanie Messmer**: Formal analysis; Investigation; Methodology; Writing—review and editing. **Floriane Gucciardi**: Formal analysis; Investigation; Methodology. **Sebastien Pfeffer**: Conceptualization; Supervision; Funding acquisition; Validation; Writing—original draft; Project administration; Writing—review and editing.

## Disclosure and competing interests statement

The authors declare no competing interests.

# Expanded View Figures

**Figure EV1.   Dicer N1 retains the same miRNA profile and interacting partners upon infection.**

(**A**) Representation of small RNA reads distribution (in percent) upon AGO-IP in the three replicates for NoDice FHA:DICER WT and N1. Green: reads mapping to human RNAs only; Red: reads mapping to viral RNAs only; gray: reads mapping to both human and viral RNAs. (**B**) Heatmap representing the relative expression levels of the 50 most abundant human miRNAs in all three replicates upon AGO-IP followed by small RNA sequencing in SINV-GFP infected NoDice FHA:DICER WT and N1 cells at an MOI of 0.02 for 24 h. (**C**) MA plot for miRNA enrichment upon AGO-IP followed by small RNA sequencing in SINV-GFP infected NoDice FHA:DICER N1 vs WT cells at an MOI of 0.02 for 24 h. Each dot represents a miRNA either up- (red), down- (blue) or un-regulated (gray). $n = 3$ biological replicates (Wald test, DESeq2 package). (**D**) Western blot analysis of DICER and CAPSID expression in SINV-GFP infected polyclonal NoDice FHA:DICER WT, N1 and N1-CM cells at an MOI of 0.02 for 24 h. Alpha-Tubulin was used as loading control. Band intensity was quantified and normalized to Tubulin for three independent biological replicates, then represented as mean ($+/-$ SD) under the corresponding lane. (**E**) Mean ($+/-$ SEM) of SINV-GFP viral titers in the same samples as in (**D**), infected at an MOI of 0.02 for 24 h ($n = 3$ biological replicates) from plaque assay quantification. Ordinary one-way ANOVA test with Dunnett's correction. ****$p < 0.0001$. (**F**) Western blot analysis of Dicer interacting partners upon HA-IP in NoDice FHA:DICER WT, N1 and N1-CM cells, in mock (left) or SINV-GFP infected (right) conditions at an MOI of 2 for 6 h ($n = 3$ biological replicates). Anti-HA antibodies were used to validate the immunoprecipitation and Ponceau was used as a loading control. Source data are available online for this figure.

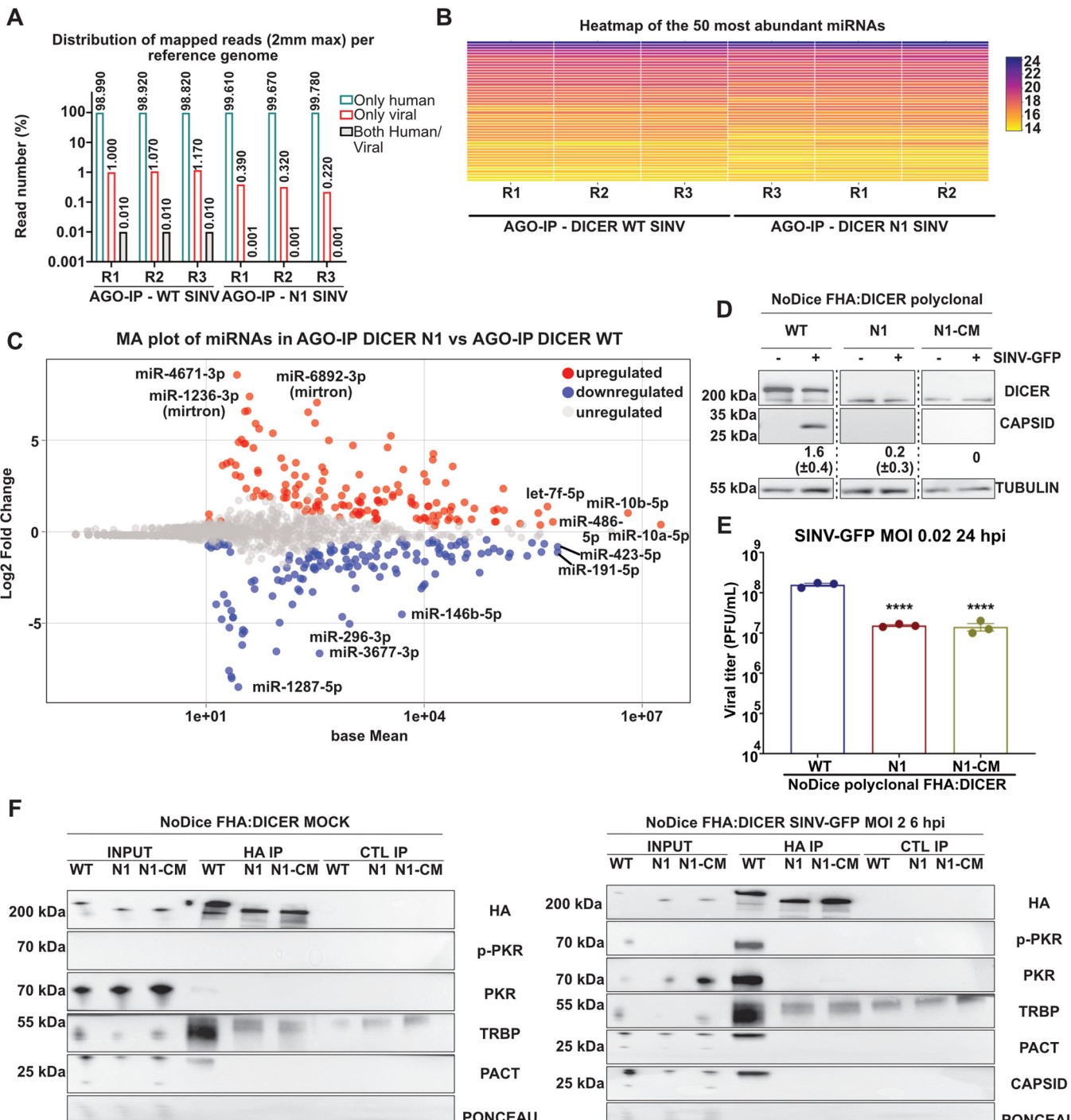

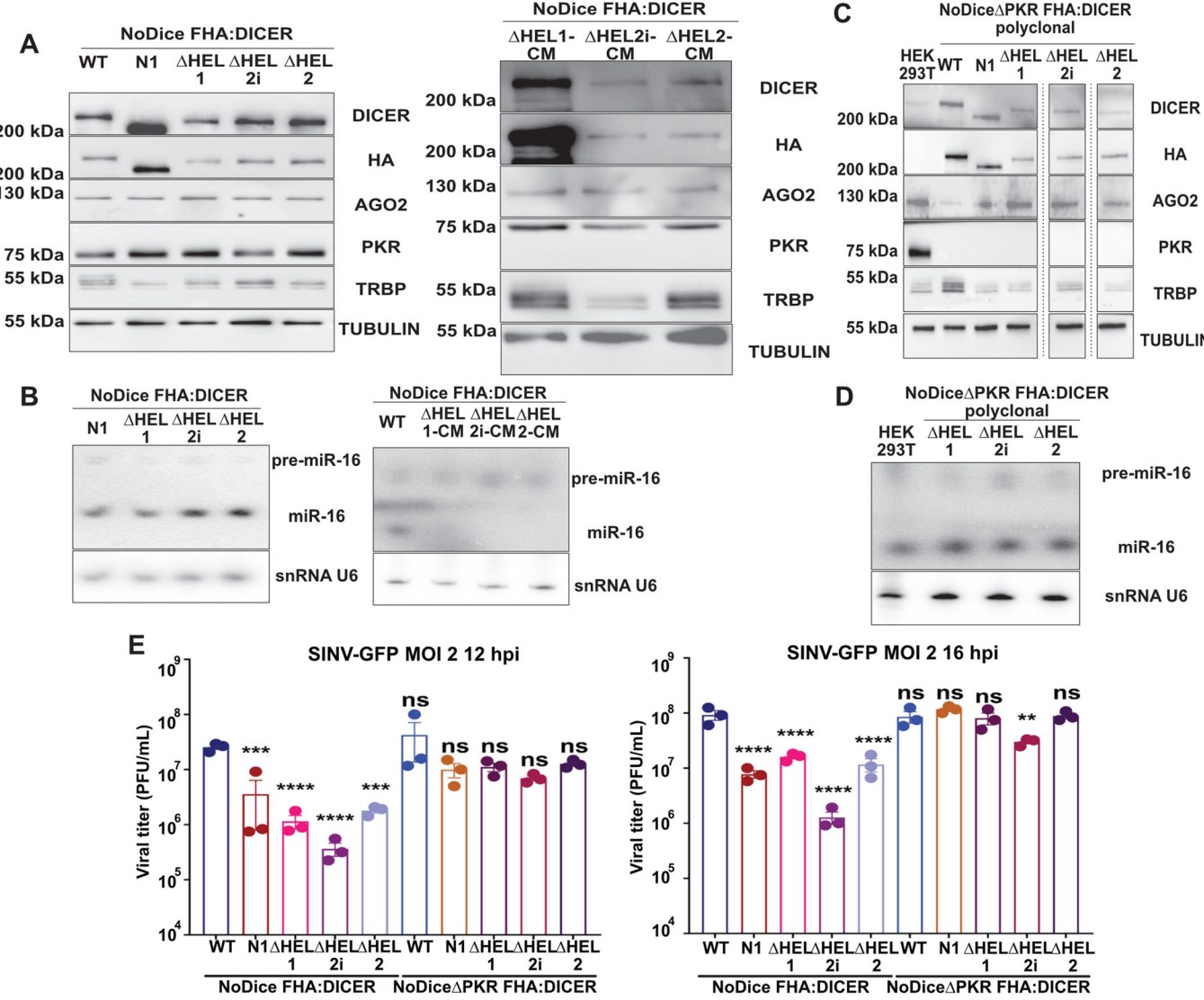

**Figure EV2.  The helicase sub-domains as well as PKR are important for Dicer antiviral activity.**

(A) Western blot analysis of DICER, HA, AGO2, PKR and TRBP expression in the monoclonal cell lines NoDice FHA:DICER WT, N1, ΔHEL1, ΔHEL2i and ΔHEL2 (left, $n = 3$ biological replicates) and ΔHEL1-CM, ΔHEL2i-CM and ΔHEL2-CM (right, $n = 2$ biological replicates). Alpha-Tubulin was used as a loading control. (B) Northern blot analysis of mirR-16 expression in the same samples as in (A) ($n = 3$ biological replicates for ΔHEL1, 2i and 2; $n = 2$ biological replicates for ΔHEL1-CM, 2i-CM and 2-CM). Expression of snRNA U6 was used as a loading control. (C) Western blot analysis of DICER, HA, AGO2, PKR and TRBP expression in the polyclonal cell lines NoDiceΔPKR FHA:DICER WT, N1, ΔHEL1, ΔHEL2i and ΔHEL2 ($n = 1$ biological replicate). Alpha-Tubulin was used as a loading control. (D) Northern blot analysis of mirR-16 expression in the same samples as in (C) ($n = 2$ biological replicates). Expression of snRNA U6 was used as a loading control. (E) Mean ($+/-$ SEM) of SINV-GFP viral titers in the polyclonal cell lines NoDice and NoDiceΔPKR FHA:DICER WT, N1, ΔHEL1, ΔHEL2i and ΔHEL2, infected at an MOI of 2 for 12 h (left) or 16 h (right) from plaque assay quantification ($n = 3$ biological replicates). Ordinary one-way ANOVA test with Dunnett's correction. **$p < 0.01$; ***$p < 0.001$; ****$p < 0.0001$; ns: non-significant. Source data are available online for this figure.

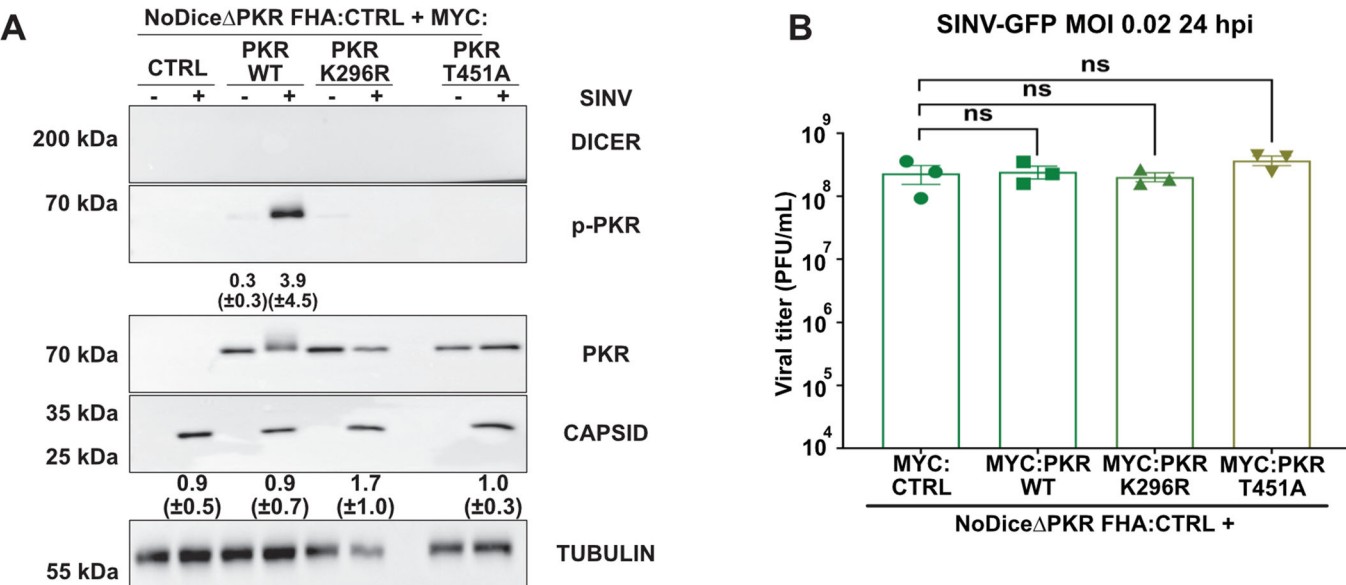

**Figure EV3. PKR dimerization and/or catalytic activities do not change the infection outcome in NoDiceΔPKR FHA:CTRL cells.**

(A) Western blot analysis of DICER, p-PKR, PKR and CAPSID expression in SINV-GFP infected NoDiceΔPKR FHA:CTRL cells expressing MYC:EMPTY CTRL vector, MYC:PKR, MYC K296R or MYC:T451A at an MOI of 0.02 for 24 h. Alpha-Tubulin was used as loading control. Band intensity was quantified and normalized to Tubulin for three independent biological replicates, then represented as mean (+/− SD) under the corresponding lane; p-PKR = p-PKR/PKR quantification. (B) Mean (+/− SEM) of SINV-GFP viral titers in the same samples as in (A), infected at an MOI of 0.02 for 24 h (*n* = 3 biological replicates) from plaque assay quantification. Ordinary one-way ANOVA test with Dunnett's correction. ns: non-significant. Source data are available online for this figure.

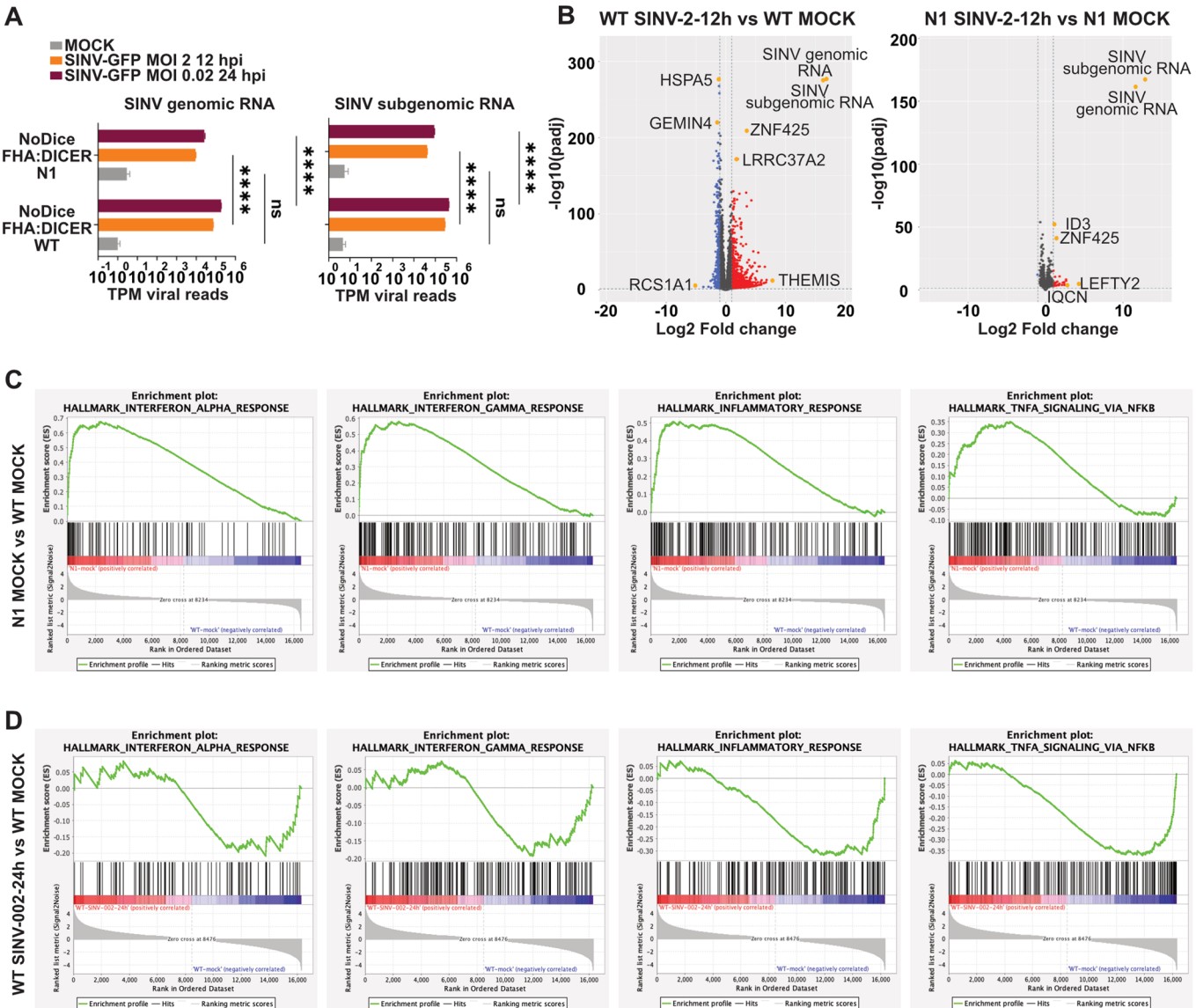

**Figure EV4.  Dicer N1 cells express a different gene set from Dicer WT cells.**

(A) SINV-GFP genomic and subgenomic read distribution detected by RNA-sequencing analysis in NoDice FHA:DICER WT or N1 cells uninfected (Mock, gray) or infected at an MOI of 2 for 12 h (orange) or 0.02 for 24 h (purple). The viral reads number (mean +/− SEM) is normalized to the total mapped reads in each condition. TPM: transcripts per million. Ordinary two-way ANOVA test with Sidak's correction. ****$p$ < 0.0001; ns: non-significant. $n$ = 3 biological replicates, error bars represent SEM. (B) Volcano plots showing for each gene the log2 fold change and adjusted p value (Wald test, DESeq2 package) between SINV-infected (MOI of 2 for 12 h) and mock NoDice FHA:DICER WT cells (left), or SINV-infected and mock NoDice FHA:DICER N1 cells (right). Each gene is marked as a dot (red: upregulated, blue: downregulated, gray: unchanged). The horizontal line denotes an adjusted $p$-value of 0.05 and the vertical ones the Log2 fold change cut-offs (−1 and 1) ($n$ = 3 biological replicates). (C,D) GSEA enrichment plots for selected biological states and processes linked to inflammatory and antiviral pathways for mock NoDice FHA:DICER N1 vs mock NoDice FHA:DICER WT (C), or SINV-infected (MOI of 0.02 for 24 h) vs. mock NoDice FHA:DICER WT cells (D). Source data are available online for this figure.

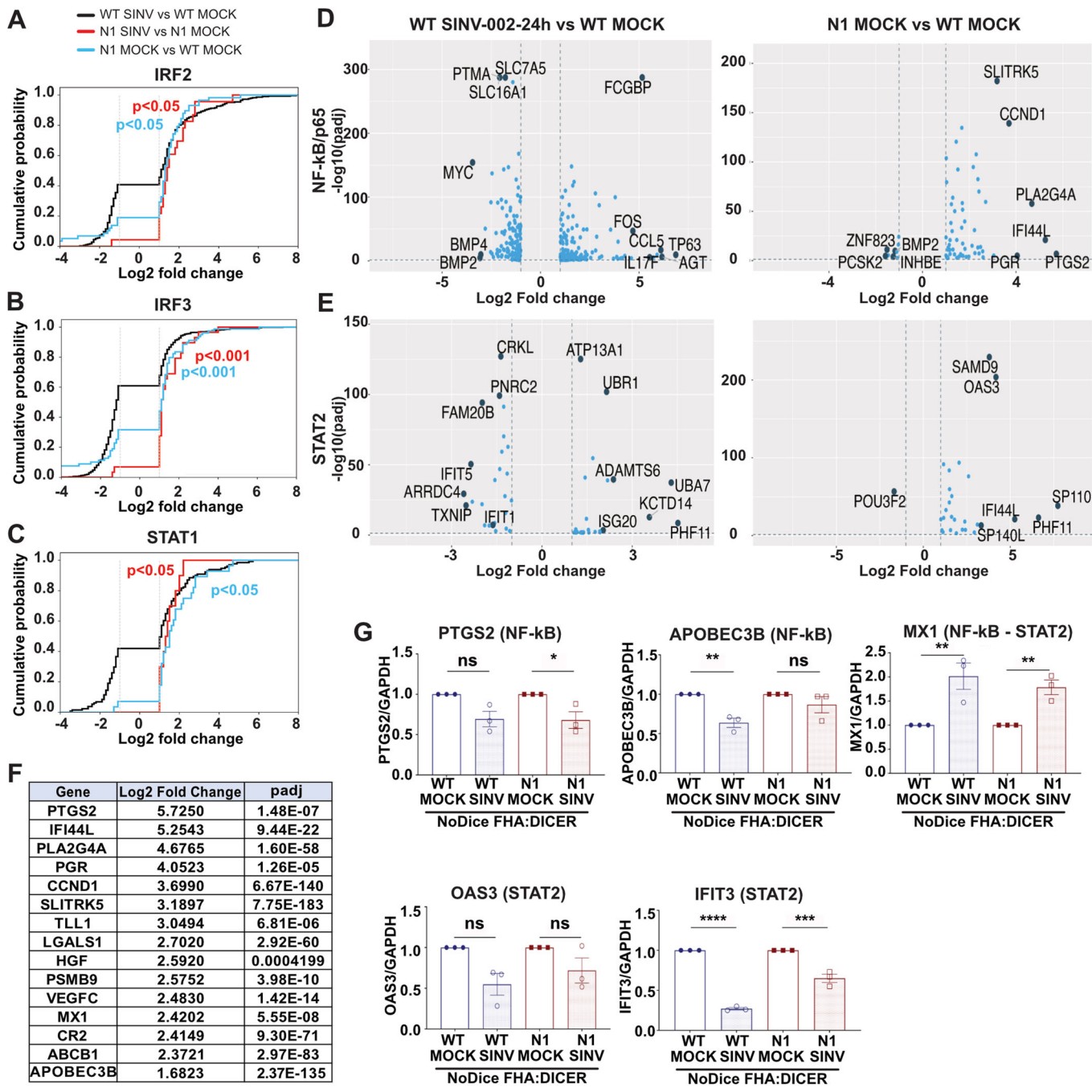

**Figure EV5.  Immune-related transcription factors activation is involved in the deregulation of Dicer N1 cells mRNAs.**

(A–C) Histograms representing the cumulative probability of differentially expressed genes controlled by the transcription factors IRF2 (**A**), IRF3 (**B**) or STAT1 (**C**), plotted according to their Log2 fold change. The vertical lines stand for the Log2 fold change cut-offs (−1 and 1). The two-sample Kolmogorov–Smirnov test was used to assess whether each distribution was statistically different from the distribution of NoDice FHA:DICER WT cells infected with SINV vs. mock. p-values are indicated on each histogram. Black: WT SINV-002-24h vs WT MOCK; red: N1 SINV-002-24h vs N1 MOCK; blue: N1 MOCK vs WT MOCK. (**D,E**) Volcano plots for differentially expressed genes (DEGs) under the control of NF-kB/p65 (**D**) or STAT2 (**E**). Each gene is marked as a dot and plotted based on its log2 fold change and adjusted p values (Wald test, DESeq2 package) comparing SINV-infected (MOI of 0.02 for 24 h) vs mock NoDice FHA:DICER WT cells (left), or mock NoDice FHA:DICER WT vs mock NoDice FHA:DICER N1 cells (right). The horizontal line denotes an adjusted p-value of 0.05 and the vertical ones the Log2 fold change cut-offs (−1 and 1). n = 3 biological replicates. (**F**) Table of 15 representative upregulated NF-kB/p65 targets from the DEGs in the comparison mock NoDice FHA:DICER N1 vs mock NoDice FHA:DICER WT. Classification was made according to their Log2 fold change values. padj: adjusted p-value (Wald test, DESeq2 package). (**G**) RT-qPCR on selected DEGs controlled by either NF-kB/p65 or STAT2 in NoDice FHA:DICER WT and N1 cells infected or not with SINV-GFP at an MOI of 0.02 for 24 h. Mean (+/− SEM); n = 3 biological replicates. One-way ANOVA with Sidak's correction. *p < 0.05; **p < 0.01; ***p < 0.001; ****p < 0.0001; ns: non-significant. Source data are available online for this figure.

