## [Peer Review File · The EMBO Journal]

The helicase domain of human Dicer prevents RNAi-independent activation of antiviral and inflammatory pathways

Morgane Baldaccini, Lea Gaucherand, Béatrice Chane-Woon-Ming, Mélanie Messmer, Floriane Gucciardi, and Sebastien Pfeffer

Corresponding author(s): Sebastien Pfeffer (s.pfeffer@ibmc-cnrs.unistra.fr)

Review Timeline:

Transfer from Review Commons:	4th Oct 23
Editorial Decision:	11th Oct 23
Revision Received:	29th Nov 23
Editorial Decision:	15th Dec 23
Revision Received:	19th Dec 23
Accepted:	8th Jan 24

Editor: Ioannis Papaioannou

Transaction Report: This manuscript was transferred to The EMBO JOURNAL following peer review at Review Commons.

Review #1

1. Evidence, reproducibility and clarity:

Evidence, reproducibility and clarity (Required)

The study largely focuses on the use of a 293 cell line that lacks a functional Dicer gene originally identified by the Cullen group. Baldaccini use this cell line, referred to as NoDice cells, to reconstitute various Dicer isoforms that have thus far been described in a variety of settings (e.g., stem cells and oocytes). Collectively, these data demonstrate the capacity of certain N-terminal truncations of Dicer to inhibit Sindbis virus and reduce the presence of viral dsRNA, supporting some of the observations made thus far concerning an antiviral role for mammalian Dicer. For other viruses, this impact was significantly more modest (SFV reduction is less than a log) or was not observed at all (VSV and SARS-CoV-2). The authors then go on to characterize the nature of the observed antiviral activity and ultimately implicate PKR and the induction of NF- κ B in priming the cell's antiviral defenses. Importantly, the group also found that this antiviral activity neither required the nuclease activity of Dicer nor the kinase activity of PKR - providing evidence against antiviral RNAi in mammals. In all, the data would seem to suggest that Dicer can act as a dsRNA sensor and can mediate the activation of an NF- κ B response - akin to what is observed in response to NOD or some TLR engagement. In all, it is the opinion of this reviewer that this work brings additional clarity to a concept that remains controversial in the field and therefore embodies something meaningful for the community.

With that said, there are a few issues that require additional attention. The first of these is textual. The introduction of the paper accurately describes the evidence in support of mammalian RNAi but does not invest the same time in discussing the data to the contrary. For example, Seo et al demonstrated that virus infection results in poly-adp-ribosylation of RISC preventing RNAi activity (PMID: 24075860), Uhl et al showed that IFN-induced ADAR1 resolves dsRNA in the cell and prevents RNAi (PMID: 37017521), and Tsai et al showed that virus-derived small RNAs are not loaded into the RISC in a manner that would enable antiviral activity (PMID 29903832). None of this work is referenced in this manuscript and it generates an unbalanced introduction as it relates to the controversy surrounding the idea of RNAi in mammals.

The second issue that would further strengthen this paper relates to the fact that the authors spend a considerable amount of time discussing the data of Figure 6 and 7 as conditions that are defined by a Dicer that can not be processive in its nuclease

activity (WT) vs. one that can (N1). However, there seems to be little consideration about the fact that the introduction of WT Dicer into these cells also restores miRNA biology whereas N1 appears to remain only partially functional (based on the data of Fig 3E). Given this, it seems the authors should verify that the high baseline of NFkB signaling that is being observed when comparing WT to N1 is not a product of restored miRNA function in WT cells, in contrast to the hypotheses outlined in the manuscript. This could be addressed by silencing Drosha or DGCR8 in the Dicer knockout cells prior to their reconstitution of Dicer. In the opinion of this reviewer, this experimental control would significantly strengthen the conclusions the authors are making here.

2. Significance:

Significance (Required)

In the manuscript entitled, "Canonical and non-canonical contributions of human Dicer helicase domain in antiviral defense" Baldaccini et al. describe their findings concerning the ability of certain N-terminal deletion variants of Dicer in contributing to mammalian antiviral activity. The concept of a functional antiviral RNAi system in mammals is a contentious one with many publications including data both in support of its existence and opposing this idea. In this manuscript, Baldaccini et al. perform a wide range of well-controlled experiments to specifically address aspects of those reports to both provide clarity in what has been documented thus far and to expand on those concepts further.

3. How much time do you estimate the authors will need to complete the suggested revisions:

Estimated time to Complete Revisions (Required)

(Decision Recommendation)

Less than 1 month

4. *Review Commons* values the work of reviewers and encourages them to get credit for their work. Select 'Yes' below to register your reviewing activity at Web of Science Reviewer Recognition Service (formerly Publons); note that

the content of your review will not be visible on Web of Science.

Yes

Review #2

1. Evidence, reproducibility and clarity:

Evidence, reproducibility and clarity (Required)

****Summary****

Whether RNAi is used as an antiviral mechanism in mammals has been a hotly debated issue. The research team previously published several papers on the roles of Dicer in siRNA/miRNA biogenesis and in antiviral responses. They have recently reported that the helicase domain of human Dicer specifically interacts with several proteins that are involved in the IFN response, including PKR. In this study, Baldaccini et al. investigated the involvement of Dicer in antiviral response using various mutants of human Dicer. They showed that deletion mutants of helicase domain exhibit antiviral activity that requires the presence of PKR. They further demonstrated that one of the mutants, N1-Dicer showed antiviral activity in an RNAi-independent manner but depending on the presence of either native PKR or kinase deficient mutants. Transcriptomic analysis revealed that numerous genes involved the IFN and inflammatory response were upregulated in the cells that express N1-Dicer, which is likely due to an increased activation of the NF κ B pathway. Based on these findings, the authors propose that Dicer may act as antiviral molecule using its helicase domain, which representing a novel non-canonical function of Dicer.

****Major comments:****

1. The results from experiments with SARS-CoV2 are intriguing (Fig.2). The authors speculated that NF κ b activation is in favor of the replication of this virus. It would be interesting to see the infection and replication of SARS-CoV2 in PKR deficient cells and cells expressing PKR mutants (as described in Fig.5). The results may

prove/disapprove the authors' speculation and yield additional findings.

2. Western blot analysis. In the method section, it is stated that proteins were quantified with Bradford method and equal loading was verified by Ponceau S staining. The members of also probed with gamma-tubulin (It was stated that antibodies against alpha-tubulin was used in the method section) as a loading control, however, the bend intensity of tubulin shows great variations among different lanes in several figures while Ponceau S staining is similar (Fig.s, 4, 5, and 8). The differences compromise the accuracy of the results.

3. RNA-seq analysis revealed that Dicer N1 cells have significantly increased expression levels of signaling molecules in type I IFN response even in uninfected cells. While this provides a potential explanation for the antiviral phenotype of N1-Dicer cells. I wonder why the expression levels of type I IFNs (probably the most potent antiviral molecules) were not analyzed in WT and Dicer N1 cells.

Measurement of the levels of IFN α and IFN β by ELISA in the cells before and after infection could provide the important and direct data to support their conclusion.

4. While the data presented in Fig. 5 provides convincing evidences that the antiviral activity of mediated by PKR against SINV is independent of its kinase activity in N1-Dicer cells. An interesting question is that whether antiviral activity associated with PKR is N1-Dicer dependent,, which could be addressed by comparing the viral infection of NoDice Δ PKR and NoDicer expressing PKR mutants.

5. In the concluding paragraph of the discussion, the authors presented an oversimplified discription of a complex model that involves a crosstalk between IFN-I and RNAi and Dicer-PKR interaction, which is difficult for the reader to compose a clear picture of mechanisms involved. It could be helpful to use a schematic illustration to summarize the action model of PKR incorporated with the canonical and non-canonical Dicer functions.

****Minor comments:****

1. It stated that NoDice FHA-Dicer WT #4 and NoDice FHA:Dicer N1 110 #6 are referred to as Dicer WT and Dicer N1 cells (p.6). For simplicity, Dicer WT and Dicer N1 cells should be used throughout manuscript, including in all figures. The labels in the figures are difficult to read and are confusing in some cases.

2. It is to note that p-PKR was only detected at in N1-Dicer cells at 24 hpi (Fig.8A). This is an interesting observation that was not discussed. It appears that this could be due to a delayed viral replication since these cells are already in an elevated antiviral state. This possibility could be tested by examining viral replication and dsRNA accumulation at more time points in the experiments described in Fig.1.

3. The authors may point out the limitations of the studies. For examples, all cells used in the study are engineered HEK cell lines and were tested with limited number of viruses. As such, the observations may reflect Dicer-PKR interaction under artificially

overexpressed conditions, but how the model established from the current study applies to primary cells require further investigation.

2. Significance:

Significance (Required)

The findings reported in this study shed some new light on a long-debated issue regarding the potential roles of RNAi as physiologically relevant antiviral mechanism in mammals. Identification of a new antiviral function of Dicer helicase domain via interaction with PKR is a new advancement of the field, and it also adds a new dimension to a complex subject that overlaps of innate immunity , RNA biology, and developmental biology associated with Dicer.

Field of expertise: Innate immunity, cell signaling, cytokine biology

Areas that that I do not have sufficient expertise to evaluate: Small RNA cloning, sequencing and, analysis.

3. How much time do you estimate the authors will need to complete the suggested revisions:

Estimated time to Complete Revisions (Required)

(Decision Recommendation)

Between 3 and 6 months

Yes

Review #3

1. Evidence, reproducibility and clarity:

Evidence, reproducibility and clarity (Required)

This work by Baldaccini et al. explores the interplay between Dicer and the antiviral protein PKR in the context of viral infection. It builds on a previous publication of the team which demonstrates that the Dicer helicase interacts with multiple RNA binding proteins, including PKR (see Montavon et al.). In this work from 2021, they demonstrate that an artificially-truncated form of Dicer (Dicer-N1) lacking part of the helicase is antiviral against RNA viruses in a PKR-dependent fashion. This was an interesting finding because the field largely assumed that Dicer-N1 performs its antiviral function via canonical dicing of dsRNA, as part of an antiviral RNAi pathway. The present manuscript follows up on this initial discovery and deciphers the specifics of Dicer-N1 antiviral phenotype, as well as delineates the interplay between Dicer's helicase and PKR. The authors main claims are as follow:

- i) Dicer-N1 antiviral effect does not require its catalytic activity, therefore is completely RNAi-independent.
- ii) Neither does it require canonical PKR activation, but relies instead on NF-kB-driven inflammation. The origin of this inflammation is not studied.
- ii) Truncated Dicers other than Dicer-N1 are antiviral through RNAi, but are also PKR-dependent.

The authors claims are mostly supported by the data, although I suggest below some improvements regarding experimental approaches and data presentation. This work details in an interesting manner the interplay between the machinery of RNAi and the classical pathway of innate immunity (PKR). As explained by the authors, there is solid data in the literature demonstrating the mutual exclusivity of IFN and antiviral RNAi in differentiated cells. This mostly goes through the receptors LGP2, which inhibits dsRNA dicing by Dicer. The authors data suggest that, conversely, Dicer may play a role in preventing the unwanted activation of PKR (a non-canonical activation leading to inflammation). Given that PKR activation does not depend on virus, the authors discuss potential mechanisms of PKR triggering. This is an interesting topic

that deserves further investigation (not necessarily within the frame of this work - it can be a follow-up). Another interesting piece of information is that different truncated Dicers behave differently with respect to implementing antiviral RNAi. Whilst Dicer-N1 isn't proficient in doing so, the other forms are. It shows that lab-generated truncations do not fully recapitulate what is observed with existing truncated Dicers (DicerO and aviD).

Experimental design and data interpretation

1. The authors should compare infection between different cell lines across a range of time points (ie, a virus growth curve). In Fig 4E for example, I worry that cells expressing or not PKR will reach the plateau of viral particle accumulation at different time points. One could imagine that cells lacking PKR do not show any differences in particle production at 24h, but do at earlier time points.
2. Western blots should be accompanied with proper quantifications plotted as bar graph with biological replicates (p-PKR, p-eIF2a and capsid).
3. Microscopy images should be properly quantified across biological replicates (Fig 1&2 for the J2 staining, for example).
4. Confounding factors hinder the interpretation of siRNA accumulation (Suppl Fig 2): i) the efficiency of dsRNA dicing from different Dicers will generate different amounts of siRNAs from a given amount of dsRNA and ii) the higher antiviral response translates into decreased infection, so decreased dsRNA substrate. I suggest that the authors normalise the amount of viral siRNAs over the total amount of viral genomes. This should allow to assess if Dicer-N1 is better at dicing dsRNA than WT in these conditions.
5. In Fig 8, the authors should verify that phospho-p65 increase depends on PKR by repeating the experiment in PKR KO cells.

Data representation

1. Levels of phospho-PKR and eIF2a need to be normalised on the total amount of PKR and eIF2a, respectively. The authors should quantify the blots and present bar graphs with biological replicates and statistics.
2. Could the authors add the names of representative genes on the volcano plots of Fig 7?

Points of discussion

1. In Fig 4C, catalytically-dead mutants of truncated Dicers (other than N1) do not display an antiviral effect. Presumably, such proteins implement canonical antiviral RNAi. Is there a reason why the authors interpret this data as Dicers being "partially"

antiviral through RNAi (1. 92). This data instead suggest that is it totally dependent on RNAi.

2. Gurung et al. demonstrate that PKR is activated in Dicer KO mouse ES cells, which results in phosphorylation of eIF2a at steady-state. This is different from the authors' data, in which PKR activation does not affect eIF2a phosphorylation. Could the authors discuss this discrepancy?

3. Do the authors expect that truncated Dicers other than N1 trigger an inflammatory response such as the one described for N1? Would it be possible to have this antiviral inflammatory response in conjunction with antiviral RNAi?

2. Significance:

Significance (Required)

This is a study that conceptually advances the field of antiviral RNAi in mammals, including its interplay with the machinery of innate immunity. It is of interest for virologists and immunologists. My expertise is centered on the mechanisms of innate immunity in mammalian cells, including antiviral RNAi.

3. How much time do you estimate the authors will need to complete the suggested revisions:

Estimated time to Complete Revisions (Required)

(Decision Recommendation)

Between 1 and 3 months

Yes

Revision Plan

Manuscript number: RC-2023-02147

Corresponding author(s): Sébastien Pfeffer

1. General Statements [optional]

We would like to thank the reviewers for their evaluation of our work and for suggesting experiments that will help improve the quality of our manuscript. Overall, we think that we can respond to most request with some exceptions as indicated below.

2. Description of the planned revisions

Reviewer #1

The study largely focuses on the use of a 293 cell line that lacks a functional Dicer gene originally identified by the Cullen group. Baldaccini use this cell line, referred to as NoDice cells, to reconstitute various Dicer isoforms that have thus far been described in a variety of settings (e.g., stem cells and oocytes). Collectively, these data demonstrate the capacity of certain N-terminal truncations of Dicer to inhibit Sindbis virus and reduce the presence of viral dsRNA, supporting some of the observations made thus far concerning an antiviral role for mammalian Dicer. For other viruses, this impact was significantly more modest (SFV reduction is less than a log) or was not observed at all (VSV and SARS-CoV-2). The authors then go on to characterize the nature of the observed antiviral activity and ultimately implicate PKR and the induction of NF- κ B in priming the cell's antiviral defenses. Importantly, the group also found that this antiviral activity neither required the nuclease activity of Dicer nor the kinase activity of PKR - providing evidence against antiviral RNAi in mammals. In all, the data would seem to suggest that Dicer can act as a dsRNA sensor and can mediate the activation of an NF- κ B response - akin to what is observed in response to NOD or some TLR engagement. In all, it is the opinion of this reviewer that this work brings additional clarity to a concept that remains controversial in the field and therefore embodies something meaningful for the community.

With that said, there are a few issues that require additional attention. The first of these is textual. The introduction of the paper accurately describes the evidence in support of mammalian RNAi but does not invest the same time in discussing the data to the contrary. For example, Seo et al demonstrated that virus infection results in poly-adp-ribosylation of RISC preventing RNAi activity (PMID: 24075860), Uhl et al showed that IFN-induced ADAR1 resolves dsRNA in the cell and prevents RNAi (PMID: 37017521), and Tsai et al showed that virus-derived small RNAs are not loaded into the RISC in a manner that would enable antiviral activity (PMID 29903832). None of this work is referenced in this manuscript and it generates an unbalanced introduction as it relates to the controversy surrounding the idea of RNAi in mammals.

Reply: We thank the reviewer for their positive comments and suggestions. In the revised version of this manuscript, we will rewrite the introduction to take into

Revision Plan

account the published data that are not in favor of an antiviral role of RNAi in mammals and we will add the suggested references

Reviewer #2

Whether RNAi is used as an antiviral mechanism in mammals has been a hotly debated issue. The research team previously published several papers on the roles of Dicer in siRNA/miRNA biogenesis and in antiviral responses. They have recently reported that the helicase domain of human Dicer specifically interacts with several proteins that are involved in the IFN response, including PKR. In this study, Baldaccini et al. investigated the involvement of Dicer in antiviral response using various mutants of human Dicer. They showed that deletion mutants of helicase domain exhibit antiviral activity that requires the presence of PKR. They further demonstrated that one of the mutants, N1-Dicer showed antiviral activity in an RNAi-independent manner but depending on the presence of either native PKR or kinase deficient mutants. Transcriptomic analysis revealed that numerous genes involved the IFN and inflammatory response were upregulated in the cells that express N1-Dicer, which is likely due to an increased activation of the NF κ B pathway. Based on these findings, the authors propose that Dicer may act as antiviral molecule using its helicase domain, which representing a novel non-canonical function of Dicer.

Major comments:

1. The results from experiments with SARS-CoV2 are intriguing (Fig.2). The authors speculated that NF κ b activation is in favor of the replication of this virus. It would be interesting to see the infection and replication of SARS-CoV2 in PKR deficient cells and cells expressing PKR mutants (as described in Fig.5). The results may prove/disapprove the authors' speculation and yield additional findings.

Reply: We thank the reviewer for this suggestion. We have cells that are double knock-out for Dicer and PKR (NoDice Δ PKR) that were transduced to stably express Dicer WT or Dicer N1 and further transduced to express ACE2. We will infect those cell lines with SARS-CoV-2, which will allow us to see whether the difference in viral accumulation can still be observed in the absence of PKR. However, it might prove more difficult to reconstitute PKR expression (WT or mutants) in these cells since they are already transduced twice with two different constructs (Dicer and ACE2).

2. Western blot analysis. In the method section, it is stated that proteins were quantified with Bradford method and equal loading was verified by Ponceau S staining. The members of also probed with gamma-tubulin (It was stated that antibodies against alpha-tubulin was used in the method section) as a loading control, however, the bend intensity of tubulin shows great variations among different lanes in several figures while Ponceau S staining is similar (Fig.s, 4, 5, and 8). The differences compromise the accuracy of the results.

Revision Plan

Reply: We apologize for the difference in Tubulin signal in some of our western blots. There are several possibilities to explain those inconsistencies between Ponceau staining and Tubulin blotting, including an effect of viral infection on Tubulin expression. To remove ambiguities around this issue, we could quantify the signal across several blot replicates and provide the quantification after normalization. In addition, we would like to stress that regarding quantification of the infection, we think that the plaque assay experiments are more reliable than quantification of western blot signals.

4. While the data presented in Fig. 5 provides convincing evidences that the antiviral activity of mediated by PKR against SINV is independent of its kinase activity in N1-Dicer cells. An interesting question is that whether antiviral activity associated with PKR is N1-Dicer dependent, which could be addressed by comparing the viral infection of NoDice Δ PKR and NoDicer expressing PKR mutants.

Reply: Yes indeed, we have generated NoDice/ Δ PKR cells expressing PKR WT or mutant and we will infect them with SINV to confirm whether the presence of Dicer N1 is needed for the observed phenotype.

5. In the concluding paragraph of the discussion, the authors presented an oversimplified discription of a complex model that involves a crosstalk between IFN-I and RNAi and Dicer-PKR interaction, which is difficult for the reader to compose a clear picture of mechanisms involved. It could be helpful to use a schematic illustration to summarize the action model of PKR incorporated with the canonical and non-canonical Dicer functions.

Reply: We will add a schematic model in the revised version of our manuscript to summarize our main findings.

Minor comments:

1. It stated that NoDice FHA-Dicer WT #4 and NoDice FHA:Dicer N1 110 #6 are referred to as Dicer WT and Dicer N1 cells (p.6). For simplicity, Dicer WT and Dicer N1 cells should be used throughout manuscript, including in all figures. The labels in the figures are difficult to read and are confusing in some cases.

Reply: This will be changed in the revised version to increase the clarity of the figures.

2. It is to note that p-PKR was only detected at in N1-Dicer cells at 24 hpi (Fig.8A). This is an interesting observation that was not discussed. It appears that this could be due to a delayed viral replication since these cells are already in an elevated antiviral state. This possibility could be tested by examining viral replication and dsRNA accumulation at more time points in the experiments described in Fig.1.

Revision Plan

Reply: We have performed a kinetic of infection at more time points and we will incorporate these experiments in the revision.

3. The authors may point out the limitations of the studies. For examples, all cells used in the study are engineered HEK cell lines and were tested with limited number of viruses. As such, the observations may reflect Dicer-PKR interaction under artificially overexpressed conditions, but how the model established from the current study applies to primary cells require further investigation.

Reply: This is indeed important; we will add a sentence about this in the discussion.

Reviewer #3

This work by Baldaccini et al. explores the interplay between Dicer and the antiviral protein PKR in the context of viral infection. It builds on a previous publication of the team which demonstrates that the Dicer helicase interacts with multiple RNA binding proteins, including PKR (see Montavon et al.). In this work from 2021, they demonstrate that an artificially-truncated form of Dicer (Dicer-N1) lacking part of the helicase is antiviral against RNA viruses in a PKR-dependent fashion. This was an interesting finding because the field largely assumed that Dicer-N1 performs its antiviral function via canonical dicing of dsRNA, as part of an antiviral RNAi pathway. The present manuscript follows up on this initial discovery and deciphers the specifics of Dicer-N1 antiviral phenotype, as well as delineates the interplay between Dicer's helicase and PKR. The authors main claims are as follow:

- i) Dicer-N1 antiviral effect does not require its catalytic activity, therefore is completely RNAi-independent.
- ii) Neither does it require canonical PKR activation, but relies instead on NF- κ B-driven inflammation. The origin of this inflammation is not studied.
- ii) Truncated Dicers other than Dicer-N1 are antiviral through RNAi, but are also PKR-dependent.

The authors claims are mostly supported by the data, although I suggest below some improvements regarding experimental approaches and data presentation. This work details in an interesting manner the interplay between the machinery of RNAi and the classical pathway of innate immunity (PKR). As explained by the authors, there is solid data in the literature demonstrating the mutual exclusivity of IFN and antiviral RNAi in differentiated cells. This mostly goes through the receptors LGP2, which inhibits dsRNA dicing by Dicer. The authors data suggest that, conversely, Dicer may play a role in preventing the unwanted activation of PKR (a non-canonical activation leading to inflammation). Given that PKR activation does not depend on virus, the authors discuss potential mechanisms of PKR triggering. This is an interesting topic that deserves further investigation (not necessarily within the frame of this work - it can be a follow-up). Another interesting piece of information is that different truncated Dicers behave differently with respect to implementing antiviral RNAi. Whilst Dicer-N1 isn't proficient in doing so, the other forms are. It shows that lab-generated truncations do not fully recapitulate what is observed with existing truncated Dicers (DicerO and aviD).

Revision Plan

Experimental design and data interpretation

1. The authors should compare infection between different cell lines across a range of time points (ie, a virus growth curve). In Fig 4E for example, I worry that cells expressing or not PKR will reach the plateau of viral particle accumulation at different time points. One could imagine that cells lacking PKR do not show any differences in particle production at 24h, but do at earlier time points.

Reply: This is an interesting suggestion; we can perform a kinetic experiment by looking at more time points to address this point. This will allow us to determine the time needed for every cell line to reach the plateau of infection.

2. Western blots should be accompanied with proper quantifications plotted as bar graph with biological replicates (p-PKR, p-eIF2a and capsid).

Reply: We have biological replicates for our western blot experiments, and we will quantify those to better determine the observed changes. However, in the case of p-eIF2a, we do not think it is pertinent to measure it since there are other kinases than PKR that are known to induce eIF2a phosphorylation upon SINV infection. It might therefore not prove very informative to precisely quantify this particular signal.

4. Confounding factors hinder the interpretation of siRNA accumulation (Suppl Fig 2): i) the efficiency of dsRNA dicing from different Dicers will generate different amounts of siRNAs from a given amount of dsRNA and ii) the higher antiviral response translates into decreased infection, so decreased dsRNA substrate. I suggest that the authors normalise the amount of viral siRNAs over the total amount of viral genomes. This should allow to assess if Dicer-N1 is better at dicing dsRNA than WT in these conditions.

Reply: This is a valid concern and we agree that it is important to be able to normalize small RNA reads between conditions before reaching a conclusion. The problem is that there is no easy way to do this since we do not get a direct measurement of viral genomes accumulation from our small RNA sequencing data. To better compare the two conditions, we could normalize the individual viral siRNA to the total number of viral reads. Another problem that we face is that we are looking here at the AGO-loaded small RNAs, which makes it more difficult to assess dicing efficiency since not every generated siRNA might be loaded into an Argonaute protein. In fact, this has been proposed by the Cullen laboratory in a paper published in 2018 (Tsai et al. doi: 10.1261/rna.066332.118). They showed that although viral siRNAs were generated during IAV infection, those were inefficiently loaded and thus did not significantly impact the infection.

5. In Fig 8, the authors should verify that phospho-p65 increase depends on PKR by repeating the experiment in PKR KO cells.

Revision Plan

Reply: Yes, good point. We will check what happens to phosphorylation of p65 in PKR KO cells. In addition, we can also measure the effect on a known NF- κ B target by RT-qPCR (e.g. PTGS2).

Data representation

1. Levels of phospho-PKR and eIF2a need to be normalised on the total amount of PKR and eIF2a, respectively. The authors should quantify the blots and present bar graphs with biological replicates and statistics.

Reply: As mentioned above in our reply to point 2, we can add the quantification for phospho PKR, but we do not think it is pertinent to do it for eIF2a.

2. Could the authors add the names of representative genes on the volcano plots of Fig 7?

Reply: Yes, this will be done.

Points of discussion

1. In Fig 4C, catalytically-dead mutants of truncated Dicers (other than N1) do not display an antiviral effect. Presumably, such proteins implement canonical antiviral RNAi. Is there a reason why the authors interpret this data as Dicers being "partially" antiviral through RNAi (l. 92). This data instead suggest that is it totally dependent on RNAi.

Reply: Indeed, and we do not say the contrary. It seems that some of this helicase-truncated Dicer proteins can act through RNAi. However, they also depend on PKR, so in the end it might be a combination of the two that allows their antiviral effect.

2. Gurung et al. demonstrate that PKR is activated in Dicer KO mouse ES cells, which results in phosphorylation of eIF2a at steady-state. This is different from the authors' data, in which PKR activation does not affect eIF2a phosphorylation. Could the authors discuss this discrepancy?

Reply: The problem that we face here is that SIN3 is known to also activate GCN2 and therefore eIF2a phosphorylation does not strictly rely on PKR in our experimental conditions. In addition, we did not check eIF2a phosphorylation in Dicer KO cells, but we always compare Dicer WT and Dicer N1 expressing cells.

3. Do the authors expect that truncated Dicers other than N1 trigger an inflammatory response such as the one described for N1? Would it be possible to have this antiviral inflammatory response in conjunction with antiviral RNAi?

Reply: This goes back to Point 1 mentioned previously. We think indeed that there might be a dual action of Dicer and that it will be important to check whether in

other cellular systems or animal model such a phenomenon can be observed as well. This is a point that we did address in the discussion of our manuscript (line 522-525).

3. Description of the revisions that have already been incorporated in the transferred manuscript

4. Description of analyses that authors prefer not to carry out

Reviewer #1

The second issue that would further strengthen this paper relates to the fact that the authors spend a considerable amount of time discussing the data of Figure 6 and 7 as conditions that are defined by a Dicer that can not be processive in its nuclease activity (WT) vs. one that can (N1). However, there seems to be little consideration about the fact that the introduction of WT Dicer into these cells also restores miRNA biology whereas N1 appears to remain only partially functional (based on the data of Fig 3E). Given this, it seems the authors should verify that the high baseline of NFkB signaling that is being observed when comparing WT to N1 is not a product of restored miRNA function in WT cells, in contrast to the hypotheses outlined in the manuscript. This could be addressed by silencing Drosha or DGCR8 in the Dicer knockout cells prior to their reconstitution of Dicer. In the opinion of this reviewer, this experimental control would significantly strengthen the conclusions the authors are making here.

Reply: This would indeed be an ideal experiment to rule out the contribution of miRNAs in the observed phenotype. We believe however that this particular experiment would prove difficult to realize given that we reconstitute Dicer expression by lentiviral transduction and keep the cells under selection for a couple of weeks before using them for further experiments. This time frame is therefore not compatible with the use of siRNA to knock-down Drosha or DGCR8. Alternatively, we could knock them out by CRISPR-Cas9, but this would take too long and is not feasible in the frame of this work.

We can however address the concern regarding the role played by miRNAs in the observed phenotype of the Dicer N1 cells. Indeed, we can determine the miRNA profile from our small RNA sequencing data and compare them between the Dicer WT and Dicer N1 cells. We have done this comparison and could not find striking differences in miRNA expression between the two cell lines. We will add this additional piece of evidence in our revised manuscript.

Reviewer #2

3. RNA-seq analysis revealed that Dicer N1 cells have significantly increased expression levels of signaling molecules in type I IFN response even in uninfected cells. While this provides a potential explanation for the antiviral phenotype of N1-Dicer cells. I wonder why the expression levels of

Revision Plan

type I IFNs (probably the most potent antiviral molecules) were not analyzed in WT and Dicer N1 cells. Measurement of the levels of IFN α and IFN β by ELISA in the cells before and after infection could provide the important and direct data to support their conclusion.

Reply: This an interesting suggestion but unfortunately, we do not believe that it would be possible to quantify IFN α and IFN β by ELISA in the cell line that we used in our experiments. Indeed, the level of expression might just be too low to be able to measure something meaningful. We could measure the induction of IFN β expression at the mRNA level by RT-qPCR though. However, we do not believe that the observed increased expression of genes that belong to the type I IFN response is solely the effect of an increased production of IFN. These genes are also under the control of other transcription factors, including NF-kB for some of them, and it might prove difficult to make a direct link with IFN α or IFN β production.

Reviewer #3

3. Microscopy images should be properly quantified across biological replicates (Fig 1&2 for the J2 staining, for example).

Reply: We could do a proper quantification of the J2 signal across replicates, but we do not think it would bring much to our message. Here, we mostly used J2 staining as a qualitative indication that the infection was impacted or not. We have a proper quantification of the effect with our plaque assay experiments, which are way more robust to determine the levels of infection between conditions.

Dear Dr. Pfeffer,

Thank you for submitting for consideration by the EMBO Journal your manuscript (EMBOJ-2023-115792) along with the reports of the three referees who evaluated it at Review Commons. I have now carefully read your manuscript, the referee comments, and your revision plan, and I have also discussed them with the other members of our editorial team, as well as with an external expert whom we contacted for additional advice on the potential suitability of the manuscript for our journal.

The referees are positive about the manuscript, and they mention that the findings are interesting and the advance over the previous literature considerable. They point out that this work constitutes a significant contribution to an important and contentious topic, which was further endorsed by our advisor.

Given the referees' positive comments and recommendations, I would like to invite you to submit a revised version of your manuscript, addressing the comments of all three reviewers along the lines described in your revision plan. I would like to note that we do understand that the second suggestion of referee #1 (i.e., silencing Drosha or DGCR8 in the Dicer knockout cells prior to Dicer reconstitution to rule out the contribution of the restored miRNA system to the observed phenotype) is not feasible in your experimental setup, and -following our discussion with the referee- we will only require your suggested analysis of miRNA profiles in wild-type and Dicer-N1 cells as further evidence to support your conclusions. Please also include the (temporary) sequencing data access information in the Data availability section of your revised manuscript for the information of the referees.

I should add that it is EMBO Journal policy to allow only a single round of major revision, and acceptance of your manuscript will therefore depend on the completeness of your responses in this revised version. If you have any questions or comments, we can also discuss the revisions in a video chat, if you like.

We generally allow three months as standard revision time (10th January 2024). As a matter of policy, competing manuscripts published during this period will not negatively impact our assessment of the conceptual advance presented by your study. However, we request that you contact us as soon as possible upon publication of any related work, to discuss how to proceed. Should you foresee a problem in meeting this three-month deadline, please let us know in advance and we may be able to grant an extension.

Thank you for the opportunity to consider your work for publication in the EMBO Journal. I look forward to your revision.

Yours sincerely,

Instructions for preparing your revised manuscript

1. When you are ready to submit the revision, please upload:

- A Word file of the manuscript text (including legends of main Figures, EV Figures and Tables). Please make sure that changes are highlighted (or "tracked") to be clearly visible.

- Individual production-quality figure files (one file per figure). When assembling your figures, please refer to our figure preparation guidelines in order to ensure proper formatting and readability in print as well as on screen:

If the data shown in a figure are obtained from n {less than or equal to} 2, please use scatter plots showing the individual data points.

- i. the name of the statistical test used to generate error bars and P values
- ii. the number (n) of independent experiments (please specify technical or biological replicates) underlying each data point (discussion of statistical methodology can be reported in the Materials and Methods section, but figure legends should contain a basic description of n , P , and the test applied)
- iii. the nature of the bars and error bars (s.d., s.e.m.).

- A point-by-point response to the referees' comments, with a detailed description of the changes made (as a word file). All referees' concerns must be fully addressed and their suggestions taken on board. When preparing your letter of response to the referees' comments, please bear in mind that this will form part of the Review Process File and will therefore be available online to the community. Please note that you have the possibility to opt out of the transparent process at any stage prior to publication by letting the editorial office know (contact@embojournal.org); if you do opt out, the Review Process File link will point to the following statement: "No Review Process File is available with this article, as the authors have chosen not to make the review process public in this case.". For more details on our Transparent Editorial Process, please visit our website: <https://www.embopress.org/page/journal/14602075/authorguide#transparentprocess>

- Expanded View (EV) files (replacing Supplementary Information) that are collapsible/expandable online. A maximum of 5 EV Figures can be typeset. EV Figures should be cited as "Figure EV1, Figure EV2" etc. in the text, and their respective legends should be included in the manuscript file after the legends of regular figures. See detailed instructions regarding Expanded View files here: <https://www.embopress.org/page/journal/14602075/authorguide#expandedview>

- For the figures that you do NOT wish to display as Expanded View figures, they should be bundled together with their legends in a single PDF file called "Appendix", which should start with a short Table of Contents (including page numbers). Appendix figures should be referred to in the main text as: "Appendix Figure S1, Appendix Figure S2" etc. Please see detailed instructions here: <https://www.embopress.org/page/journal/14602075/authorguide#expandedview>

- A complete author checklist, which you can download from our author guidelines (<https://www.embopress.org/page/journal/14602075/authorguide>). Please note that the checklist will also be part of the Review Process File.

2. Please note that no statistics should be calculated if $n=2$.

3. Before submitting your revision, primary datasets (and computer code, where appropriate) produced in this study need to be deposited in appropriate public databases (see <https://www.embopress.org/page/journal/14602075/authorguide#dataavailability>). Specifically, we would kindly ask you to provide public access to the following datasets/data:

- Small RNA and mRNA sequencing data.

The accession numbers and database should be listed in a formal "Data availability" section (placed after Materials and Methods) that follows the model below (see also <https://www.embopress.org/page/journal/14602075/authorguide#dataavailability>):

Data availability

- RNA-seq data: Gene Expression Omnibus GSE46843 (<https://www.ncbi.nlm.nih.gov/geo/query/acc.cgi?acc=GSE46843>)
- [data type]: [name of the resource] [accession number/identifier/doi] ([URL or identifiers.org/DATABASE:ACCESSION])

*** Note: all links should resolve to a page where the data can be accessed. ***

*** Note: the Data Availability Section is restricted to new primary data that are part of this study. ***

4. Please check that the title and the abstract of the manuscript are brief, yet explicit, even to non-specialists. The length of the title should not exceed 100 characters (including spaces), and the abstract should be a single paragraph not exceeding 175 words.

5. Please also note our reference format: <https://www.embopress.org/page/journal/14602075/authorguide#referencesformat>.

7. Please remember: digital image enhancement is acceptable practice, as long as it accurately represents the original data and conforms to community standards. If a figure has been subjected to significant electronic manipulation, this must be noted in the figure legend or in the "Materials and Methods" section. The editors reserve the right to request original versions of figures and the original images that were used to assemble the figure.

8. Our journal encourages inclusion of data citations in the reference list to directly cite datasets that were obtained from public

databases. Data citations in the article text are distinct from normal bibliographical citations and should directly link to the database records from which the data can be accessed. In the main text, data citations are formatted as follows: "Data ref: Smith et al, 2001" or "Data ref: NCBI Sequence Read Archive PRJNA342805, 2017". In the Reference list, data citations must be labeled with "[DATASET]". A data reference must provide the database name, accession number/identifiers, and a resolvable link to the landing page from which the data can be accessed at the end of the reference. Further instructions are available at: <https://www.embopress.org/page/journal/14602075/authorguide#referencesformat>.

9. We request authors to consider both actual and perceived competing interests. Please review our policy (<https://www.embopress.org/page/journal/14602075/authorguide#conflictsofinterest>) and update your competing interests statement if necessary. Please name this section 'Disclosure and competing interests statement' and place it after the Acknowledgements section.

10. Please note that all corresponding authors are required to provide an ORCID ID upon submission of a revised manuscript (<https://orcid.org/>). Please find instructions on how to link your ORCID ID to your account in our manuscript tracking system in our Author guidelines (<https://www.embopress.org/page/journal/14602075/authorguide#authorshipguidelines>).

11. We use CRediT to specify the contributions of each author in the journal submission system. CRediT replaces the author contribution section, which should be removed from the manuscript. Please use the free text box to provide more detailed descriptions. See also guide to authors: <https://www.embopress.org/page/journal/14602075/authorguide#authorshipguidelines>.

13. We would also welcome the submission of cover suggestions or motifs to be used by our Graphics Illustrator in designing a cover.

14. Please use the link below to submit your revision:
<https://emboj.msubmit.net/cgi-bin/main.plex>

Yours sincerely,

Rev_Com_number: RC-2023-02147
New_manu_number: EMBOJ-2023-115792
Corr_author: Pfeffer
Title: Canonical and non-canonical contributions of human Dicer helicase domain in antiviral defense

Replies to reviewers

Reviewer #1 (Evidence, reproducibility and clarity (Required)):

The study largely focuses on the use of a 293 cell line that lacks a functional Dicer gene originally identified by the Cullen group. Baldaccini use this cell line, referred to as NoDice cells, to reconstitute various Dicer isoforms that have thus far been described in a variety of settings (e.g., stem cells and oocytes). Collectively, these data demonstrate the capacity of certain N-terminal truncations of Dicer to inhibit Sindbis virus and reduce the presence of viral dsRNA, supporting some of the observations made thus far concerning an antiviral role for mammalian Dicer. For other viruses, this impact was significantly more modest (SFV reduction is less than a log) or was not observed at all (VSV and SARS-CoV-2). The authors then go on to characterize the nature of the observed antiviral activity and ultimately implicate PKR and the induction of NF- κ B in priming the cell's antiviral defenses. Importantly, the group also found that this antiviral activity neither required the nuclease activity of Dicer nor the kinase activity of PKR - providing evidence against antiviral RNAi in mammals. In all, the data would seem to suggest that Dicer can act as a dsRNA sensor and can mediate the activation of an NF- κ B response - akin to what is observed in response to NOD or some TLR engagement. In all, it is the opinion of this reviewer that this work brings additional clarity to a concept that remains controversial in the field and therefore embodies something meaningful for the community.

With that said, there are a few issues that require additional attention. The first of these is textual. The introduction of the paper accurately describes the evidence in support of mammalian RNAi but does not invest the same time in discussing the data to the contrary. For example, Seo et al demonstrated that virus infection results in poly-adp-ribosylation of RISC preventing RNAi activity (PMID: 24075860), Uhl et al showed that IFN-induced ADAR1 resolves dsRNA in the cell and prevents RNAi (PMID: 37017521), and Tsai et al showed that virus-derived small RNAs are not loaded into the RISC in a manner that would enable antiviral activity (PMID 29903832). None of this work is referenced in this manuscript and it generates an unbalanced introduction as it relates to the controversy surrounding the idea of RNAi in mammals.

Reply: We thank the reviewer for their positive comments and suggestions. In the revised version of this manuscript, we have rewritten the introduction to take into account the published data that are not in favor of an antiviral role of RNAi in mammals and we have added the suggested references.

The second issue that would further strengthen this paper relates to the fact that the authors spend a considerable amount of time discussing the data of Figure 6 and 7 as conditions that are defined by a Dicer that can not be processive in its nuclease activity (WT) vs. one that can (N1). However, there seems to be little consideration about the fact that the introduction of WT Dicer into these cells also restores miRNA biology whereas N1 appears to remain only partially functional (based on the data of Fig 3E). Given this, it seems the authors should verify that the high baseline of NF κ B signaling that is being observed when comparing WT to N1 is not a product of restored miRNA function in WT cells, in contrast to the hypotheses outlined in the manuscript. This could be addressed by silencing Drosha or DGCR8 in the Dicer knockout cells prior to their reconstitution of Dicer. In the opinion of this reviewer, this experimental control would significantly strengthen the conclusions the authors are making here.

Reply: This would indeed be an ideal experiment to rule out the contribution of miRNAs in the observed phenotype. We believe however that this particular experiment would prove difficult to realize given that we reconstitute Dicer expression by lentiviral transduction and keep the cells under selection for a couple of weeks before using them for further experiments. This time frame is therefore not compatible with the use of siRNA to knock-down Drosha or DGCR8. Alternatively, we could have knocked them out by CRISPR-Cas9, but this would have taken too long and was not feasible in the time allowed for revising this manuscript.

We have however addressed the concern regarding the role played by miRNAs in the observed phenotype of the Dicer N1 cells. Indeed, we determined the miRNA profiles from our small RNA sequencing data and compared them between the Dicer WT and Dicer N1 cells. This comparison did not allow us to find striking differences in miRNA expression between the two cell lines. We have added these results in the new figure EV1 (Panels B and C) and described them in the Results section on Page 11.

Reviewer #1 (Significance (Required)):

In the manuscript entitled, "Canonical and non-canonical contributions of human Dicer helicase domain in antiviral defense" Baldaccini et al. describe their findings concerning the ability of certain N-terminal deletion variants of Dicer in contributing to mammalian antiviral activity. The concept of a functional antiviral RNAi system in mammals is a contentious one with many publications including data both in support of its existence and opposing this idea. In this manuscript, Baldaccini et al. perform a wide range of well-controlled experiments to specifically address aspects of those reports to both provide clarity in what has been documented thus far and to expand on those concepts further.

Reviewer #2 (Evidence, reproducibility and clarity (Required)):

Summary

Whether RNAi is used as an antiviral mechanism in mammals has been a hotly debated issue. The research team previously published several papers on the roles of Dicer in siRNA/miRNA biogenesis and in antiviral responses. They have recently reported that the helicase domain of human Dicer specifically interacts with several proteins that are involved in the IFN response, including PKR. In this study, Baldaccini et al. investigated the involvement of Dicer in antiviral response using various mutants of human Dicer. They showed that deletion mutants of helicase domain exhibit antiviral activity that requires the presence of PKR. They further demonstrated that one of the mutants, N1-Dicer showed antiviral activity in an RNAi-independent manner but depending on the presence of either native PKR or kinase deficient mutants. Transcriptomic analysis revealed that numerous genes involved in the IFN and inflammatory response were upregulated in the cells that express N1-Dicer, which is likely due to an increased activation of the NF κ B pathway. Based on these findings, the authors propose that Dicer may act as antiviral molecule using its helicase domain, which representing a novel non-canonical function of Dicer.

Major comments:

1. The results from experiments with SARS-CoV2 are intriguing (Fig.2). The authors speculated that NF κ B activation is in favor of the replication of this virus. It would be interesting to see the infection and replication of SARS-CoV2 in PKR deficient cells and cells expressing PKR mutants (as described in Fig.5). The results may prove/disapprove the authors' speculation and yield additional findings.

Reply: We thank the reviewer for this suggestion. We transduced the hACE2 expression construct in cells that were double knock-out for Dicer and PKR (NoDice Δ PKR) and stably expressing FHA tagged Dicer WT or Dicer N1. We have infected those cell lines with SARS-CoV-2, which allowed us to see that the difference in viral accumulation between Dicer WT and Dicer N1 cells was lost in the absence of PKR, thereby confirming the PKR-dependency for Dicer N1 proviral effect on SARS-CoV-2. These results have been added in Panels F and G of Figure 5 and are discussed in the Results section on Page 17. However, we did not reconstitute PKR expression (WT or mutants) in these cells since they were already transduced twice with two different constructs (Dicer and ACE2) and performing a third transduction would have been technically challenging.

2. Western blot analysis. In the method section, it is stated that proteins were quantified with Bradford method and equal loading was verified by Ponceau S staining. The members of also probed with gamma-tubulin (It was stated that antibodies against alpha-tubulin was used in the method section) as a loading control, however, the bend intensity of tubulin shows great variations among different lanes in several figures while Ponceau S staining is similar (Figs. 4, 5, and 8). The differences compromise the accuracy of the results.

Reply: We apologize for the mix-up between gamma and alpha tubulin, we indeed used an antibody against alpha tubulin, and we corrected this in the text. Regarding the difference in Tubulin signal in some of our western blots, we have added the quantification of the normalized signals under each relevant blots (calculated from 3 biological replicates) and we also provided all the replicate experiments in the Source Data folder. In addition, we would like to stress that regarding quantification of the infection, we think that the plaque assay experiments are more reliable than quantification of western blot signals.

3. RNA-seq analysis revealed that Dicer N1 cells have significantly increased expression levels of signaling molecules in type I IFN response even in uninfected cells. While this provides a potential explanation for the antiviral phenotype of N1-Dicer cells. I wonder why the expression levels of type I IFNs (probably the most potent antiviral molecules) were not analyzed in WT and Dicer N1 cells. Measurement of the levels of IFN α and IFN β by ELISA in the cells before and after infection could provide the important and direct data to support their conclusion.

Reply: This an interesting suggestion but unfortunately, we do not believe that it would be possible to quantify IFN α and IFN β by ELISA in the cell line that we used in our experiments. Indeed, the level of expression might just be too low to be able to measure something meaningful. We did measure the induction of IFN β expression at the mRNA level by RT-qPCR though. The results indicate that there is a small increase in expression in Dicer N1 cells compared to Dicer WT cells but it is not significant (see new Figure 7F). It has been described that HEK293T cells are generally not producing high amounts of cytokines including IFN β ,

although they do respond to stimulatory treatments. We think that most of the observed changes are therefore triggered by a pro-inflammatory response via NF- κ B, and we now discuss this in the revised manuscript (see Page 20).

4. While the data presented in Fig. 5 provides convincing evidences that the antiviral activity of mediated by PKR against SINV is independent of its kinase activity in N1-Dicer cells. An interesting question is that whether antiviral activity associated with PKR is N1-Dicer dependent, which could be addressed by comparing the viral infection of NoDice Δ PKR and NoDicer expressing PKR mutants.

Reply: To address this point, we have generated NoDice Δ PKR cells expressing PKR WT or mutant and we infected them with SINV. The results, shown in Figure EV3, indicate that expression of PKR WT or mutant forms has no impact on SINV infection levels when Dicer N1 is not expressed. We could thus confirm that the antiviral activity of PKR is also Dicer N1 dependent. This is described on Page 16 of the Results section.

5. In the concluding paragraph of the discussion, the authors presented an oversimplified discription of a complex model that involves a crosstalk between IFN-I and RNAi and Dicer-PKR interaction, which is difficult for the reader to compose a clear picture of mechanisms involved. It could be helpful to use a schematic illustration to summarize the action model of PKR incorporated with the canonical and non-canonical Dicer functions.

Reply: We have now added a schematic model in the revised version of our manuscript to summarize our main findings (Figure 9).

Minor comments:

1. It stated that NoDice FHA-Dicer WT #4 and NoDice FHA:Dicer N1 110 #6 are referred to as Dicer WT and Dicer N1 cells (p.6). For simplicity, Dicer WT and Dicer N1 cells should be used throughout manuscript, including in all figures. The labels in the figures are difficult to read and are confusing in some cases.

Reply: This has been changed in the revised version of the text and figures to increase the clarity of the message.

2. It is to note that p-PKR was only detected at in N1-Dicer cells at 24 hpi (Fig.8A). This is an interesting observation that was not discussed. It appears that this could be due to a delayed viral replication since these cells are already in an elevated antiviral state. This possibility could be tested by examining viral replication and dsRNA accumulation at more time points in the experiments described in Fig.1.

Reply: This is indeed an important point. First, we confirmed that at later time points (48 and 72 hpi), the difference in viral accumulation can still be observed between Dicer N1 and Dicer WT cells (see Appendix Figure S1B). We have also performed a kinetic of infection by measuring GFP expression in cells infected with SINV-GFP at a MOI of 2 every hour for 24 h (Figure 4F). We have also measured viral accumulation by plaque assay in the same experiments at 12 and 16 hpi (Figure EV2E). To go further, we performed this analysis not only in Dicer N1 cells but also in Dicer Δ Hel1, Δ Hel2i and Δ Hel2 cells expressing or not PKR. The results indicate that in cells expressing PKR, the delayed viral replication

can be observed with every helicase mutant Dicer expressing cells, but is more pronounced in Dicer N1 and Dicer ΔHel2i cells. However, in ΔPKR cells, no significant difference could be observed, excepted for a modest effect with Dicer ΔHel2i. These observations are now mentioned and discussed in the revised text on Pages 8 and 14.

3. The authors may point out the limitations of the studies. For examples, all cells used in the study are engineered HEK cell lines and were tested with limited number of viruses. As such, the observations may reflect Dicer-PKR interaction under artificially overexpressed conditions, but how the model established from the current study applies to primary cells require further investigation.

Reply: This is indeed important; we have added a sentence about this in the discussion (on Page 26).

Reviewer #2 (Significance (Required)):

The findings reported in this study shed some new light on a long-debated issue regarding the potential roles of RNAi as physiologically relevant antiviral mechanism in mammals. Identification of a new antiviral function of Dicer helicase domain via interaction with PKR is a new advancement of the field, and it also adds a new dimension to a complex subject that overlaps of innate immunity, RNA biology, and developmental biology associated with Dicer.

Field of expertise: Innate immunity, cell signaling, cytokine biology

Areas that that I do not have sufficient expertise to evaluate: Small RNA cloning, sequencing and, analysis.

Reviewer #3 (Evidence, reproducibility and clarity (Required)):

This work by Baldaccini et al. explores the interplay between Dicer and the antiviral protein PKR in the context of viral infection. It builds on a previous publication of the team which demonstrates that the Dicer helicase interacts with multiple RNA binding proteins, including PKR (see Montavon et al.). In this work from 2021, they demonstrate that an artificially-truncated form of Dicer (Dicer-N1) lacking part of the helicase is antiviral against RNA viruses in a PKR-dependent fashion. This was an interesting finding because the field largely assumed that Dicer-N1 performs its antiviral function via canonical dicing of dsRNA, as part of an antiviral RNAi pathway. The present manuscript follows up on this initial discovery and deciphers the specifics of Dicer-N1 antiviral phenotype, as well as delineates the interplay between Dicer's helicase and PKR. The authors main claims are as follow:

- i) Dicer-N1 antiviral effect does not require its catalytic activity, therefore is completely RNAi-independent.
- ii) Neither does it require canonical PKR activation, but relies instead on NF-kB-driven inflammation. The origin of this inflammation is not studied.
- ii) Truncated Dicercs other than Dicer-N1 are antiviral through RNAi, but are also PKR-dependent.

The authors claims are mostly supported by the data, although I suggest below some improvements regarding experimental approaches and data presentation. This work details in

an interesting manner the interplay between the machinery of RNAi and the classical pathway of innate immunity (PKR). As explained by the authors, there is solid data in the literature demonstrating the mutual exclusivity of IFN and antiviral RNAi in differentiated cells. This mostly goes through the receptors LGP2, which inhibits dsRNA dicing by Dicer. The authors data suggest that, conversely, Dicer may play a role in preventing the unwanted activation of PKR (a non-canonical activation leading to inflammation). Given that PKR activation does not depend on virus, the authors discuss potential mechanisms of PKR triggering. This is an interesting topic that deserves further investigation (not necessarily within the frame of this work - it can be a follow-up). Another interesting piece of information is that different truncated Dicers behave differently with respect to implementing antiviral RNAi. Whilst Dicer-N1 isn't proficient in doing so, the other forms are. It shows that lab-generated truncations do not fully recapitulate what is observed with existing truncated Dicers (DicerO and aviD).

Experimental design and data interpretation

1. The authors should compare infection between different cell lines across a range of time points (ie, a virus growth curve). In Fig 4E for example, I worry that cells expressing or not PKR will reach the plateau of viral particle accumulation at different time points. One could imagine that cells lacking PKR do not show any differences in particle production at 24h, but do at earlier time points.

Reply: This is an interesting suggestion; we have performed a kinetic as described above in our reply to Reviewer 2's point 2. The results described in Figure 4F and EV2E show that even at earlier time points cells lacking PKR and expressing Dicer mutants do not appear to show any differences in infection levels compared to cells expressing Dicer WT.

2. Western blots should be accompanied with proper quantifications plotted as bar graph with biological replicates (p-PKR, p-eIF2a and capsid).

Reply: We have performed biological replicates for our western blot experiments, and we now quantified those to better determine the observed changes. We added the mean +/- SD of the normalized signal quantification for all relevant western blot displayed in the figures (i.e. 1B, 2E, 3F, 4B, D, 5B, D, F, 8A, B, D, EV1D and EV3A). We did not plot those as bar graphs though because it would have cluttered the figures too much. We provide all the replicates and quantification of the western blots in the Source Data folder. However, in the case of p-eIF2a, we did not think it was pertinent to measure it since there are other kinases than PKR that are known to induce eIF2a phosphorylation upon SINV infection. For the sake of simplifying the figures, which were already quite complex, we have therefore removed p-eIF2a western blots in the revised version.

3. Microscopy images should be properly quantified across biological replicates (Fig 1&2 for the J2 staining, for example).

Reply: We did not quantify the J2 signal across replicates, since we mostly used it as a qualitative indication that the infection was impacted or not. We have a proper quantification of the effect with our plaque assay experiments, which are way more robust to determine the levels of infection between conditions. We

provided the biological replicate images of J2 immunostaining in the Source Data folder.

4. Confounding factors hinder the interpretation of siRNA accumulation (Suppl Fig 2): i) the efficiency of dsRNA dicing from different Dicers will generate different amounts of siRNAs from a given amount of dsRNA and ii) the higher antiviral response translates into decreased infection, so decreased dsRNA substrate. I suggest that the authors normalise the amount of viral siRNAs over the total amount of viral genomes. This should allow to assess if Dicer-N1 is better at dicing dsRNA than WT in these conditions.

Reply: This is a valid concern and we agree that it is important to be able to normalize small RNA reads between conditions before reaching a conclusion. The problem is that there is no easy way to do this since we do not get a direct measurement of viral genomes accumulation from our small RNA sequencing data. To better compare the two conditions, we now normalized the individual viral siRNA to the total number of viral reads (see Figure 3B and Appendix Figure S3). The results indicate that there is no difference in terms of viral siRNA accumulation between Dicer WT and Dicer N1 cells.

We would like to point out that it is however difficult to really draw conclusion about dicing efficiency here, since we are looking at the AGO-loaded small RNAs, and not every generated siRNA might be loaded into an Argonaute protein. In fact, this has been proposed by the Cullen laboratory in a paper published in 2018 (Tsai et al. doi: 10.1261/rna.066332.118). They showed that although viral siRNAs were generated during IAV infection, those were inefficiently loaded and thus did not significantly impact the infection.

5. In Fig 8, the authors should verify that phospho-p65 increase depends on PKR by repeating the experiment in PKR KO cells.

Reply: Yes, good point. We have now checked what happened to phosphorylation of p65 and activity in PKR KO cells. This is shown in Figure 8B and C. We can see that p-P65 increases and that NF-kB target PTGS2 expression goes up when PKR expression is restored in NoDiceΔPKR Dicer N1 cells.

Data representation

1. Levels of phospho-PKR and eIF2a need to be normalised on the total amount of PKR and eIF2a, respectively. The authors should quantify the blots and present bar graphs with biological replicates and statistics.

Reply: see our reply to Point 2 above.

2. Could the authors add the names of representative genes on the volcano plots of Fig 7?

Reply: We have added the names of representative genes as required.

Points of discussion

1. In Fig 4C, catalytically-dead mutants of truncated Dicers (other than N1) do not display an antiviral effect. Presumably, such proteins implement canonical antiviral RNAi. Is there a reason why the authors interpret this data as Dicers being "partially" antiviral through RNAi (1. 92). This data instead suggest that is it totally dependent on RNAi.

Reply: Indeed, and we do not say the contrary. It seems that some of this helicase-truncated Dicer proteins can act through RNAi. However, they also depend on PKR, so in the end it might be a combination of the two that allows their antiviral effect.

2. Gurung et al. demonstrate that PKR is activated in Dicer KO mouse ES cells, which results in phosphorylation of eIF2a at steady-state. This is different from the authors' data, in which PKR activation does not affect eIF2a phosphorylation. Could the authors discuss this discrepancy?

Reply: The problem that we face here is that SIN3 is known to also activate GCN2 and therefore eIF2a phosphorylation does not strictly rely on PKR in our experimental conditions. In addition, we did not check eIF2a phosphorylation in Dicer KO cells, but we always compare Dicer WT and Dicer N1 expressing cells.

3. Do the authors expect that truncated Dicers other than N1 trigger an inflammatory response such as the one described for N1? Would it be possible to have this antiviral inflammatory response in conjunction with antiviral RNAi?

Reply: This goes back to Point 1 mentioned previously. We think indeed that there might be a dual action of Dicer and that it will be important to check whether in other cellular systems or animal model such a phenomenon can be observed as well. This is a point that we did address in the discussion of our manuscript.

Reviewer #3 (Significance (Required)):

This is a study that conceptually advances the field of antiviral RNAi in mammals, including its interplay with the machinery of innate immunity. It is of interest for virologists and immunologists. My expertise is centered on the mechanisms of innate immunity in mammalian cells, including antiviral RNAi.

Dear Dr. Pfeffer,

Thank you for the submission of your revised manuscript to The EMBO Journal. We have now received the comments of the three referees that were asked to re-evaluate your study (included below). As you will see, all referees are satisfied with the revision, acknowledge that the previous concerns have been addressed satisfactorily, and support publication of the study.

From the editorial side, there are a few minor changes that we need from you before we can proceed with acceptance of the manuscript:

- The funding information in the manuscript can be included in the Acknowledgements section.
- Please enter all relevant funding information in our online manuscript handling system. It should match exactly the information provided in the Acknowledgements section of your manuscript.
- Please change "Material and Methods" to "Materials and Methods".
- Please change the heading of your conflict-of-interest statement to "Disclosure and competing interests statement".
- Please make sure that the deposited datasets will be publicly available at the time of publication. The reviewer token can now be removed from the Data availability section.
- Please add the heading "EV Figure legends" before the EV Figure legends in the main manuscript file.
- The general info table at the top of your Author Checklist has not been completed; please upload a completed version of your checklist.
- The nomenclature of Appendix Figures and Tables should be "Appendix Figure S#" and "Appendix Table S#". Please update accordingly: i. the table of contents on the first page of the Appendix, ii. the Appendix Figure/Table legends/captions, and iii. the respective callouts in the main manuscript file (NB: "Appendix Table S1" instead of Supplementary Table 1).
- Please note that the requested source data (for Fig. 1A, 1D, 2E, 2F, 3A, 3B, 3G, 4C, 4E, 5C, 5E, 6A-D, 7A-E, 8E, 8F) should be uploaded before we can proceed with acceptance of the manuscript. Source data files need to be organized in one file/folder per figure and all data referring to each main Figure should be zipped together (i.e. one ZIP folder per main Figure). All source data of EV and Appendix Figures should be zipped together in a single folder.
- Please note that the final dimensions of the synopsis image should be 550 pixels (width) x 300-600 pixels (height). Could you please edit your synopsis image so that its height falls within this range?
- Please indicate the statistical test used for data analysis in the legends of Figures 6b-d; EV 4b; EV 5d-f.
- Please note that in Figure 2b there is a mismatch between the annotated p values in the figure legend and the annotated p values in the figure file that should be corrected.
- Please note that information related to n is missing in the legends of Figures 6b-d; EV 4a-b; EV 5d-e.
- Although the sample size/number of replicates ('n') is provided, please describe the nature of entity for 'n' in the legend of Figure 7e.
- Please note that the error bars are not defined in the legend of Figure EV 4a.

As soon as these issues are resolved, I will contact you again to discuss with you a few suggestions for minor textual improvements in the title, abstract and synopsis text.

Please also note that as part of the EMBO publications' Transparent Editorial Process, The EMBO Journal publishes online a Peer Review File along with each accepted manuscript. This File will be published in conjunction with your paper and will include the referee reports, your point-by-point response and all pertinent correspondence relating to the manuscript. You can opt out of this by letting the editorial office know (contact@embojournal.org). If you do opt out, the Peer Review File link will point to the following statement: "No Peer Review File is available with this article, as the authors have chosen not to make the review process public in this case."

We look forward to seeing a final version of your manuscript as soon as possible. Please use this link to submit your revision:

Link Not Available

Yours sincerely,

Referee #1:

The authors provided satisfactory answers to my questions. I therefore recommend this work for publication. It is a nice article, congratulations!

Referee #2:

The findings reported in this study shed some new light on a long-debated issue regarding the potential roles of RNAi in mammalian cells and identified a new antiviral function of Dicer helicase domain via its interaction with PKR. The authors have addressed most of my major concerns in the previous version of the manuscript.

Referee #3:

This revision adequately addressed the concerns raised during the initial submission. This reviewer is satisfied with this final version and feels it should be accepted.

Rev_Com_number: RC-2023-02147
New_manu_number: EMBOJ-2023-115792R
Corr_author: Pfeffer
Title: Non-canonical contribution of human Dicer helicase domain in antiviral innate immune response

All editorial and formatting issues were resolved by the authors.

Dear Sébastien,

I am pleased to inform you that your manuscript has been accepted for publication in The EMBO Journal.

Best regards,

Ioannis

Rev_Com_number: RC-2023-02147

New_manu_number: EMBOJ-2023-115792R1

Corr_author: Pfeffer

Title: Non-canonical contribution of human Dicer helicase domain in antiviral innate immune response